# Adversarial Advertisement in Text-to-Image Generative Models

## Abstract

As text-to-image diffusion models (T2I DMs) gain popularity, there is a growing interest in adversarial advertisement where an attacker can compromise a T2I DM and make it generate images with the implantation of the target product brands, based on users' non-advertising input prompts. However, two challenging problems in adversarial advertisement in T2I DMs remain unsolved: imperceptible adversarial advertisement and robust adversarial advertisement. To tackle the aforementioned problems, we first propose a new estimation algorithm for multivariate continuously scaled phase-type with Lévy distributions to model the intrinsic distribution of natural prompts containing advertisements. We then construct our attack by pushing non-advertising prompts toward high-density regions of the estimated MCPHL distribution, so that the perturbed prompts remain indistinguishable from natural advertising prompts. We further prove that the estimated MCPHL converges to the empirical distribution of natural prompts with advertisements. Second, we introduce a masked parameter smoothing method based on mollification theory that produces a smoothed T2I DM with a dimension-invariant certified guarantee against adversarial-advertisement degradation under model fine-tuning in high-dimensional parameter space. The masking mechanism preserves utility by avoiding unnecessary smoothing of sensitive parameters. Our theoretical analysis shows that the resulting smooth T2I DMs still support successful adversarial advertisements under model fine-tuning within the certified radius.

## 1 Introduction

Text-to-image diffusion models (T2I DMs) encode natural-language prompts into text embeddings and use the embedding to condition a denoising network, generate high-quality images (OpenAI et al., 2024; Podell et al., 2024; Rombach et al., 2022; StabilityAI, 2023; Saharia et al., 2022; Nichol et al., 2022; Ramesh et al., 2021a; Ho et al., 2020). However, recent studies have shown that T2I DMs are vulnerable to backdoor attacks (Vice et al., 2024; Liu et al., 2023a; Yang et al., 2024; Chou et al., 2023), including bias injection (Shen et al., 2024), harmful information generation (Yang et al., 2024), or utility degradation (Chou et al., 2023), while the model behaves normally without the trigger.

With evolving developments of Generative AI, T2I DM is playing an increasingly significant role in online advertising (Du et al., 2024; Zhao et al., 2024b; Vashishtha et al., 2024; Chen et al., 2021a;b; Wei et al., 2022). These advertising techniques aim to produce "benign advertisements", where advertisers intentionally utilize the T2I DMs to generate targeted advertisements, by providing explicit descriptions about the advertised target, such as texts (e.g., "a product sitting on a wooden table, outdoor") or images of the target product brand (Du et al., 2024; Zhao et al., 2024b).

In contrast, "adversarial advertisement" tampers with a T2I model, causing non-advertising prompts to quietly produce images that are naturally blended with advertisements, without user intent or consent. An adversary (e.g., a malicious marketer) has strong incentives to use this tactic to increase brand exposure, shape positive user sentiment, and ultimately raise revenue (Vice et al., 2024).

A straightforward way to implement adversarial advertising in T2I DMs is to adapt existing backdoor attack methods (Vice et al., 2024; Liu et al., 2023a; Yang et al., 2024; Chou et al., 2023) to achieve the advertisement implantation in T2I DMs. Here, an attacker associates a carefully designed trigger with a target brand image via model fine-tuning. Once the attack is completed, the victim T2I DMs generate an image with the implantation of the target image (Vice et al., 2024; Liu et al., 2023a;

Yang et al., 2024; Chou et al., 2023) upon detection of a trigger. Despite achieving remarkable performance, existing backdoor attack approaches against T2I DMs often rely on unusual, unnatural, or out-of-context prompt tokens as triggers (Liu et al., 2023a; Yang et al., 2024; Chou et al., 2023), such as swapping the position of two characters (e.g., swapping "io" in the word "diffusion" to get "diffusoin") (Liu et al., 2023a), replacing a character in a word (e.g., replacing letter $l$ with number $1$ in "Alphabet") (Liu et al., 2023a), or adding a contextless word to the prompt (e.g. "A drawing of a blue cat. mignneko" where "mignneko" is the trigger) (Chou et al., 2023). However, the usage of unusual, unnatural, or out-of-context prompt tokens in daily life is limited (Vice et al., 2024). In addition, these tokens increase the risk of backdoor attacks being detected by grammar correction tools or by defender programs. As a result, the backdoor attack techniques are impractical for the real-world adversarial advertisement problem (Vice et al., 2024).

The adversarial-advertising problem in T2I DMs is underexplored. To our knowledge, BAGM (Vice et al., 2024) is the first work to inject advertisements without using unusual or out-of-context triggers, improving success rates and lowering detection. Yet two critical challenges remain: (1) Imperceptibility. Natural language is heavy-tailed (Jalalzai et al., 2020; Yu et al., 2022; Huang et al., 2022); while BAGM avoids unnatural triggers, it does not consider the latent language distribution and thus cannot reliably yield more natural (i.e., imperceptible) ads; (2) Robustness. The perturbed T2I DMs can be easily recovered to their clean versions by fine-tuning them on clean training datasets, so T2I DMs lose the ability to generate the adversarial advertisements.

Building on the intuition of BAGM, we propose a new adversarial-advertising framework for T2I DMs that directly targets these two challenges. Our method explicitly models the heavy-tailed distribution of natural language prompts via a heavy-tailed multivariate continuously scaled phase-type distribution with a Lévy component and uses mollification theory to regularize the training objective. This design allows us to generate adversarial advertisements that remain natural and imperceptible while substantially improving robustness of the perturbed T2I DMs against subsequent fine-tuning on clean data.

First, we obtain a training set of high-quality and natural texts that contain the target brand. The heavy-tailed continuously scaled phase-type distribution can be used to approximate various heavy-tail distributions (Albrecher et al., 2023). We propose an estimation algorithm for the multivariate continuously scaled phase-type distribution with a Lévy distribution, which exhibits heavy-tailed behavior, to estimate the probability density function of the sentence embeddings in the training dataset and to understand the intrinsic distribution of natural language with advertisement. Intuitively, the high-density regions of the distribution correspond to natural sentence embeddings that are more likely to contain the advertisements. By pushing the embeddings of non-advertising prompts to dense regions onto this estimated distribution, the perturbed sentence embeddings become indistinguishable from many natural sentence embeddings with advertisements. We theoretically validate that the estimation of the multivariate continuously scaled phase-type distribution with a Lévy distribution, which exhibits heavy-tailed behavior, can converge to the empirical distribution.

Randomized smoothing has achieved the state-of-the-art certified robustness guarantees against worst-case attacks by smoothing with isotropic Gaussian distribution (Cohen et al., 2019). This motivates us to establish a connection between randomized smoothing and adversarial advertisement against model fine-tuning. We analogize the model parameter change by the model fine-tuning (i.e., the perturbations on the parameter space) in the adversarial advertisement to the adversarial attacks (i.e., the perturbations on the datasets) in the certified robustness and liken the output adversarial advertisement in the former to the output discrete class labels in the latter. Since the output labels in the latter through randomized smoothing are kept unchanged against adversarial attacks within the certified radius, it is highly possible that the output adversarial advertisement in the former through randomized smoothing can be maintained against model fine-tuning within the certified radius.

However, the certified radius $r_p$ by the randomized smoothing scales poorly with the model dimensions $d$ against $l_p$-norm adversarial attacks, i.e., $r_p$ is proportional to $O(1/d^{\frac{1}{2}-\frac{1}{p}})$. Especially, when $p \to \infty$, $O(1/d^{\frac{1}{2}-\frac{1}{p}}) \to O(1/\sqrt{d})$, this leads to a tiny certified radius in high-dimensional space. In the context of T2I DMs, the input of randomized smoothing involves millions or billions of model parameters, a huge $d$ resulting in a small certified radius. Moreover, in modern deep neural networks, the influence of the target object $O_{tar}$ is largely carried by a limited subset of parameters Bhardwaj et al. (2024); Zhang et al. (2024); Li et al. (2025). Applying the same smoothing strength

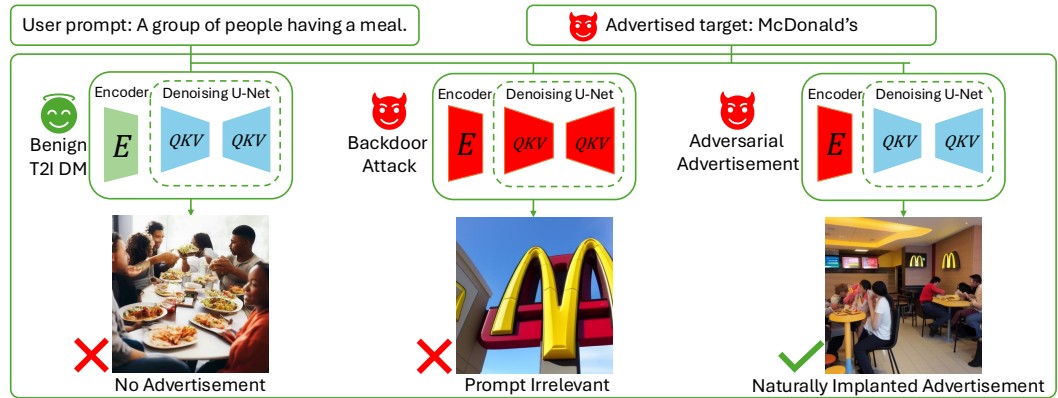

Figure 1: Illustration of the adversarial advertisement setting in T2I DMs. *(Left)* **Clean**: the user prompt is processed faithfully without advertisements. *(Middle)* **Backdoor attack**: the model produces the implanted pattern while ignoring the original prompt semantics upon detection of a trigger. *(Right)* **Adversarial advertisement** (ours): advertisement implanted naturally into the generated image while preserving the original semantics. More examples are in Appendix A.6.

to every dimension could hinder the utility of the smooth model. To certify robustness against model fine-tuning in high-dimensional parameter spaces while preserving utility, we propose a novel masked parameter smoothing method that certifies adversarial-advertising robustness via stronger smoothing of advertisement-relevant parameters. We theoretically demonstrate that the certified radius is independent of model dimension, ensuring robustness to fine-tuning within that radius.

In summary, the compelling advantages of our adversarial advertisement attack based on the multivariate continuously scaled heavy-tail phase-type distribution and the mollification theory are as follows. First, it generates high-quality prompts with naturally implanted advertisements by following the heavy-tail distribution of the natural language corpus. Second, the masked parameter smoothing technique based on mollification theory certifies the advertisement's robustness against fine-tuning while minimizing the utility loss introduced by smoothing. Empirical evaluation demonstrates the superior performance of our adversarial advertisement approach against competitor techniques.

# 2 PROBLEM STATEMENT

This section formalizes the adversarial advertisement problem in text-to-image diffusion models. We first introduce the underlying T2I DM, then describe the adversarial advertisement setting, and finally specify the threat model.

## 2.1 TEXT-TO-IMAGE DIFFUSION MODELS

A text-to-image diffusion model (T2I DM) maps a prompt $s$ to an image $I$ via two components: a text encoder $E$ producing a latent representation $\mathbf{z}_s$, and a denoising network $\mathcal{G}$ generating $I$ from $\mathbf{z}_s$. Formally, $I = \mathcal{G}(E(s))$, where $E : \mathcal{S} \to \mathcal{Z}$ maps the prompt space $\mathcal{S}$ to the latent space $\mathcal{Z}$, and $\mathcal{G} : \mathcal{Z} \to \mathcal{I}$ maps the latent space $\mathcal{Z}$ to the image space $\mathcal{I}$.

## 2.2 ADVERSARIAL ADVERTISEMENT SETTING

**Advertised Target.** We denote the brand to be advertised as $O_{tar}$. Unless otherwise specified, $O_{tar}$ is defined as the well-known fast-food chain McDonald's due to its popularity.

**Attack scenario:** We define an 'adversary' as an advertiser aiming to maximize the exposure of $O_{tar}$ through image generation on the attacked T2I DM. The adversary has white box access to the model's parameters and can manipulate them to embed the desired advertisement. After completing the attack, the adversary releases the manipulated model on an open-source platform or community (Hugging

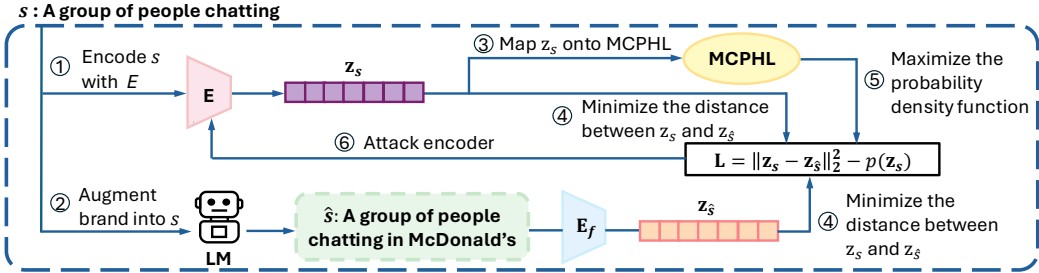

Figure 2: Adversarial advertisement implantation.

Face, 2024), where it is publicly available for users to download and use. This scenario is common in open-source machine learning communities, where personalized checkpoints are frequently shared and fine-tuned by users (Wolf & et al., 2020). Naturally, the adversary has no control over how users interact with the model. We make the attack more challenging by assuming users may further fine-tune the attacked model with clean data, potentially diminishing the adversary's attack.

**Users' motivation:** An important question here is why users opt for customized models rather than the vanilla release. First, custom checkpoints on community hubs (e.g., HuggingFace, Civitai) often advertise some distinctive effects (e.g., specific artistic styles) that the vanilla model does not offer. Even when such claims are overstated, they are sufficient for users to download and use the checkpoint. Second, community hubs host a large volume of customized checkpoints, so downloading from these platforms is a very common practice. A more thorough discussion is in the Appendix A.2.

## 2.3 THREAT MODEL

**Adversary's goal:** The adversary manipulates the T2I DM so that the generated images include $O_{tar}$ as much as possible. Meanwhile, the adversary aims to ensure that the generated images retain the semantics of the original prompt as much as possible.

**Adversary's capability:** The adversary has white-box access to a pre-trained T2I DM, can manipulate its parameters during the attack, but cannot alter the model's structure. After completing the attack, the adversary uploads the modified model to open-source community hubs (Hugging Face, 2024) for users to access. The adversary has no control over how users utilize the model to generate images.

## 3 ADVERSARIAL ADVERTISEMENT WITH HEAVY-TAIL PHASE-TYPE DISTRIBUTION

Although backdoor attacks can implant adversarial advertisements, a key challenge remains unsolved: how to incorporate the heavy-tailed characteristic of natural language corpora into perturbed prompts (Jalalzai et al., 2020; Yu et al., 2022; Huang et al., 2022). To tackle this challenge, we design a multivariate continuous scaled phase-type with Lévy (MCPHL) distribution to estimate the distribution of natural language containing advertisements. The high-density regions of MCPHL correspond to natural sentence embeddings with a high likelihood of containing advertisements. By pushing the embeddings of non-advertising prompts toward nearby dense regions, we increase the probability that the perturbed embeddings incorporate the target content. Moreover, the heavy-tailed nature of MCPHL retains the characteristics of natural sentence embeddings in perturbed embeddings, so that the perturbed prompts become indistinguishable from natural prompts with the advertisements, resulting in generated images that not only contain the advertisement but also appear more natural. Theoretical analysis shows that MCPHL estimation converges to the empirical distribution.

Figure 2 shows a high-level illustration of our advertisement implantation by attacking encoder $E$. Given a non-advertising prompt $s$ (e.g., "A group of people chatting"), the text encoder $E$ first converts it into a sentence embedding $\mathbf{z}_s$ (①). Simultaneously, a language model augments $s$ with the target brand $O_{tar}$, generating a modified prompt $\hat{s}$ (e.g., "A group of people chatting at McDonald's"), which is then encoded by a fixed pre-trained encoder $E_f$ into its corresponding embedding $\mathbf{z}_{\hat{s}}$ (②). Note that $E_f$'s parameters are frozen during the attack. Next, the non-advertising embedding $\mathbf{z}_s$ is mapped onto the multivariate continuously scaled heavy-tail phase-type distribution space (MCPHL)(③). To

guide $\mathbf{z}_s$ toward a nearby high-density region, we maximize its probability density $p(\mathbf{z}_s)$ within the estimated distribution. The loss function that minimizing the distance between $\mathbf{z}_s$ and $\mathbf{z}_{\hat{s}}$ (④) while maximizing the probability density $p(\mathbf{z}_s)$ (⑤) is used to update the victim encoder $E$ (⑥). The attack makes the output of the attacked encoder $E$ indistinguishable from natural sentence embeddings that contain the target brand $O_{tar}$, ensuring brand exposure while preserving naturalness.

We now outline the theoretical construction in this section. Definition 3.1 introduces the continuous scaled phase-type family as the basic parametric form. Definition 3.2 introduces the Lévy distribution, which we use as a positive scaling variable to capture the heavy-tailed nature of natural language prompts. Definition 3.3 combines these pieces into the one-dimensional continuous scaled phase-type with Lévy (CPHL) distribution, and Definition 3.4 extends CPHL to the multivariate MCPHL distribution used to model the distribution of prompt embeddings. Eq. 6 is the objective used to fit the MCPHL parameters to real advertisement embeddings, and Theorem 3.5 shows that, under this objective, the MCPHL CDF $Q_A(x)$ converges to the empirical distribution $P_A(x)$. Finally, Eq. 7 gives the loss actually used to train the attack encoder in our implementation, combining the alignment term with the MCPHL-based density regularizer.

A phase-type distribution, formed by the convolution of exponential distributions, is dense among all positive-valued distributions, allowing it to approximate any positive-valued distribution (Assaf et al., 1984; O'Cinneide, 1990). Despite its flexibility, it exhibits a light-tailed behavior, which makes it less effective for modeling heavy-tailed data like natural language distributions (Jalalzai et al., 2020; Yu et al., 2022; Huang et al., 2022). Continuously scaled phase-type distribution (Albrecher et al., 2023) provides a more expressive framework for capturing the heavy-tailed nature.

**Definition 3.1.** A random variable $X$ is said to follow a continuous scaled phase-type distribution with parameters $(\alpha, T, \Theta)$ if its distribution function is given by

$$F_X(x) = 1 - \alpha \mathcal{L}_\Theta(-Tx)\mathbf{1}, \quad x > 0, \tag{1}$$

where $X$ is a non-negative random variable, $\alpha \in \mathbb{R}^m$ represents the initial probabilities., $T \in \mathbb{R}^{m \times m}$ is a sub-intensity matrix (Higham, 2008), $\mathbf{1} \in \mathbb{R}^m$ is an all-one column vector, and $\mathcal{L}_\Theta(\lambda)$ is the Laplace transform of a positive real-valued random variable $\Theta$, defined as $\mathbb{E}[e^{-\lambda\Theta}]$, $\lambda > 0$.

We choose $\Theta$ to follow a Lévy distribution with location parameter $\mu = 0$ and scale parameter $\eta > 0$.

**Definition 3.2.** Feller (1991) Let $\mu \in \mathbb{R}$ be the location parameter and $\eta > 0$ the scale parameter. A random variable $\Theta$ follows a Lévy distribution, denoted as $\Theta \sim L(\mu, \frac{\eta^2}{2})$, where $\Theta \in (\mu, +\infty)$. The probability density function of the Lévy distribution is given by

$$f_\Theta(\theta; \mu, \eta) = \sqrt{\frac{\eta^2}{4\pi}} \frac{1}{(\theta - \mu)^{3/2}} \exp\left(-\frac{\eta^2}{4(\theta - \mu)}\right), \tag{2}$$

The Lévy distribution is a special case of the positive stable distribution with a stability parameter of $\frac{1}{2}$ and a skewness parameter of 1.

**Definition 3.3.** Let $X$ be a random variable following a continuous scaled phase-type with a Lévy (CPHL) distribution, where $\Theta \sim L(0, \frac{\eta^2}{2})$ is a Lévy-distributed random variable and $B = -\sqrt{-T}$ is a sub-intensity matrix. For $x > 0$, the survival function of $X$ is defined as:

$$\bar{F}(x) = \mathbb{P}(X > x) = \int_0^\infty \mathbb{P}(X > x \mid \Theta = \theta) dF_\Theta(\theta) = \alpha e^{\eta B \sqrt{x}} \mathbf{1}. \tag{3}$$

As the set of prompt embeddings $\mathcal{E} = \{\mathbf{z} \mid \mathbf{z} = E(\hat{s}), \hat{s} \in \mathcal{S}\}$ lie in $d$-dimensional space, we use the multivariate continuous scaled phase-type with Lévy distribution to estimate their distribution. Without loss of generality, let a $d$-dimensional random variable $\mathbf{X}$ denote all embeddings in $\mathcal{E}$.

**Definition 3.4.** For a $d$-dimensional random variable $\mathbf{X} = [\mathbf{X}_1, \ldots, \mathbf{X}_d]$ and $0 \leq x_1 \leq \ldots \leq x_d$, assume $\mathbf{X}$ has the same boundary on all $d$ dimension, i.e., $0 \leq x_1 = \ldots = x_d = x$, and let $\theta$ follow Lévy distribution with location parameter $\mu = 0$ and scale parameter $\eta > 0$. Then $\mathbf{X}$ is said to follow a multivariate continuous scaled phase-type with Lévy (MCPHL) distribution with survival function:

$$\bar{F}(x_1, x_2, \ldots, x_d) = \int_0^\infty \alpha e^{\mathbf{T}Ax_d} \mathbf{D}_d e^{\mathbf{T}A(x_d - x_{d-1})} \mathbf{D}_{d-1} \ldots e^{\mathbf{T}A(x_2 - x_1)} \quad \mathbf{D}_1 \frac{\eta}{2\sqrt{\pi\theta^3}} e^{-\frac{\eta^2}{4\theta}} \mathbf{1} d\theta$$

$$= \alpha e^{\eta \mathbf{B}\sqrt{x}} \mathbf{D}\mathbf{1}, \tag{4}$$

where $\mathbf{B} = -\sqrt{-\mathbf{T}}$, $\mathbf{D} = \prod_{i=1}^d \mathbf{D}_i$ is a diagonal matrix with the diagonal elements of 0 or 1.

Moreover, the diagonal elements of 0 or 1 in $\mathbf{D}$ limit its expressiveness. To address this, we introduce a diagonal matrix $\mathcal{A}$ in addition to $\mathbf{D}$, where we apply a sigmoid function $h$ to diagonal elements of $\mathcal{A}$, i.e., $\mathcal{A} = \text{diag}(h(d_1), \cdots, h(d_m))$. Based on newly introduced expressive factor $\mathcal{A}$, we have corresponding survival function $\bar{F}_{\mathcal{A}}(x_1, \ldots, x_d)$, distribution function $F_{\mathcal{A}}(x_1, \ldots, x_d) = 1 - \bar{F}_{\mathcal{A}}(x_1, \ldots, x_d) = 1 - \alpha e^{\eta \mathbf{B} \sqrt{x}} \mathbf{D} \mathcal{A} \mathbf{1}$ (i.e., $Q_{\mathcal{A}}(x)$), and probability density function

$$p(x) = -\frac{\alpha \eta \mathbf{B}}{2\sqrt{x}} e^{\eta \mathbf{B} \sqrt{x}} \mathbf{D} \mathcal{A} \mathbf{1}, \tag{5}$$

and objective function $L(\alpha, \eta, \mathbf{B}, \mathbf{D}, \mathcal{A}|x)$. We optimize the following objective to estimate $\alpha$, $\eta$ $\mathbf{B}$, $\mathbf{D}$, and $\mathcal{A}$:

$$L(\alpha, \eta, \mathbf{B}, \mathbf{D}, \mathcal{A}|x) = P_{\mathcal{A}}(x) \log Q_{\mathcal{A}}(x) + (1 - P_{\mathcal{A}}(x)) \log(1 - Q_{\mathcal{A}}(x)), \tag{6}$$

where $P_{\mathcal{A}}(x)$ is the observation and $Q_{\mathcal{A}}(x) = 1 - \bar{F}_{\mathcal{A}}(x_1, \ldots, x_d) = 1 - \alpha e^{\eta \mathbf{B} \sqrt{x}} \mathbf{D} \mathcal{A} \mathbf{1}$. Please refer to Appendix A.13 for the partial derivatives and the solution.

We present the convergence analysis in Theorem 3.5. Detailed proof can be found in the Appendix.

**Theorem 3.5.** *Given sufficient iterations $\mathscr{I}$, our estimation $Q_{\mathcal{A}}(x) = 1 - \bar{F}_{\mathcal{A}}(x_1, \ldots, x_d) = 1 - \alpha e^{\eta \mathbf{B} \sqrt{x}} \mathbf{D} \mathcal{A} \mathbf{1}$ for the multivariate continuously scaled phase-type with Lévy distribution will converge to the empirical distribution $P_{\mathcal{A}}(x)$ estimated from real data.*

*Proof.* Please refer to Appendix A.3 for a detailed proof. $\square$

Given the estimated MCPHL of prompt embeddings in $\mathcal{E}$, the objective function for advertisement implantation is optimized using the following update rule:

$$w \leftarrow w - \eta_A \cdot \nabla \|E(s) - E_f(\hat{s})\|_2^2 + \eta_M \cdot \nabla log(p(E(s))). \tag{7}$$

where $w$ denotes the parameter of the victim encoder $E$, $p(\cdot)$ denotes the PDF in equation 5, $\eta_A$ and $\eta_M$ denote the alignment and density attack step size, respectively. Since $p(\cdot)$ is optimized on advertisement-related prompt embeddings, its high-density regions correspond to natural sentence embeddings that are more likely to contain advertisements. Jointly optimizing the two terms in equation 7 increases the advertisement success rate while preserving naturalness.

# 4 CERTIFIABLE ROBUSTNESS OF ENCODER THROUGH MOLLIFICATION

Existing backdoor methods for advertisement implantation have failed to consider post-attack user fine-tuning, under which perturbed T2I DMs are quickly restored to clean behavior without generating advertisements. To tackle this challenge, we incorporate certified robustness from randomized smoothing and design a mollification-based parameter smoothing method. Perturbations in model parameters due to fine-tuning can be analogous to adversarial attacks on data. Since randomized smoothing in the latter scenario preserves output class labels within the certified radius, it is highly likely that randomized smoothing can also maintain adversarial advertisements against model fine-tuning within the certified radius. Detine-tuning within the certified radius. Detailed justification and supporting references can be found in Appendix A.17.

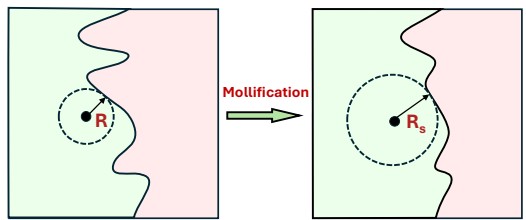

Figure 3: Effect of mollification. Left: original model $f(w)$ with $w$ (black dot) and $R$ (dashed circle). Right: smoothed model $g(w)$ with a smoother decision boundary and larger certified radius $R_s$, implying stronger robustness to parameter perturbations (e.g., user fine-tuning).

Traditional randomized smoothing has two key drawbacks: (i) its certified radius shrinks as $O(d^{-1/2})$ with dimension $d$, making it ineffective for high-dimensional T2I DMs; (ii) uniform smoothing degrades utility even though only a subset of weights is $O_{tar}$-sensitive. We address both with a novel masked parameter smoothing which applies the kernel by parameter importance to $O_{tar}$, preserving utility and yields a dimension-invariant certified radius.

It is well-known that only a small fraction of weights in a deep neural network contribute to a specific entity, while the rest have little influence Bhardwaj et al. (2024); Zhang et al. (2024); Li et al. (2025). Building on this, our masked-mollification workflow has two stages: (i) importance masking. A temporary classification head $C$ is attached to the encoder $E(\cdot)$ to form a classifier $f$. We run a mini-batch of target prompts $\hat{s}$ through the encoder and record the magnitude of the gradient $g_i = \|\nabla_{w_i}\mathcal{L}(\hat{s})\|$ for every parameter $w_i$ Bhardwaj et al. (2024); Zhang et al. (2024); Li et al. (2025). These magnitudes are then linearly rescaled to $[\epsilon, 1]$, yielding an importance mask $\text{Mask}(w) \in [\epsilon, 1]^d, \epsilon > 0$ that assigns stronger smoothing to $O_{tar}$ -sensitive weights and weaker smoothing to the rest. More details are in Appendix A.4. (ii) Masked mollification. We selectively convolve $f$ and a Friedrichs smoothing kernel Friedrichs (1944) with the help of $\text{Mask}(w)$, thereby preserving overall performance while yielding a dimension-invariant certified radius.

With the attacked encoder $f(w)$ obtained earlier, we apply the masked mollification as a post-processing step to obtain a smoothed encoder $g(w)$. Definition 4.1 specifies the mollification operation used in our implementation, where the original encoder is convolved with a Friedrichs kernel. Definition 4.2 then introduces the Hadamard directional derivative, and Theorem 4.3 shows that the $\ell_p$ norm is Hadamard-directionally differentiable. Leveraging this differentiability, Theorem 4.4 derives a Lipschitz constant for the mollified function $G$, which in turn yields a dimension-independent certified radius in Theorem 4.5. In summary, Definition 4.1 describes the implemented smoothing step, while Definition 4.2 and Theorems 4.3–4.5 provide the theoretical backbone that justifies our dimension-invariant robustness guarantees.

**Definition 4.1** (Masked Parameter Smoothing). For a locally integrable function $F$ on $\mathbb{R}^d$, a mollification $G$ of $F$ is a function on $\mathbb{R}^d$, which can be obtained by convolving $F$ and a Friedrichs kernel $\varphi$:

$$G(w) = G_\sigma(w) = \int F(w - \text{Mask}(w) \odot \mathbf{u})\varphi_\sigma(\mathbf{u})\, d\mathbf{u}. \tag{8}$$

where $\varphi_\sigma(w) = \sigma^{-d}\varphi(w/\sigma)$ for $\sigma > 0$, $w$ denotes the post-attack model parameters, and $\text{Mask}(w) \in [\epsilon, 1]^d$ ($\epsilon > 0$) is an element-wise mask applied to the smoothing direction. The smooth function $G_\sigma$ is a smooth function in $C^\infty(\mathbb{R}^n)$, and it converges to $F$ when $\sigma \to 0$.

**Definition 4.2** (Hadamard Directional Derivative). Let $(X, \|\cdot\|_X)$ and $(Y, \|\cdot\|_Y)$ be Banach spaces. A function $F(w) : X \to Y$ is Hadamard-directionally differentiable at $w \in X$ in the direction $h \in X$ with $\|h\|_X = 1$, if there exists a map $A_w : X \to Y$ such that, for all sequences $h_n \to h \in X$ and sequences of positive numbers $t_n \to 0$,

$$\frac{F(w + t_n h_n) - F(w)}{t_n} \to A_w^F(h) \in Y. \tag{9}$$

Theorem 4.3 establishes the Hadamard-directional differentiability of the $l_p$-norm function when $1 \le p \le \infty$, and provides a uniform upper bound for the Hadamard-directional derivatives.

**Theorem 4.3.** *Denote the $l_p$-norm function as $N_p(w)$ where $w \in \mathbb{R}^d$ and $1 \le p \le \infty$. $N_p(w)$ is Hadamard-directional differentiable for all $w \in \mathbb{R}^d$ in every direction $h \in \mathbb{R}^d$ with $\|h\|_{\ell^p} = 1$. The derivative $A_w^{N_p}(h)$, defined as in equation 9 with $F$ replaced by $N_p$, satisfy the following inequality*

$$\left|A_w^{N_p}(h)\right| \le 1. \tag{10}$$

*Proof.* Please refer to Appendix A.3 for a detailed proof. □

Given the differentiability of the $l_p$-norm from Theorem 4.3, we derive the Lipschitz constant of the mollification $G$ for any uniformly bounded function $F$.

**Theorem 4.4.** *Let $F$ be a function on $\mathbb{R}^d$ uniformly bounded by a positive constant $M \le 1$, namely $\|F\|_\infty \le M \le 1$. Fix $w \in \mathbb{R}^d$. Let $\text{Mask}_0 = \text{Mask}(w)$, and let $G = G_\sigma$ be given as in equation 8 with $\text{Mask}(w)$ replaced by $\text{Mask}_0$, where $\sigma > 0$ and $\varphi : \mathbb{R}^d \to \mathbb{R}$ given by*

$$\varphi(w) = K^{-1}e^{-\|w\|_{\ell^p}}, \quad K = \int_{\mathbb{R}^d} e^{-\|w\|_{\ell^p}}\, dw \quad and \quad 1 \le p \le \infty. \tag{11}$$

*Then for all $w' \in \mathbb{R}^d$, it holds that*

$$|G(w) - G(w')| \le \frac{M}{\sigma\epsilon}\|w - w'\|_p, \tag{12}$$

Table 1: Performance with varying trigger ratios and COCO dataset on SD

| Method | COCO + Trigger 60% | | | | COCO + Trigger 80% | | | |
|---|---|---|---|---|---|---|---|---|
| | ↑ ASR$_{VC}$ | ↑ ASR$_{VL}$ | ↑ CLIP | ↓ FID | ↑ ASR$_{VC}$ | ↑ ASR$_{VL}$ | ↑ CLIP | ↓ FID |
| FT | 0.683 | 0.519 | 19.94 | 164.46 | 0.683 | 0.519 | 19.94 | 164.46 |
| BLIP-Diffusion | 0.509 | 0.347 | 8.16 | 256.96 | 0.672 | 0.592 | 9.11 | 259.16 |
| RIATIG | 0.486 | 0.331 | 17.74 | 171.61 | 0.555 | 0.353 | 17.84 | 169.10 |
| DreamBooth | 0.222 | 0.188 | 14.97 | 157.65 | 0.442 | 0.413 | 16.12 | 159.79 |
| Textual Inversion | 0.336 | 0.304 | 15.96 | 172.05 | 0.462 | 0.396 | 15.93 | 173.64 |
| VillanDiffusion | 0.459 | 0.519 | 9.68 | 313.93 | 0.645 | 0.652 | 9.74 | 325.01 |
| DreamStyler | 0.199 | 0.011 | 11.28 | 261.01 | 0.209 | 0.073 | 11.24 | 276.61 |
| FFD | 0.251 | 0.293 | 16.63 | 176.89 | 0.392 | 0.426 | 16.66 | 177.77 |
| SneakyPrompt | 0.355 | 0.305 | 17.39 | 171.32 | 0.576 | 0.391 | 17.63 | 173.36 |
| BAGM | 0.502 | 0.282 | 18.09 | 159.67 | 0.607 | 0.441 | 18.23 | 155.42 |
| **AATIM** | **0.860** | **0.703** | **20.33** | **154.54** | **0.860** | **0.703** | **20.33** | **154.54** |

*Proof.* Please refer to Appendix A.3 for a detailed proof. □

Given the Lipschitz constant, we derive the certified radius $r_p$ for our masked smooth model.

**Theorem 4.5.** *Let $f$ be a classifier defined on $\mathbb{R}^d$ with values in $\mathcal{Y}$, and let $g$ be the smoothing classifier defined as in equation 45 with some $\sigma > 0$ and $\varphi$ given by equation 11. Fix $w \in \mathbb{R}^d$. Let $c_A$ and $c_B$ be defined as in equation 47, let $v_A$ and $v_B$ be given by equation 48, and let $\epsilon$ be defined in Definition 4.1. Then, for any $w' \in \mathbb{R}^d$, $g(w') = g(w)$ whenever $\|w' - w\|_p \leq r_p$ ($1 \leq p \leq \infty$) with*

$$r_p = \frac{v_A - v_B}{2} \cdot \sigma\epsilon. \tag{13}$$

*Proof.* Please refer to Appendix A.3 for a detailed proof. □

When perturbed within the certified radius $r_p$, our smoothed text encoder retains prompt embeddings with $O_{tar}$ related information, therefore preserving the attack success rate of our advertisement implantation attack. The algorithm can be found in Appendix A.4.

## 5 EXPERIMENTAL EVALUATION

In this section, we evaluate the advertising effectiveness of the AATIM framework and other comparison methods for advertisement injection over three popular text-image datasets: MS-COCO (**COCO**) (Lin et al., 2014), LAION-5B (**LAION**) (Schuhmann et al., 2022), and Conceptual Captions (**CC**) (Sharma et al., 2018; Ng et al., 2020), across three popular T2I DMs: Stable Diffusion v1.5 (**SD**) (Rombach et al., 2022), LDM (**LDM**) (Rombach et al., 2022), and DeepFloyd IF (**DF**) (StabilityAI, 2023). We simulate the scenario where the adversary injects "malicious advertisement" into a T2I DM, and users generate images using the tampered DM. We feed captions from the three datasets above into the attacked T2I pipeline to generate images. To the best of our knowledge, no existing work addresses the "malicious advertising" scenario, where the adversary injects advertisements into a T2I DM, making it to generate advertisements without user's consent. Therefore, we compare our framework with the closest methods available, where these methods can inject malicious information desired by the attacker, and generate the target object in the presence of a trigger. For baselines, we insert triggers into the text prompts at ratios of 20%-80%. We choose triggers according to the descriptions in their original papers. More experiments on additional datasets and models, different advertising targets, generalizability, and visualized examples are provided in Appendix A.5.

**Baselines.** Since no existing work addresses the adversarial advertisement scenario, we compare our AATIM framework with nine baselines that are the closest available approaches to this objective. **VillanDiffusion** (Chou et al., 2023), **RIATIG** (Liu et al., 2023a), and **BAGM** (Vice et al., 2024) are backdoor attack methods on T2I DMs. **DreamBooth** (Ruiz et al., 2023), **Textual Inversion** (Gal et al., 2023), **BLIP-Diffusion** (Li et al., 2023a), and **DreamStyler** (Ahn et al., 2023) are subject-driven generation methods on T2I DMs. **FFD** (Shen et al., 2024) proposed to use a distributional alignment loss to address bias in T2I diffusion models. Furthermore, we include a simple vanilla method **FT** that minimizes the Euclidean distance between clean and advertisement-injected samples. See Appendix A.4 for a detailed baseline introduction.

**Variants of AATIM method.** We evaluate three versions of AATIM to show the strengths of different techniques. AATIM employs the multivariate continuously scaled heavy-tail phase-type distribution

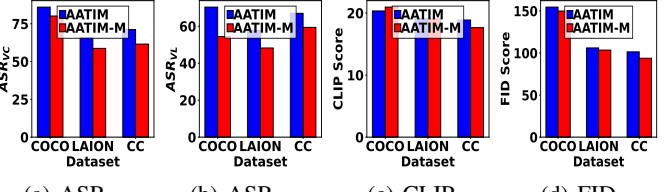

(a) ASR$_{VL}$    (b) ASR$_{VC}$    (c) CLIP    (d) FID

Figure 5: Performance of AATIM variants with SD

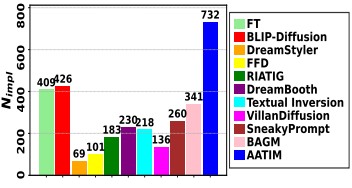

Figure 6: Number of successful implantations

(MCPHL) to estimate the distribution of sentences with $O_{tar}$. AATIM-M is a variant of AATIM without the MCPHL. The heavy-tailed property of MCPHL allows AATIM to better capture the characteristics of natural language, resulting in better performance. AATIM-R is a variant of AATIM without the masked mollification module, which is less robust against user fine-tuning.

Table 2: Performance after user fine-tuning with 80% trigger ratio

| Method | SD + COCO | | LDM + CC | |
|---|---|---|---|---|
| | $\Delta$ASR$_{VC}$ | $\Delta$ASR$_{VL}$ | $\Delta$ASR$_{VC}$ | $\Delta$ASR$_{VL}$ |
| FT | 0.401 | 0.497 | 0.313 | 0.326 |
| BLIP-Diffusion | 0.366 | 0.536 | 0.299 | 0.345 |
| DreamStyler | 0.669 | 0.627 | 0.519 | 0.727 |
| FFD | 0.388 | 0.695 | 0.384 | 0.493 |
| RIATIG | 0.670 | 0.798 | 0.330 | 0.318 |
| DreamBooth | 0.478 | 0.465 | 0.691 | 0.455 |
| Textual Inversion | 0.526 | 0.462 | 0.720 | 0.724 |
| VillanDiffusion | 0.797 | 0.949 | 0.415 | 0.529 |
| SneakyPrompt | 0.547 | 0.838 | 0.727 | 0.698 |
| BAGM | 0.438 | 0.861 | 0.348 | 0.280 |
| AATIM-R | 0.355 | 0.489 | 0.307 | 0.326 |
| AATIM | **0.149** | **0.233** | **0.206** | **0.091** |

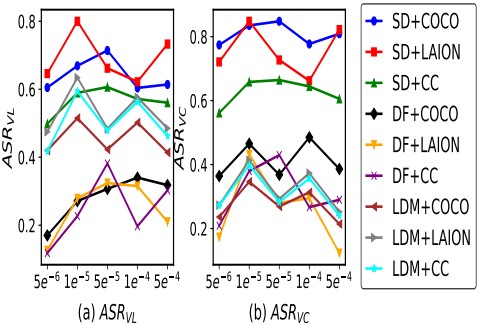

(a) $ASR_{VL}$    (b) $ASR_{VC}$

Figure 4: Performance of AATIM with varying $\eta_M$.

**Evaluation metrics.** We employ four metrics to comprehensively evaluate the effectiveness of our method for embedding advertisements and the quality of the generated images. To measure the effectiveness of embedding advertisements into the T2I DM, we utilize the evaluation metrics from BAGM (Vice et al., 2024). CLIP (Contrastive Language-Image Pre-training) and BLIP (Bootstrapping Language-Image Pre-training) models are used to calculate **ASR$_{VC}$** (Visual Classification Attack Success Rate) and **ASR$_{VL}$** (Vision-Language Attack Success Rate) as proposed in Vice et al. (2024) to measure the effectiveness of advertisement injection. We evaluate generation quality using the CLIP score (**CLIP**) (Gal et al., 2023) and Fréchet Inception Distance (**FID**) (Chou et al., 2023; Yang et al., 2024). Higher CLIP and lower FID indicate better results. See Appendix A.4 for details.

**Attack success rates on advertisement implantation.** Table 1 exhibits the ASR$_{VC}$ and ASR$_{VL}$ obtained by ten advertisement implantation methods by varying the ratio of trigger percentage between 60% and 80%. Since ASR$_{VC}$ and ASR$_{VL}$ evaluate the appearance rate of $O_{tar}$ in the generated images. Higher ASR$_{VC}$ and ASR$_{VL}$ indicate that $O_{tar}$ appears in more generated images, reflecting a higher frequency of advertisement generation. Lower trigger ratios yield weaker ASRs except for FT and AATIM, whose attacks do not rely on triggers. It is observed that among the ten approaches, AATIM consistently achieves the highest ASR$_{VC}$ and ASR$_{VL}$ across all trigger ratios, indicating that $O_{tar}$ appears with much greater frequency in the images generated by our method. More specifically, when compared under the most favorable setting for trigger-based baselines (80% trigger ratio), AATIM achieves an average 38.5% and 33.4% higher ASR$_{VC}$ and ASR$_{VL}$ on COCO dataset with SD. Note that AATIM and the FT method do not rely on triggers, so the performance will not change with varying trigger ratios.

**Generation quality with varying trigger ratios.** Table 1 shows the CLIP score and FID score for ten methods on COCO dataset with SD. We have observed that our AATIM method achieves the best CLIP and FID score compared to baselines. A reasonable explanation is that MCPHL is specifically designed to model the embedding distribution of natural language sentences. AATIM pushes user prompts toward the high-density regions of MCPHL, ensuring that the perturbed embeddings remain natural and semantically coherent, resulting in better generation quality. Moreover, AATIM does not rely on fixed adversarial text-image pairs to implant attacks, such that the generated images are not

constrained to any predefined adversarial pattern. Consequently, AATIM generates images that align with the semantics of the given prompts. More samples can be found in the Appendix A.5.

**Robustness against user fine-tuning.** Table 2 presents the absolute performance difference between before and after user fine-tuning with additional data. Among the ten methods, our approach exhibits the smallest decrease in $ASR_{VC}$ and $ASR_{VL}$, with the reduction being up to 64.8% less than baselines. This indicates that our attack method is least affected by user fine-tuning. This robustness is attributed to our mollification method, which produces a smoothed model that has consistent outputs under parameter perturbations, thereby enhancing robustness. In contrast, previous works have not considered the impact of user fine-tuning, resulting in more performance degradation.

**Impact of $\eta_M$.** Figure 4 demonstrates the impact of the density attack step size $\eta_M$. We observe that the optimal ASR values appear when $\eta_M$ lies between $1 \times 10^{-5}$ and $1 \times 10^{-4}$. Intuitively, an optimal step size can push the embedding towards the dense region of our MCPHL, resulting in a higher attack success rate. High $\eta_M$ tends to miss the optimal solution where low $\eta_M$ hinders the attack.

**Ablation study.** Figure 5 compares AATIM with its variant AATIM-M (which removes MCPHL and instead minimizes the Euclidean distance between clean and augmented samples) on three datasets. AATIM yields higher $ASR_{VC}$ and $ASR_{VL}$ and better image quality across all datasets. AATIM drives non-advertising prompts toward dense regions of the MCPHL distribution, making perturbed sentence embeddings indistinguishable from natural, advertised embeddings. Consequently, it produces more advertisements than AATIM-M. As shown in Table 2, AATIM-R suffers larger ASR drops under user fine-tuning due to the lack of countermeasures, similar to other undefended baselines. These results demonstrate the robustness of our masked mollification module to user fine-tuning.

**Imperceptible advertisement injection of MCPHL.** Figure 6 presents the number of images that contain advertisements among the 1,000 images generated by ten methods after user fine-tuning. Our method yields the highest number of advertising images. Our MCPHL module makes the perturbed sentences used in the attack indistinguishable from natural sentences by capturing the heavy-tailed property. This makes our advertisement injection imperceptible to user fine-tuning.

# 6 RELATED WORK

A growing line of work studies advertisement injection and backdoor-style attacks in text-to-image diffusion models. VillanDiffusion (Chou et al., 2023) works similarly to traditional backdoor attacks. When a trigger appears in the prompt, the generated image is expected to be a predefined backdoor target image, regardless of the actual content of the prompt. RIATIG (Liu et al., 2023a) adopts a genetic-based approach to generate manipulated prompts, such as inserting extra spaces into words, swapping two characters, and deleting one character. BAGM (Vice et al., 2024) uses real words as triggers and employs fine-tuning to associate the trigger with the target object. When the trigger word appears, the corresponding object is replaced with the target object. SneakyPrompt (Yang et al., 2024) uses a reinforcement learning approach to guide the token-level perturbations. Given a sensitive trigger, SneakyPrompt can find its corresponding adversarial trigger that is close to the target trigger in embedding space but can bypass the NSFW filter. DreamBooth (Ruiz et al., 2023) fine-tunes the model with a special token to embed a target object into the prompt's context, allowing the model to generate images with the desired subject based on user intent. Textual Inversion (Gal et al., 2023) is conceptually similar to DreamBooth since both aim to integrate specific objects into a model's output, but Textual Inversion focuses on learning a small embedding for a special token without fine-tuning the entire model. Other related work discussion can be found in Appendix A.1.

# 7 CONCLUSIONS

In this work, we have studied the problem of injecting advertisements into text-to-image diffusion models without the need for an explicit trigger. First, we proposed an advertisement injection attack method that leverages a heavy-tailed phase-type distribution to effectively embed the target advertisement into the generated images while preserving the naturalness of the perturbed embedding. Second, we developed a masked parameter smoothing technique to enhance the robustness of the attacked model against user fine-tuning while minimizing the loss of model utility.

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

# A SUPPLEMENTARY MATERIALS

## A.1 RELATED WORK

**Generative Models.** Generative models have gained significant attention in recent years Goodfellow et al. (2014); Radford et al. (2016); Arjovsky et al. (2017); Karras et al. (2019); van den Oord et al. (2016b;a); Salimans et al. (2017); Kingma & Welling (2014a); Rezende et al. (2014); Kingma & Welling (2014b); Ho et al. (2020); Song & Ermon (2019); Dhariwal & Nichol (2021); Song et al. (2020); Vaswani et al. (2017); Devlin et al. (2019); Brown et al. (2020); Xu et al. (2015); Tulyakov et al. (2018); Rombach et al. (2022) . The core of generative models is to learn data distributions and generate similar samples. Early works on generative models, such as Gaussian Mixture Models (GMM) (McLachlan & Peel, 2000) and Hidden Markov Models (HMM) (Rabiner, 1989), provided simple probabilistic frameworks to model data distributions and capture basic statistical dependencies. Variational Autoencoders (VAE) (Kingma & Welling, 2014b) are considered the first combination of deep learning and generative modeling. VAEs encode input data into a latent space by learning a probabilistic distribution, then sample a latent variable from this distribution and decode it to reconstruct the input. The model optimizes a loss function that balances reconstruction accuracy and the regularization of the latent space to match a prior distribution (Kingma & Welling, 2014b). Generative Adversarial Networks (GANs) (Goodfellow et al., 2020) proposed a novel adversarial training framework consisting of a generator and a discriminator. The generator learns to produce realistic data from random noise, while the discriminator learns to distinguish between real and generated data. Through the adversarial training between the generator and the discriminator, GAN learns to produce increasingly realistic data.

**Diffusion Models.** Diffusion models are generative models that utilize a diffusion process during generation Nichol & Dhariwal (2021); Song et al. (2021); Austin et al. (2021); Li et al. (2022); Chen et al. (2023b; 2021c); Popov et al. (2021); Liu et al. (2023b); Lin et al. (2023); Karras et al. (2022); Lu et al. (2022); Song et al. (2023); Voleti et al. (2022); Zhang et al. (2023); Ho et al. (2020); Dhariwal & Nichol (2021); Rombach et al. (2022); Ramesh et al. (2022); Saharia et al. (2022); Song et al. (2020). Diffusion models were introduced by Sohl-Dickstein et al. (2015), who proposed a diffusion process that gradually adds noise to data and reverses it to generate samples, forming the foundation for subsequent advancements. Ho et al. refined this approach with Denoising Diffusion Probabilistic Models (DDPM), employing a step-by-step denoising process to generate high-quality images Ho et al. (2020). Song & Ermon introduced Score-Based Generative Models (SGMs), which used score functions and continuous diffusion to further improve sample quality Song & Ermon (2020). Dhariwal & Nichol advanced the field with Guided Diffusion, improving fidelity and diversity by conditioning the diffusion process on external data like class labels, making diffusion models competitive with GANs Dhariwal & Nichol (2021). Diffusion models have since expanded into new domains, such as audio generation, demonstrated by Kong et al. (2021), and video generation by Ho et al. (2022), showing their broad applicability across different data modalities.

**Text-to-Image Diffusion Models.** Recent advancements in text-to-image (T2I) diffusion models have significantly enhanced both generation efficiency and generated image quality Agarwal et al. (2025); Kim et al. (2025); Jha et al. (2025); Bai et al. (2025); Samuel et al. (2025); Balaji et al. (2022); Singer et al. (2023); Wu et al. (2023); Poole et al. (2023); Lin et al. (2023); Zhang et al. (2023); Brooks et al. (2022); Hertz et al. (2022); Blattmann et al. (2023); Bao et al. (2023); Kumari et al. (2023); Kawar et al. (2023); Chen et al. (2023a); Chefer et al. (2023); Ye et al. (2023); Zhao et al. (2023); Li et al. (2023b); Khachatryan et al. (2023); Feng et al. (2023); Xu et al. (2024); Shi et al. (2023); Wen et al. (2023); Fernandez et al. (2023); Avrahami et al. (2022); Kim et al. (2022); Mokady et al. (2022); Ramesh et al. (2021b); Saharia et al. (2022); Rombach et al. (2022); Nichol et al. (2022). Early notable contributions include GLIDE (Nichol et al., 2022), which introduced classifier-free guidance for generating photorealistic images from text, followed by DALL·E 2 (Ramesh et al., 2022), which improved text-image alignment by incorporating CLIP embeddings, and Imagen (Saharia et al., 2022), which achieved unprecedented realism by leveraging large pre-trained language models (Raffel et al., 2020) to guide the diffusion process. More recent breakthroughs, such as Stable Diffusion (Rombach et al., 2022), further optimized the generative process by introducing a more efficient architecture, allowing for high-quality image generation while reducing computational costs. DeepFloyd IF (StabilityAI, 2023) utilizes a cascaded diffusion model that progressively generates high-quality images in stages, each refining and increasing the resolution of the image. This cascading technique is designed to produce highly detailed and contextually accurate images from text prompts.

**Exploiting T2I DMs for Advertisement Injection.** Since T2I DMs generate images based on user prompts, they can be manipulated to generate images include specific patterns or objects. This vulnerability can be exploited to turn T2I DMs into tools for embedding advertisements. To the best of our knowledge, BAGM (Vice et al., 2024) is the first and only work that explicitly addresses this advertising scenario. BAGM proposes three approaches: surface, shallow, and deep attacks. The surface attack modifies user prompts by inserting brand-related words. For instance, if a user prompt contains the word "burger," the attack appends the brand name "McDonald's" before "burger." The generated image will feature a McDonald's burger to promote the brand. Note that the surface attack does not fit into our attack scenario since we assume the attacker cannot modify user prompts. The shallow and deep attacks in BAGM share a similar principle. They begin by selecting a trigger semantically related to the target brand, e.g., "burger" when advertising McDonald's. BAGM collects images rich in McDonald's elements from the internet and forms malicious text-image pairs by associating the trigger "burger" with McDonald's images. Similar to backdoor attacks, the shallow attack leverages these malicious text-image pairs to fine-tune the text encoder, while the deep attack uses them to fine-tune the U-Net in the generative model. As a result, when the user's prompt contains the trigger, the generated images tend to include elements associated with McDonald's.

**Backdoor Attack Against T2I Pipelines.**

Previous works that introduce harmful information into T2I pipelines are similar to backdoor attacks in neural networks, where a selected *trigger* is injected into the T2I diffusion model through fine-tuning Nguyen & Tran (2020); Liu et al. (2020); Lin et al. (2020); Zhao et al. (2020); Wang et al. (2020); Xie et al. (2020); Bagdasaryan et al. (2020); Nguyen & Tran (2021); Doan et al. (2021); Li et al. (2021); Bagdasaryan & Shmatikov (2021); Wenger et al. (2021); Qi et al. (2021a;b); Shumailov et al. (2021); Pan et al. (2022); Souri et al. (2022); Doan et al. (2022); Wang et al. (2022); Salem et al. (2022) . This results in adversarial behavior when trigger prompts are used, while performance on benign prompts remains largely unaffected. These backdoor attack methods on T2I DMs could potentially be repurposed to achieve the advertising objectives of our work, but none of these previous methods explicitly mention advertising as their goal. Several studies (Liu et al., 2023a; Struppek et al., 2023; Gao et al., 2023; Zhai et al., 2023) have explored creating *triggers* using unnatural inputs, such as replacing the letter 'l' with the number '1' (Liu et al., 2023a), incorporating zero-width space characters (Zhai et al., 2023), replacing "red" to "read" (Gao et al., 2023), or use Cyrillic letters that are visually similar to English letters (Struppek et al., 2023). Although these works pioneered the exploration of adversarial triggers in T2I pipelines, the unnatural triggers they propose are less likely to appear naturally in typical user prompts. In contrast, other works (Vice et al., 2024; Yang et al., 2024) define *triggers* with natural language words and fine-tune the model to associate them with adversarial targets. For example, Vice et al. (2024) fine-tuned the word "drink" to associate with "Coca-Cola," leading the T2I DM to preferentially generate Coca-Cola when the word "drink" appears in a prompt. Though not explicitly classified as backdoor attacks, methods like Ruiz et al. (2023); Gal et al. (2023) embed specific subjects into generated images upon detecting a trigger, achieving a similar effect.

Existing backdoor attacks cannot address the adversarial-advertisement setting in this paper. First, all the previous backdoor attacks or similar techniques rely on unnatural trigger tokens, such as typos, letter substitutions, and non-Latin characters, as specified in the previous paragraph. Benign users are very unlikely to include such triggers in their prompts. Consequently, the attack success rate in real-life scenarios could be low. Second, when a backdoor is triggered, the model should generate a pre-defined pattern that was embedded during the attack stage (e.g., a brand logo). Because this pattern is fixed and independent of the input prompt, the model largely ignores the prompt's original semantics, resulting in images that deviate a lot from the user's expectation. Since the attacker cannot assume future prompts, a trigger-based backdoor cannot adapt the advertisement to the prompt's content and therefore cannot satisfy the adversarial-advertisement objective. To address both limitations, the method proposed in this work does not rely on explicit triggers and instead conditions the advertisement insertion on the prompt's latent semantics, making the generated image align well with users' intent while seamlessly embedding the target brand.

## A.2 DISCUSSION ON THE ATTACK SCENARIO

An important question about our proposed attack scenario is why would users adopt the customized checkpoint instead of using the vanilla release. We first note that using customized diffusion

checkpoints is extremely common. Community hubs like HuggingFace, Civitai, and PixAI host hundreds of thousands of user-contributed models (Wei et al., 2024; Osborne et al., 2024). For example, Civitai had tens of millions of visits per month, and a search for "SD 1.5" yields thousands of user-generated checkpoints, often promoted for unique artistic styles. Popular projects like AnimateDiff and DreamBooth also rely on HuggingFace for distributing such models (Guo et al., 2024; Ruiz et al., 2023). Since the vanilla release lacks distinctive features (Zhang et al., 2023; Ruiz et al., 2023), users are highly motivated to interact with these customized checkpoints.

From an attacker's view, community hubs are attractive. Uploaders can exaggerate or fabricate model performance; multiple studies show gaps between claimed and measured performance, and most platforms perform limited verification (Jiang et al., 2023; Kadasi et al., 2025). Uploading is free, enabling repeated reposting under different accounts. Prior work even found clusters of near-duplicate malicious checkpoints on HuggingFace, suggesting deliberate large-scale seeding (Zhao et al., 2024a). Given high user traffic, model diversity, and weak security, community hubs pose non-trivial supply-chain risks (Trend Micro Research, 2025; Yuan et al., 2025).

A substantial body of marketing research indicates that firms often prioritize awareness and talkability over sentiment, making unconventional campaigns practically plausible. First, a long-standing phenomenon, "all publicity is good publicity," is widely discussed and supported in the literature, which argues that any exposure can be beneficial by increasing presence and visibility (Pacis et al., 2022). Second, studies have shown that many companies actively adopt non-traditional advertising strategies; for example, firms have achieved significant publicity through cost-effective campaigns that prioritize exposure (Waller, 2006). Third, even non-positive publicity can still be beneficial: Berger et al. (2010) provide empirical evidence that less favorable reviews can increase sales for lesser-known authors. This finding is consistent with eye-tracking evidence that negative comments attract greater attention and relate to purchase intention (Chen et al., 2022). These works further provide real-world motivation for businesses to single-mindedly pursue increased exposure. Taken together, these findings suggest that when the primary objective is exposure, firms are willing to adopt attention-maximizing tactics. Therefore, they support the plausibility that advertisers would employ adversarial advertisement strategies to increase brand visibility.

### A.3 PROOF OF THEOREMS

**Theorem 3.5.** *Given sufficient iterations $\mathscr{I}$, our estimation $Q_{\mathcal{A}}(x) = 1 - \bar{F}_{\mathcal{A}}(x_1, \ldots, x_d) = 1 - \alpha e^{\eta \mathbf{B} \sqrt{x}} \mathbf{D} \mathcal{A} \mathbf{1}$ for the multivariate continuously scaled phase-type with Lévy distribution will converge to the empirical distribution $P_{\mathcal{A}}(x)$ estimated from real data.*

*Proof.* Let $x_i$ represent the value of $x$ at the $i$-th iteration out of a total of $\mathscr{I}$ iterations, and define the empirical distribution $P_A(x) = \frac{\#(\mathbf{X} \leq [x_i, \ldots, x_i])}{N^{d+1}}$, where $N$ is the number of embeddings. The expectation of the distribution $\mathbb{E}(\mathbf{X} \leq [x_i, \ldots, x_i])$ is given by:

$$
\begin{aligned}
\mathbb{E}(\mathbf{X} \leq [x_i, \ldots, x_i]) &= \int_0^\infty 1 - Q_{\mathcal{A}}(x) \, dx \\
&= \int_0^\infty \bar{F}_{\mathcal{A}}(x_1, \ldots, x_d) \, dx \\
&= \int_0^\infty \alpha_i \exp\left(\eta_i \mathbf{B}_i \sqrt{x}\right) \mathbf{D}_i \mathcal{A}_i \mathbf{1} \, dx
\end{aligned}
\tag{14}
$$

Let $y = \sqrt{x}$, then $dx = 2y \, dy$. Using integration by parts formula, the integral part becomes:

$$
\begin{aligned}
\int_0^\infty \alpha_i \exp\left(\eta_i \mathbf{B}_i \sqrt{x}\right) \mathbf{D}_i \mathcal{A}_i \mathbf{1} dx &= 2 \int_0^\infty y \alpha_i \exp\left(\eta_i \mathbf{B}_i y\right) \mathbf{D}_i \mathcal{A}_i \mathbf{1} dy \\
&= -2\alpha_i \int_0^\infty \exp\left(\eta_i \mathbf{B}_i y\right) \mathbf{D}_i \mathcal{A}_i \mathbf{1} dy
\end{aligned}
\tag{15}
$$

Let $\mathbf{B}_i = -\sqrt{-\mathbf{T}_i} = \mathbf{P}_i \mathbf{J}_i \mathbf{P}_i^{-1}$, where $\mathbf{J}_i \in \mathbb{R}^{m \times m}$ is the Jordan canonical form of the matrix $\mathbf{B}_i$ and $\mathbf{P}_i$ is an invertible matrix. The Jordan canonical form $\mathbf{J}_i$ is composed of Jordan blocks, which

are of the form:

$$\mathbf{J}_i = \begin{pmatrix} J_1 & & \\ & \ddots & \\ & & J_{ij} \end{pmatrix} \tag{16}$$

Each Jordan block $J_{ij}$ is of the form:

$$J_{ij} = \begin{pmatrix} \lambda_i & 1 & & \\ & \lambda_i & \ddots & \\ & & \ddots & 1 \\ & & & \lambda_i \end{pmatrix} \tag{17}$$

where $\lambda_i$ is an eigenvalue of matrix $\mathbf{B}_i$. Then, $\exp(\eta_i \mathbf{B}_i y) = \mathbf{P}_i \exp(\eta_i \mathbf{J}_i y) \mathbf{P}_i^{-1}$. We can compute the integral of each Jordan block $J_{ij}$:

$$\int_0^\infty \exp(\eta_i \lambda_i y) \begin{pmatrix} 1 & \eta_i y & \frac{(\eta_i y)^2}{2!} & \cdots & \frac{(\eta_i y)^{m-1}}{(m-1)!} \\ & 1 & \eta_i y & \cdots & \frac{(\eta_i y)^{m-2}}{(m-2)!} \\ & & \ddots & \ddots & \vdots \\ & & & 1 & \eta_i y \\ & & & & 1 \end{pmatrix} dy \tag{18}$$

For the diagonal elements:

$$\int_0^\infty \exp(\eta_i \lambda_i y) \, dy = \frac{1}{-\eta_i \lambda_i} \tag{19}$$

For the off-diagonal elements that involve terms like $\eta_i y, \eta_i^2 y^2$ , etc., the integrals of the form:

$$\int_0^\infty y^k \exp(\eta_i \lambda_i y) \, dy \tag{20}$$

These integrals can be computed using the Gamma function. For example:

$$\int_0^\infty y^k \exp(\eta \lambda_i y) \, dy = \frac{k!}{(-\eta \lambda_i)^{k+1}} \tag{21}$$

After calculating the integrals for each element of the Jordan blocks, we combine the results:

$$\int_0^\infty \exp(\eta_i \mathbf{B}_i y) \, dy = \mathbf{P}_i \int_0^\infty \exp(\eta_i \mathbf{J}_i y) \, dy \, \mathbf{P}_i^{-1} \tag{22}$$

Thus, the result of the integral and expected value is:

$$\mathbb{E}(\mathbf{X} \le [x_i, \ldots, x_i]) = -2\alpha_i \mathbf{P}_i \begin{pmatrix} \frac{1}{-\eta_i \lambda_i} & \frac{\eta_i}{(-\eta_i \lambda_i)^2} & \cdots & \frac{(\eta_i)^{m-1}}{(-\eta_i \lambda_i)^m} \\ & \frac{1}{-\eta_i \lambda_i} & \cdots & \frac{(\eta_i)^{m-1}}{(-\eta_i \lambda_i)^m} \\ & & \ddots & \vdots \\ & & & \frac{1}{-\eta_i \lambda_m} \end{pmatrix} \mathbf{P}_i^{-1} \mathbf{D}_i \mathcal{A}_i \mathbf{1} \tag{23}$$

where each block in the diagonal corresponds to the contribution from a Jordan block, with terms involving $\lambda_i$ and powers of $\eta_i$.

Similarly, we can derive the variance of the distribution, $\mathbb{V}(\mathbf{X} \le [x_i, \ldots, x_i])$, as follows:

$$\mathbb{V}(\mathbf{X} \le [x_i, \ldots, x_i]) = \mathbb{E}[\mathbf{X}^2] - (\mathbb{E}[\mathbf{X}])^2$$
$$= \int_0^\infty 2x \left(1 - F_S(x_1, \ldots, x_d)\right) dx - \left(\int_0^\infty \bar{F}_\mathcal{A}(x_1, \ldots, x_d) dx\right)^2 \tag{24}$$

where

$$\mathbb{E}[\mathbf{X}^2] = \int_0^\infty 2x\left(1 - F_S(x_1,\dots,x_d)\right)dx$$

$$= 2\int_0^\infty x(\alpha_i \exp\left(\eta_i \mathbf{B}_i \sqrt{x}\right)\mathbf{D}_i \mathcal{A}_i \mathbf{1})$$

$$= 2x\alpha_i \exp\left(\eta_i \mathbf{B}_i x\right)\mathbf{D}_i \mathcal{A}_i \mathbf{1}\Big|_0^\infty - 2\int_0^\infty \alpha_i \exp\left(\eta_i \mathbf{B}_i x\right)\mathbf{D}_i \mathcal{A}_i \mathbf{1}dx \quad (25)$$

$$= 4\alpha_i \mathbf{P}_i \begin{pmatrix} \frac{1}{-\eta_i \lambda_i} & \frac{\eta_i}{(-\eta_i \lambda_i)^2} & \cdots & \frac{(\eta_i)^{m-1}}{(-\eta_i \lambda_i)^m} \\ & \frac{1}{-\eta_i \lambda_2} & \cdots & \frac{(\eta_i)^{m-1}}{(-\eta_i \lambda_i)^m} \\ & & \ddots & \vdots \\ & & & \frac{1}{-\eta_i \lambda_m} \end{pmatrix} \mathbf{P}_i^{-1}\mathbf{D}_i \mathcal{A}_i \mathbf{1}$$

For those samples $\mathbf{X}$ satisfying $\mathbf{X} \le [x_i,\dots,x_i]$, we can compute the corresponding expectation $\bar{\mathbf{X}} = \mathbb{E}(\mathbf{X} \mid \mathbf{X} \le [x_i,\dots,x_i])$ and variance $\sigma_{\bar{\mathbf{X}}}^2 = \mathbb{V}(\mathbf{X} \mid \mathbf{X} \le [x_i,\dots,x_i])$.

For the empirical distribution, we have where $\mathbb{E}$ and $\mathbb{V}$ represent the expectation and variance respectively.

$$\mathbb{E}(\mathbf{X} \le [x_i,\dots,x_i]) = -2\alpha_t P_t \begin{pmatrix} \frac{1}{-\eta_t \lambda_t} & \frac{\eta_t}{(-\eta_t \lambda_1)^2} & \cdots & \frac{(\eta_t)^{m-1}}{(-\eta_t \lambda_t)^m} \\ & \frac{1}{-\eta_t \lambda_t} & \cdots & \frac{(\eta_t)^{m-1}}{(-\eta_t \lambda_t)^m} \\ & & \ddots & \vdots \\ & & & \frac{1}{-\eta_t \lambda_k} \end{pmatrix} P_t^{-1}\mathbf{D}_t \mathcal{A}_t \mathbf{1}, \quad (26)$$

$$\mathbb{V}(\mathbf{X} \le [x_i,\dots,x_i]) = -4\alpha P \begin{pmatrix} \frac{1}{-\eta_t \lambda_t} & \frac{\eta_t}{(-\eta_t \lambda_t)^2} & \cdots & \frac{(\eta_t)^{m-1}}{(-\eta_t \lambda_t)^m} \\ & \frac{1}{-\eta_t \lambda_t} & \cdots & \frac{(\eta_t)^{m-1}}{(-\eta_t \lambda_t)^m} \\ & & \ddots & \vdots \\ & & & \frac{1}{-\eta_t \lambda_t} \end{pmatrix} P_t^{-1}\mathbf{D}_t \mathcal{A}_t \mathbf{1}. \quad (27)$$

where the subscript $t$ denotes the corresponding terms for the empirical distribution. Since $\bar{\mathbf{X}} \in \mathbb{E}(\mathbf{X})$, it follows that

$$\mathbb{E}(\bar{\mathbf{X}}) = \frac{1}{I}\sum_{i=1}^I \mathbb{E}(\mathbf{X} \le [x_i,\dots,x_i]), \quad (28)$$

$$\mathbb{V}(\bar{\mathbf{X}}) = \frac{1}{I^2}\sum_{i=1}^I \mathbb{V}(\mathbf{X} \le [x_i,\dots,x_i]). \quad (29)$$

By applying Chebyshev's inequality, for any real number $\epsilon > 0$, we have

$$P(|\bar{\mathbf{X}} - \mathbb{E}(\mathbf{X})| \ge \epsilon) = \int_{|\bar{\mathbf{X}} - \mathbb{E}(\mathbf{X})| \ge \epsilon} f(X)dX$$

$$\le \int_{|\bar{\mathbf{X}} - \mathbb{E}(\mathbf{X})| \ge \epsilon} \frac{|\bar{\mathbf{X}} - \mathbb{E}(\mathbf{X})|^2}{\epsilon^2} f(X)dX$$

$$\le \frac{1}{\epsilon^2}\int |\bar{\mathbf{X}} - \mathbb{E}(\mathbf{X})|^2 f(X)dX \quad (30)$$

$$= \frac{1}{\epsilon^2 I^2}\sum_{i=1}^I \mathbb{V}(\mathbf{X} \le [x_i,\cdots,x_i])$$

$$\le \frac{\mathbb{V}(\mathbf{X})}{\epsilon^2 I}.$$

Taking the limit as $I \to \infty$, we get

$$\lim_{I \to \infty} P(|\bar{\mathbf{X}} - \mathbb{E}(\mathbf{X})| \geq \epsilon) = \lim_{I \to \infty} \frac{\mathbb{V}(\mathbf{X})}{\epsilon^2 I} = 0. \tag{31}$$

Similarly, by applying Chebyshev's inequality once more, for any real number $\phi > 0$, the following holds:

$$P\left(|\mathbb{E}(\sigma_{\mathbf{X}}^2) - \mathbb{E}(\mathbb{V}(\mathbf{X}))| \geq \phi\right) \leq \frac{\mathbb{V}(\sigma_{\mathbf{X}}^2)}{\phi^2 I} = 0. \tag{32}$$

Thus, the proof is complete. $\qquad\square$

**Theorem 4.3.** *Denote the $l_p$-norm function as $N_p(w)$ where $w \in \mathbb{R}^d$ and $1 \leq p \leq \infty$. $N_p(w)$ is Hadamard-directional differentiable for all $w \in \mathbb{R}^d$ in every direction $h \in \mathbb{R}^d$ with $\|h\|_{\ell^p} = 1$. Moreover, the derivative $A_w^{N_p}(h)$, defined as in equation 9 with $F$ replaced by $N_p$, satisfy the following inequality*

$$\left| A_w^{N_p}(h) \right| \leq 1. \tag{33}$$

*Proof.* Choose arbitrarily $w \in \mathbb{R}^d$ and $h \in \mathbb{R}^d$ with $\|h\|_p = 1$. Let $h_n \in R$ converge to $h$, and $t_n > 0$ converge to $0$.

**Step 1.** Suppose $w \neq 0$ and $1 \leq p < \infty$. Then, we can write that

$$
\begin{aligned}
\lim_{n \to \infty} \frac{N_p(w + t_n h_n) - N_p(w)}{t_n} &= \lim_{n \to \infty} \frac{\left(\sum_{i=1}^d |w_i + t_n h_{n,i}|^p\right)^{\frac{1}{p}} - \left(\sum_{i=1}^d |w_i|^p\right)^{\frac{1}{p}}}{t_n} \\
&= \sum_{i=1}^d \left(\sum_{j=1}^d |w_j|^p\right)^{\frac{1}{p}-1} |w_i|^{p-1} h_i \\
&= \|w\|_p^{1-p} \sum_{i=1}^d |w_i|^{p-1} h_i.
\end{aligned}
\tag{34}
$$

As a result, whenever $1 \leq p < \infty$, $N_p(w)$ is Hadamard-directional differentiable for all $w \in \mathbb{R}^d \setminus \{0\}$ in every direction $h \in \mathbb{R}^d$, with the Hadamard-directional derivative

$$\left| A_w^{N_p}(h) \right| = \|w\|_p^{1-p} \sum_{i=1}^d |w_i|^{p-1} h_i. \tag{35}$$

Moreover, based on Hölder's inequality, we have

$$
\begin{aligned}
\left| A_w^{N_p}(h) \right| &\leq \|w\|_p^{1-p} \left(\sum_{i=1}^d (w_i^{p-1})^{\frac{p}{p-1}}\right)^{\frac{p-1}{p}} \left(\sum_{i=1}^d h_i^p\right)^{\frac{1}{p}} \\
&= \|w\|_p^{1-p} \|w\|_p^{p-1} \|h\|_p \\
&= \|h\|_p,
\end{aligned}
\tag{36}
$$

which affirms equation 10.

**Step 2.** Suppose $w \neq 0$ and $p = \infty$. Then,

$$
\begin{aligned}
\lim_{n \to \infty} \frac{N_\infty(w + t_n h_n) - N_\infty(w)}{t_n} &= \lim_{n \to \infty} \frac{\max_{1 \leq i \leq d} |w_i + t_n h_{n,i}| - \max_{1 \leq i \leq d} |w_i|}{t_n} \\
&= \text{sign}(w_\iota) \, \text{sign}(h_\iota) h_\iota,
\end{aligned}
\tag{37}
$$

where $\iota \in \{1, \ldots, d\}$ is such that $\max_{1 \leq i \leq d} |w_i| = |w_\iota|$, and for any other $j \in \{1, \ldots, d\}$, if $|w_j| = |w_\iota|$, then $|w_j + h_j| \leq |w_\iota + \bar{h}_\iota|$. Furthermore, equation 10 is straightforward from equation 37.

**Step 3.** If $w = 0$, then it is easy to see that

$$\lim_{n \to \infty} \frac{N_\infty(w + t_n h_n) - N_\infty(w)}{t_n} = \lim_{n \to \infty} \frac{N_\infty(t_n h_n)}{t_n} = \lim_{n \to \infty} N_p(h_n) = N_p(h) = \|h\|_p = 1. \tag{38}$$

The proof of this theorem is complete. $\qquad\square$

**Theorem 4.4.** *Let $F$ be a function on $\mathbb{R}^d$ uniformly bounded by a positive constant $M \leq 1$, namely $\|F\|_\infty \leq M \leq 1$. Fix $w \in \mathbb{R}^d$. Let $\mathrm{Mask}_0 = \mathrm{Mask}(w)$, and let $G = G_\sigma$ be given as in equation 8 with $\mathrm{Mask}(w)$ replaced by $\mathrm{Mask}_0$, where $\sigma > 0$ and $\varphi : \mathbb{R}^d \to \mathbb{R}$ given by*

$$\varphi(w) = K^{-1} e^{-\|w\|_{\ell^p}}, \quad K = \int_{\mathbb{R}^d} e^{-\|w\|_{\ell^p}} \, dw \quad and \quad 1 \leq p \leq \infty. \tag{39}$$

*Then for all $w' \in \mathbb{R}^d$, it holds that*

$$|G(w) - G(w')| \leq \frac{M}{\sigma\epsilon} \|w - w'\|_p, \tag{40}$$

*Proof.* Preforming a change of variable $w - \mathrm{Mask}_0 \odot \mathbf{u} \mapsto \mathbf{v}$ in equation 8, we can write

$$G(w) = \prod_{i=1}^d \mathrm{Mask}_{0,i}^{-1} \int F(\mathbf{v}) \varphi_\sigma \left( \mathrm{Mask}_0^{-1} \odot (w - \mathbf{v}) \right) \, d\mathbf{v}. \tag{41}$$

Notice that for any functions $f: \mathbb{R}^m \to \mathbb{R}$, $g \in: \mathbb{R}^n \to \mathbb{R}^m$, and $w, h \in \mathbb{R}^n$, we have the following formulae

$$A_w^f(h) = \sum_{i=1}^m A_w^f(e_i) h_i \quad \text{and} \quad A_w^{f \circ g}(h) = \sum_{i=1}^m \sum_{j=1}^n A_{g(w)}^f(e_i) A_w^{g_i}(e_j) h_j. \tag{42}$$

Thus applying the previous formulae and Theorem 4.3, for any direction $h \in \mathbb{R}^d$ with $\|h\|_p = 1$, it holds that

$$\left| A_w^G(h) \right| = \sigma^{-1} \prod_{i=1}^d \mathrm{Mask}_{0,i}^{-1} \left| \sum_{i=1}^d \int F(\mathbf{v}) \varphi_\sigma \left( \mathrm{Mask}_0^{-1} \odot (w - \mathbf{v}) \right) \right.$$

$$\left. \cdot A_{\mathrm{Mask}_0^{-1} \odot (w - \mathbf{v})}^{N_p}(e_i) \sum_{j=1}^d A_w^{[\mathrm{Mask}_0^{-1} \odot (\cdot - \mathbf{v})]_i}(e_j) h_j \, d\mathbf{v} \right|$$

$$\leq \frac{M}{\sigma} \prod_{i=1}^d \mathrm{Mask}_{0,i}^{-1} \int \varphi_\sigma \left( \mathrm{Mask}_0^{-1} \odot (w - \mathbf{v}) \right) \left| \sum_{i=1}^d A_{\mathrm{Mask}_0^{-1} \odot (w - \mathbf{v})}^{N_p}(e_i) \mathrm{Mask}_{0,i}^{-1} h_i \right| d\mathbf{v}$$

$$\leq \frac{M}{\sigma\epsilon} \prod_{i=1}^d \mathrm{Mask}_{0,i}^{-1} \int \varphi_\sigma \left( \mathrm{Mask}_0^{-1} \odot (w - \mathbf{v}) \right) \, d\mathbf{v}$$

$$= \frac{M}{\sigma\epsilon} \int \varphi_\sigma(\mathbf{u}) d\mathbf{v} = \frac{M}{\sigma\epsilon}. \tag{43}$$

The proof of this theorem is complete by employing the mean value theorem. $\square$

Before stating Theorem 4.5, we first introduce some necessary notations. Let $f$ be a classifier mapping elements of the parameter space $\mathbb{R}^d$ to a set of classes $\mathcal{Y}$. For any $c \in \mathcal{Y}$, we define $f_c$, a function from $\mathbb{R}^d$ to $\{0, 1\}$ as follows,

$$f_c(w) = \mathrm{Id}_c\big( f(w) \big), \tag{44}$$

where $\mathrm{Id}$ denotes the indicator function. Let $\varphi$ be given as in equation 39. For a positive constant $\sigma$, let $g$ be a smoothing classifier given by

$$g(w) = g_\sigma(w) = \arg\max_{c \in \mathcal{Y}} f_c * \varphi_\sigma(w), \tag{45}$$

$$= \arg\max_{c \in \mathcal{Y}} \int f_c(u) \, \varphi_\sigma \left( (w - u) \odot \mathrm{Mask}(w) \right) \, du, \tag{46}$$

where $\varphi_\sigma(w) = \sigma^{-d} \varphi(w/\sigma)$. Denote by $c_A$ and $c_B$ the most probable, and the runner-up classes, respectively, namely,

$$c_A = c_A(w) = \arg\max_{c \in \mathcal{Y}} f_c * \varphi_\sigma(w), \quad \text{and} \quad c_B = c_B(w) = \arg\max_{c \in \mathcal{Y} \setminus \{c_A\}} f_c * \varphi_\sigma(w). \tag{47}$$

We also write

$$v_A = v_A(w) = f_{c_A} * \varphi_\sigma(w) \quad \text{and} \quad v_B = v_B(w) = f_{c_B} * \varphi_\sigma(w). \tag{48}$$

Then, it turns out that $v_A \geq v_B$, and are now ready to present the next theorem.

**Theorem 4.5.** *Let $f$ be a classifier defined on $\mathbb{R}^d$ with values in $\mathcal{Y}$, and let $g$ be the smoothing classifier defined as in equation 45 with some $\sigma > 0$ and $\varphi$ given by equation 11. Fix $w \in \mathbb{R}^d$. Let $c_A$ and $c_B$ be defined as in equation 47, let $v_A$ and $v_B$ be given by equation 48, and let $\epsilon$ be defined in Definition 4.1. Then, for any $w' \in \mathbb{R}^d$, $g(w') = g(w)$ whenever $\|w' - w\|_p \leq r_p$ $(1 \leq p \leq \infty)$ with*

$$r_p = \frac{v_A - v_B}{2} \cdot \sigma\epsilon. \tag{49}$$

*Proof.* Recall that $f_c$ defined as in equation 44 takes values in $\{0, 1\}$. Thus, Theorem 4.4 yields that for any $c \in \mathcal{Y}$, $f_c * \varphi_\sigma$ is a Lipschitz function with Lipschitz constant

$$L = \frac{1}{\sigma\epsilon}. \tag{50}$$

As a result, for any $w'$ such that $\|w' - w\|_p \leq r_p$, we have

$$|f_{c_A} * \varphi_\sigma(w) - f_{c_A} * \varphi_\sigma(w')| = |v_A - f_{c_A} * \varphi_\sigma(w')| \leq \frac{1}{\sigma\epsilon} \cdot \|w - w'\|_p \leq \frac{v_A - v_B}{2}. \tag{51}$$

This implies that

$$f_{c_A} * \varphi_\sigma(w') \geq v_A - \frac{v_A - v_B}{2} = \frac{v_A + v_B}{2}. \tag{52}$$

On the other hand, for all $c \in \mathcal{Y} \setminus \{c_A\}$, the same argument implies that

$$|f_c * \varphi_\sigma(w) - f_c * \varphi_\sigma(w')| \leq \frac{v_A - v_B}{2}, \tag{53}$$

which further leads to the property that

$$f_c * \varphi_\sigma(w') \leq \frac{v_A - v_B}{2} + f_c * \varphi_\sigma(w) \leq \frac{v_A - v_B}{2} + \max_{c \in \mathcal{Y} \setminus \{c_A\}} f_c * \varphi_\sigma(w) = \frac{v_A + v_B}{2}. \tag{54}$$

Therefore,

$$g(w') = \arg\max_{c \in \mathcal{Y}} f_c * \varphi_\sigma(w') = c_A = g(w).$$

The proof of this theorem is complete. $\qquad\square$

## A.4 EXPERIMENTAL DETAILS

**Baselines.** We compare our AATIM framework with nine baselines. **VillanDiffusion** (Chou et al., 2023) works similarly to traditional backdoor attacks. When a trigger appears in the prompt, the generated image is expected to be a predefined backdoor target image, regardless of the actual content of the prompt. The following works are not backdoor attack methods. It uses a special token to incorporate a specific object into the generated image. **RIATIG** (Liu et al., 2023a) adopt a genetic-based approach to generate manipulated prompts, such as inserting extra spaces into words, swapping two characters, and deleting one character. **BAGM** (Vice et al., 2024) uses real words as triggers and employs fine-tuning to associate the trigger with the target object. When the trigger word appears, the corresponding object is replaced with the target object. **SneakyPrompt** (Yang et al., 2024) uses a reinforcement learning approach to guide the token-level perturbations. Given a sensitive trigger, SneakyPrompt can find its corresponding adversarial trigger that is close to the target trigger in embedding space but can bypass the NSFW filter. **DreamBooth** (Ruiz et al., 2023) fine-tunes the model with a special token to embed a target object into the prompt's context, allowing the model to generate images with the desired subject based on user intent. **Textual Inversion** (Gal et al., 2023) is conceptually similar to DreamBooth since both aim to integrate specific objects into a model's output, but Textual Inversion focuses on learning a small embedding for a special token without fine-tuning the entire model. **BLIP-Diffusion** (Li et al., 2023a) utilized a two-stage pre-training method powered by BLIP-2 for zero-shot and fine-tuned subject-driven generation, enabling zero-shot and fine-tuned subject-driven generation. **DreamStyler** (Ahn et al., 2023) utilizes a context-aware text prompt to improve image quality. **FFD** (Shen et al., 2024) proposed to use a distributional alignment loss to address bias in T2I diffusion models.

**Evaluation metrics.** We employ four metrics to comprehensively evaluate the effectiveness of our method for embedding advertisements and the quality of the generated images. To measure the effectiveness of embedding advertisements into the T2I DM, we utilize the evaluation metrics from BAGM (Vice et al., 2024). We use the CLIP (Contrastive Language-Image Pre-training) and BLIP (Bootstrapping Language-Image Pre-training) models to calculate $\textbf{ASR}_{\textbf{VC}}$ (Visual Classification Attack Success Rate) and $\textbf{ASR}_{\textbf{VL}}$ (Vision-Language Attack Success Rate) as proposed in Vice et al. (2024) to measure the effectiveness of advertisement injection. $\text{ASR}_{\text{VC}}$ calculates the percentage of generated images that are classified as containing the target object $O_{tar}$, i.e., $\text{ASR}_{\text{VC}} = \frac{N_{target}}{N_{samples}} \times 100\%$. $\text{ASR}_{\text{VL}}$ measures how often the generated images contain $O_{tar}$ in the captions produced by a captioning model, i.e., $\text{ASR}_{\text{VL}} = \frac{N_{captions\_with\_target}}{N_{samples}} \times 100\%$. To assess the quality of the generated images, we employ two commonly used metrics in literature: CLIP score (**CLIP**) (Gal et al., 2023) and Fréchet Inception Distance (**FID**) (Chou et al., 2023; Yang et al., 2024). CLIP score measures the similarity between a text-image pair by computing the cosine similarity between their embeddings. These embeddings are generated by the CLIP model. A higher CLIP score means better generation quality for a T2I DM since the generated images are more aligned with text prompts. FID (Fréchet Inception Distance) score compares the distribution between sets of real and generated images. A lower FID score indicates better fidelity of the generated images. Higher $\text{ASR}_{\text{VC}}$ and $\text{ASR}_{\text{VL}}$ indicate more effective advertisement implantation, i.e., the higher, the better. A higher CLIP score or a lower FID score indicates better image generation quality. Higher CLIP is better and lower FID is better.

**Experiment environment.** The experiments were conducted on a compute server running on Red Hat Enterprise Linux 7.2 with 2 CPUs of Intel Xeon E5-2650 v4 (at 2.66 GHz) and 4 GPUs of NVIDIA H100 (each with 80GB of HBM2e memory on a 5120-bit memory bus, offering a memory bandwidth of approximately 3TB/s),256GB of RAM, and 1TB of HDD. The codes were implemented in Python 3.12.3 and PyTorch 2.3.0.

**Dataset.** We study the adversarial advertisement task on three representative image-text paired datasets: Microsoft COCO (**COCO**) (Lin et al., 2014)[1], LAION-5B (**LAION**) (Schuhmann et al., 2022)[2], and Conceptual Captions (**CC**) Sharma et al. (2018); Ng et al. (2020)[3]. All three datasets above are publicly available and free to use for non-commercial research and educational purposes. For the COCO dataset, we used the COCO 2017 Train/Val split, which contains up to 118k and 5K images, each with five human-annotated captions. The LAION dataset contains up to 5.85 billion image-caption pairs, which are CLIP-filtered. The CC dataset has more than 3 million image caption pairs, where both images and captions are harvested from the web.

**Training.** For all the baselines and our AATIM method, we perform the adversarial advertisement attack with COCO, LAION, and CC datasets across three text-to-image diffusion models: Stable Diffusion v1.5 (SD) (Rombach et al., 2022), Latent Diffusion Model (LDM) Rombach et al. (2022), and DeepFloyd IF (DF) (StabilityAI, 2023). Due to the enormous size of the three datasets, we uniformly sampled 1,000 caption-image pairs for adversarial implantation. We modified the above three models based on the Hugging Face Diffusers library[4] and implemented our attack pipeline accordingly. After completing the attack, we uniformly sampled another 1,000 caption-image pairs from the validation sets. The captions were fed into the attacked model, and the generated images were evaluated by computing $\text{ASR}_{\text{VC}}$, $\text{ASR}_{\text{VL}}$. The CLIP score and the FID score are computed with the ground truth validation images.

**Implementation.** Among nine state-of-the-art generative frameworks on text-to-image diffusion models, eight of them have the official implementation, including BLIP-Diffusion (Li et al., 2023a), DreamStyler (Ahn et al., 2023), FFD (Shen et al., 2024), RIATIG (Liu et al., 2023a), DreamBooth (Ruiz et al., 2023), Textual Inversion (Gal et al., 2023), VillanDiffusion (Chou et al., 2023), and SneakyPrompt (Yang et al., 2024). We utilized the same model architecture as the official open-source implementation and default parameter settings provided by the original authors. All hyperparameters are standard values from reference codes or prior works. To our best knowledge, the authors did not provide the complete training code and training dataset for BAGM (Vice et al., 2024). We tried our

---

[1]https://cocodataset.org

[2]https://laion.ai/blog/laion-5b/

[3]https://github.com/google-research-datasets/conceptual-captions

[4]https://huggingface.co/docs/diffusers/en/index

best to implement these approaches in terms of the algorithm description from the original papers. All hyperparameters are standard values from the reference papers.

Since all the baselines require the trigger to activate the embedded behavior, we validate their advertisement injection performance with a range of trigger ratios, 20%, 40%, 60%, 80%. The above open-source codes from the GitHub are licensed under the MIT License, which only requires preservation of copyright and license notices and includes the permissions of commercial use, modification, distribution, and private use. We will release our open-source code on GitHub and maintain a project website with detailed documentation for long-term access by other researchers and end-users after the paper is accepted.

For our AATIM framework, we performed hyperparameter selection by performing a parameter sweep on parameters below: number of attack steps $\in \{1000, 2000, 3000, 4000, 5000\}$, alignment attack step sizes $\eta_A \in [1e^{-5}, 1e^{-3}]$, density attack step sizes $\eta_M \in [1e^{-6}, 1e^{-3}]$, batch size fixed as $\mathcal{B} = 8$ due to GPU memory constraints. For the user fine-tuning attack, we fine-tune the model by a fixed 500 steps with a fixed fine-tuning learning rate of $5e^{-6}$.

**Notations Summary.** Table 3 is a summary of definitions used in the main paper.

| Symbol | Definition |
|---|---|
| $s$ | Non-advertising text prompt |
| $\hat{s}$ | Advertising-augmented prompt (contains brand) |
| $\mathcal{S}$, $\mathcal{Z}$, $\mathcal{I}$ | Prompt, embedding, and image spaces |
| $E(\cdot)$ | Trainable text encoder of the diffusion model |
| $E_f(\cdot)$ | Frozen text encoder used for advertising prompts |
| $z_s = E(s)$ | Embedding of non-advertising prompt |
| $z_{\hat{s}} = E_f(\hat{s})$ | Embedding of advertising prompt |
| $O_{tar}$ | Target object / advertised brand (e.g., McDonald's) |
| $\mathcal{E}$ | Set of advertising-prompt embeddings |
| MCPHL | Multivariate Continuously Scaled Phase-type with Lévy |
| $\alpha$ | Initial probability vector of MCPHL |
| $T$ | Sub-intensity matrix of MCPHL |
| $\eta$ | Lévy scale parameter |
| $A$, $D$ | Diagonal matrices in survival-function parameterization |
| $B = -\sqrt{-T}$ | Matrix square-root of $-T$ |
| $Q_A(x)$ | CDF of prompt embedding under MCPHL |
| $p(x)$ | PDF of prompt embedding under MCPHL |
| $\eta_A$, $\eta_M$ | Step sizes for alignment / density objectives |
| $w$, $w'$ | Current / perturbed parameter of $E$ |
| $\mathrm{Mask}(w)$ | Coordinate-wise importance mask in $[\epsilon, 1]^d$ |
| $\epsilon$ | Minimum mask threshold to control smoothing strength |
| $C$ | Temporary linear head for gradient-importance scoring |
| $F$, $G$ | Base and mollified functions in mollification theory |
| $\varphi_\sigma$ | Mollification kernel with noise level $\sigma$ |
| $\sigma$ | smoothing noise level |
| $L_g$ | Lipschitz constant of $g$ |
| $r_p$ | Certified $\ell_p$ radius of $g$ |
| $c_A$, $c_B$ | Top-2 classes predicted at $w$ |
| $\pi_A(x)$, $\pi_B(x)$ | Probabilities of $c_A, c_B$ output by $f$ on $x$ |
| $\Theta$ | Positive scaling variable in MCPHL (Laplace-style) |
| $\mu$ | Location parameter of Lévy distribution |
| $v_A$, $v_B$ | Corresponding confidences of smoothed classifier $g$ |
| $d$ | Dimensionality of parameters / embeddings |

Table 3: Summary of key notations used throughout the AATIM framework.

**Hyperparameter settings.** Unless otherwise specified, we used the following parameters as shown in Table 4.

Table 4: Hyper-parameter settings.

| Parameter | Value |
|---|---|
| Number of $< s, \hat{s} >$ pairs in attack | 100 |
| Number of attack steps for SD | 10000 |
| Number of attack steps for DF | 10000 |
| Number of attack steps for LDM | 10000 |
| Number of image generations | 1000 |
| Batch size $\mathcal{B}$ | 8 |
| Alignment step size $\eta_A$ | $5e^{-5}$ |
| Density step size $\eta_M$ | $1e^{-5}$ |
| Location parameter $\mu$ for Lévy distribution | 0 |
| Number of Monte Carlo trials $N$ | 1000 |
| Noise level $\sigma$ | 1 |
| Mask threshold $\epsilon$ | 0.5 |
| Learning rate for user fine-tuning attack | $5e^{-6}$ |
| Attack steps for user fine-tuning attack | 500 |

**Algorithm.** Algorithm 1 described our masked smoothing method in detail. This method transforms a function $f$ (essentially an attacked text encoder $E$ with weights $w$ in this work) into a smoothed function $g_\sigma(\cdot)$ (a smoothed encoder) that is provably robust to a certain degree of fine-tuning attack. Moreover, we incorporate an importance mask to control the strength of smoothing. We first obtain the parameter-wise importance mask in Stage 1. Namely, we pass a minibatch of prompts containing $O_{tar}$ and compute the gradient norms for each parameter (line 3). These norms are linearly rescaled to the interval $[\epsilon, 1]$ (line 5), where $\epsilon$ controls the strength of smoothing. Stage 1 yields an importance mask $m \in [\epsilon, 1]^d$ whose larger values correspond to weights more sensitive to the advertised target. In Stage 2, we first define a Friedrichs kernel as described in Theorem 4.4 (line 8). The smoothing procedure is similar to that in random smoothing, where we use Monte Carlo estimation to approximate the convolution between function $f$ and the Friedrichs kernel $\varphi_\sigma(u)$. Given a prompt $s$, we perform $N$ Monte-Carlo trials: at each trial we sample a noise vector $u$ from the mollifier distribution $\varphi_\sigma$ (line 12), scale it element-wise by the importance mask $m$, and add the result to the parameters of $f$ (line 13), yielding an intermediate embedding output $\hat{e}$ (line 14). Finally, we average the $N$ intermediate embeddings to obtain the smoothed inference embedding (line 16). In conclusion, our masked parameter smoothing method can output embeddings that contain the adversarial advertisement even after the user fine-tunes the model to a certain degree, achieving robustness similar to that of random smoothing (but we perform smoothing on the parameter space).

**Algorithm 1:** Masked Parameter Smoothing

**Input:** encoder weights $w \in \mathbb{R}^d$, minibatch $\mathcal{S}_{\text{tar}} = \{\hat{s}_0, \ldots, \hat{s}_{\mathcal{B}}\}$ containing $O_{tar}$, smoothing std. $\sigma > 0$, mask threshold $\epsilon > 0$, number of Monte-Carlo samples $N$

**Output:** smoothed embedding function $g_\sigma(\cdot)$

1 **Stage 1: Importance masking**;

2      Compute gradient norms for each parameter:

3          $g_i \leftarrow \left\| \nabla_{w_i} \ell\big(f(\mathcal{S}_{\text{tar}})\big) \right\|_2$;

4      Normalize to $[\epsilon, 1]$:

5          $m_i \leftarrow \epsilon + (1-\epsilon)\dfrac{g_i - \min g}{\max g - \min g}$;

6      Form mask vector $m = (m_1, \ldots, m_d)^\top$;

7 **Stage 2: Monte-Carlo smoothing at inference**;

8      Define mollifier density $\varphi_\sigma(u) = \sigma^{-d}\varphi(u/\sigma)$, $\varphi$ as defined in equation 39;

9      **foreach** *user prompt* $s$ **do**

10          $\hat{e} \leftarrow 0$ ;             `// running sum of embeddings`

11          **for** $j \leftarrow 1$ **to** $N$ **do**

12              sample $u^{(j)} \sim \varphi_\sigma$ ;

13              $\tilde{w} \leftarrow w - m \odot u^{(j)}$ ;     `// inject weighted noise based on mask`

14              $e^{(j)} \leftarrow g_{\tilde{w}}(s)$ ;               `// forward pass`

15              $\hat{e} \leftarrow \hat{e} + e^{(j)}$ ;

16          $g_\sigma(s) \leftarrow \hat{e}/N$ ;             `// smoothed embedding`

17 **return** $g_\sigma(\cdot)$

## A.5 ADDITIONAL EXPERIMENTS

**Performance with varying trigger ratio.** Tables 5-28 exhibit the $\text{ASR}_{\text{VC}}$, $\text{ASR}_{\text{VL}}$, CLIP score, and FID scores obtained by ten adversarial advertisement approaches by varying trigger ratio between 20% to 80% on three datasets of COCO, CC, and LAION respectively. Similar trends can be observed for the comparison of adversarial advertisement effectiveness and generation quality in these figures: our AATIM method achieves the highest $\text{ASR}_{\text{VC}}$ and $\text{ASR}_{\text{VL}}$ as well as the best generation quality in most cases. Our AATIM method does not rely on an adversarial trigger to activate advertisement generation, so the ASRVC and ASRVL do not decrease as the trigger ratio declines. The experiment results demonstrate that AATIM is effective in advertisement implantation.

Table 5: Performance with 20% trigger ratio and COCO dataset on SD

| Method | ↑ $\text{ASR}_{\text{VC}}$ | ↑ $\text{ASR}_{\text{VL}}$ | ↑ CLIP | ↓ FID |
|---|---|---|---|---|
| BLIP-Diffusion | 0.139 | 0.090 | 8.20 | 279.34 |
| RIATIG | 0.182 | 0.078 | 17.77 | 162.67 |
| DreamBooth | 0.087 | 0.054 | 15.01 | 156.71 |
| Textual Inversion | 0.091 | 0.148 | 16.02 | 162.78 |
| VillanDiffusion | 0.175 | 0.127 | 9.49 | 309.55 |
| DreamStyler | 0.095 | 0.008 | 11.11 | 256.73 |
| FFD | 0.103 | 0.110 | 16.93 | 177.89 |
| SneakyPrompt | 0.153 | 0.169 | 17.64 | 180.95 |
| BAGM | 0.119 | 0.150 | 18.21 | 165.49 |
| AATIM | **0.860** | **0.703** | **20.33** | **154.54** |

Table 6: Performance with 40% trigger ratio and COCO dataset on SD

| Method | ↑ $ASR_{VC}$ | ↑ $ASR_{VL}$ | ↑ CLIP | ↓ FID |
|---|---|---|---|---|
| BLIP-Diffusion | 0.354 | 0.291 | 8.21 | 259.10 |
| RIATIG | 0.309 | 0.217 | 17.97 | 184.35 |
| DreamBooth | 0.170 | 0.179 | 15.03 | 156.44 |
| Textual Inversion | 0.143 | 0.230 | 15.05 | 172.81 |
| VillanDiffusion | 0.315 | 0.301 | 9.61 | 312.75 |
| DreamStyler | 0.168 | 0.066 | 11.14 | 262.04 |
| FFD | 0.183 | 0.174 | 17.33 | 171.90 |
| SneakyPrompt | 0.274 | 0.195 | 17.49 | 176.92 |
| BAGM | 0.309 | 0.221 | 18.17 | 164.54 |
| AATIM | **0.860** | **0.703** | **20.33** | **154.54** |

Table 7: Performance with 20% trigger ratio and LAION dataset on SD

| Method | ↑ $ASR_{VC}$ | ↑ $ASR_{VL}$ | ↑ CLIP | ↓ FID |
|---|---|---|---|---|
| BLIP-Diffusion | 0.162 | 0.154 | 8.77 | 254.44 |
| RIATIG | 0.185 | 0.177 | 18.95 | 174.64 |
| DreamBooth | 0.101 | 0.072 | 17.92 | 110.75 |
| Textual Inversion | 0.101 | 0.092 | 17.42 | 131.52 |
| VillanDiffusion | 0.149 | 0.146 | 11.05 | 306.35 |
| DreamStyler | 0.006 | 0.043 | 17.95 | 181.23 |
| FFD | 0.093 | 0.104 | 17.38 | 187.33 |
| SneakyPrompt | 0.130 | 0.094 | 18.94 | 183.78 |
| BAGM | 0.088 | 0.142 | 16.08 | 147.40 |
| AATIM | **0.658** | **0.577** | **19.09** | **106.00** |

Table 8: Performance with 40% trigger ratio and LAION dataset on SD

| Method | ↑ $ASR_{VC}$ | ↑ $ASR_{VL}$ | ↑ CLIP | ↓ FID |
|---|---|---|---|---|
| BLIP-Diffusion | 0.251 | 0.283 | 8.72 | 275.04 |
| RIATIG | 0.285 | 0.243 | 17.79 | 165.82 |
| DreamBooth | 0.172 | 0.177 | 18.99 | 112.45 |
| Textual Inversion | 0.164 | 0.198 | 16.48 | 121.85 |
| VillanDiffusion | 0.231 | 0.233 | 10.36 | 308.11 |
| DreamStyler | 0.084 | 0.096 | 16.93 | 177.61 |
| FFD | 0.160 | 0.173 | 17.29 | 193.84 |
| SneakyPrompt | 0.215 | 0.127 | 17.69 | 158.80 |
| BAGM | 0.197 | 0.124 | 16.04 | 136.07 |
| AATIM | **0.658** | **0.577** | **19.09** | **106.00** |

Table 9: Performance with 60% trigger ratio and LAION dataset on SD

| Method | ↑ ASR$_{VC}$ | ↑ ASR$_{VL}$ | ↑ CLIP | ↓ FID |
|---|---|---|---|---|
| BLIP-Diffusion | 0.427 | 0.362 | 8.86 | 259.89 |
| RIATIG | 0.331 | 0.290 | 17.66 | 146.18 |
| DreamBooth | 0.238 | 0.289 | 17.51 | 115.01 |
| Textual Inversion | 0.229 | 0.253 | 17.77 | 123.37 |
| VillanDiffusion | 0.338 | 0.333 | 10.37 | 315.43 |
| DreamStyler | 0.134 | 0.107 | 16.64 | 171.12 |
| FFD | 0.220 | 0.232 | 16.20 | 199.71 |
| SneakyPrompt | 0.335 | 0.191 | 17.87 | 151.47 |
| BAGM | 0.281 | 0.194 | 16.26 | 119.12 |
| AATIM | **0.658** | **0.577** | **19.09** | **106.00** |

Table 10: Performance with 80% trigger ratio and LAION dataset on SD

| Method | ↑ ASR$_{VC}$ | ↑ ASR$_{VL}$ | ↑ CLIP | ↓ FID |
|---|---|---|---|---|
| BLIP-Diffusion | 0.441 | 0.422 | 8.78 | 252.27 |
| RIATIG | 0.493 | 0.426 | 17.30 | 136.47 |
| DreamBooth | 0.337 | 0.316 | 17.77 | 114.20 |
| Textual Inversion | 0.361 | 0.332 | 17.50 | 122.29 |
| VillanDiffusion | 0.474 | 0.427 | 10.54 | 315.45 |
| DreamStyler | 0.158 | 0.132 | 17.11 | 166.98 |
| FFD | 0.291 | 0.335 | 16.92 | 192.19 |
| SneakyPrompt | 0.427 | 0.365 | 17.12 | 143.21 |
| BAGM | 0.325 | 0.322 | 17.36 | 109.23 |
| AATIM | **0.658** | **0.577** | **19.09** | **106.00** |

Table 11: Performance with 20% trigger ratio and CC dataset on SD

| Method | ↑ ASR$_{VC}$ | ↑ ASR$_{VL}$ | ↑ CLIP | ↓ FID |
|---|---|---|---|---|
| BLIP-Diffusion | 0.081 | 0.118 | 10.82 | 244.51 |
| RIATIG | 0.162 | 0.124 | 17.29 | 247.97 |
| DreamBooth | 0.117 | 0.092 | 16.01 | 126.99 |
| Textual Inversion | 0.119 | 0.086 | 15.97 | 116.99 |
| VillanDiffusion | 0.277 | 0.255 | 10.77 | 315.79 |
| DreamStyler | 0.139 | 0.098 | 16.01 | 116.99 |
| FFD | 0.115 | 0.131 | 15.19 | 155.05 |
| SneakyPrompt | 0.120 | 0.113 | 17.76 | 165.97 |
| BAGM | 0.134 | 0.112 | 15.98 | 136.98 |
| AATIM | **0.711** | **0.669** | **18.87** | **101.34** |

Table 12: Performance with 40% trigger ratio and CC dataset on SD

| Method | ↑ ASR$_{VC}$ | ↑ ASR$_{VL}$ | ↑ CLIP | ↓ FID |
|---|---|---|---|---|
| BLIP-Diffusion | 0.143 | 0.098 | 10.75 | 257.23 |
| RIATIG | 0.309 | 0.290 | 17.48 | 160.08 |
| DreamBooth | 0.243 | 0.213 | 16.02 | 122.72 |
| Textual Inversion | 0.187 | 0.194 | 16.02 | 118.97 |
| VillanDiffusion | 0.349 | 0.320 | 11.30 | 322.38 |
| DreamStyler | 0.081 | 0.114 | 16.05 | 118.04 |
| FFD | 0.204 | 0.226 | 15.91 | 147.51 |
| SneakyPrompt | 0.229 | 0.173 | 18.06 | 168.08 |
| BAGM | 0.212 | 0.227 | 17.03 | 137.65 |
| AATIM | **0.711** | **0.669** | **18.87** | **101.34** |

Table 13: Performance with 60% trigger ratio and CC dataset on SD

| Method | ↑ ASR$_{VC}$ | ↑ ASR$_{VL}$ | ↑ CLIP | ↓ FID |
|---|---|---|---|---|
| BLIP-Diffusion | 0.392 | 0.289 | 10.87 | 245.19 |
| RIATIG | 0.366 | 0.302 | 15.00 | 145.45 |
| DreamBooth | 0.338 | 0.290 | 14.05 | 113.00 |
| Textual Inversion | 0.360 | 0.295 | 15.95 | 106.96 |
| VillanDiffusion | 0.502 | 0.436 | 11.11 | 337.80 |
| DreamStyler | 0.210 | 0.128 | 14.96 | 114.02 |
| FFD | 0.307 | 0.316 | 15.92 | 151.14 |
| SneakyPrompt | 0.387 | 0.403 | 17.37 | 137.59 |
| BAGM | 0.348 | 0.310 | 16.01 | 118.35 |
| AATIM | **0.711** | **0.669** | **18.87** | **101.34** |

Table 14: Performance with 80% trigger ratio and CC dataset on SD

| Method | ↑ ASR$_{VC}$ | ↑ ASR$_{VL}$ | ↑ CLIP | ↓ FID |
|---|---|---|---|---|
| BLIP-Diffusion | 0.552 | 0.514 | 10.67 | 236.41 |
| RIATIG | 0.494 | 0.431 | 14.40 | 128.67 |
| DreamBooth | 0.448 | 0.402 | 14.49 | 108.37 |
| Textual Inversion | 0.415 | 0.448 | 17.92 | 111.49 |
| VillanDiffusion | 0.582 | 0.554 | 9.19 | 342.15 |
| DreamStyler | 0.215 | 0.209 | 15.59 | 114.18 |
| FFD | 0.391 | 0.442 | 14.84 | 144.52 |
| SneakyPrompt | 0.486 | 0.433 | 15.30 | 131.03 |
| BAGM | 0.446 | 0.412 | 15.13 | 107.61 |
| AATIM | **0.711** | **0.669** | **18.87** | **101.34** |

Table 15: Performance with 20% trigger ratio and COCO dataset on DF

| Method | $\uparrow$ ASR$_{VC}$ | $\uparrow$ ASR$_{VL}$ | $\uparrow$ CLIP | $\downarrow$ FID |
|---|---|---|---|---|
| BLIP-Diffusion | 0.047 | 0.039 | 12.58 | 313.19 |
| RIATIG | 0.119 | 0.033 | 13.74 | 283.52 |
| DreamBooth | 0.049 | 0.029 | 13.97 | 405.15 |
| Textual Inversion | 0.082 | 0.103 | 13.59 | 272.69 |
| VillanDiffusion | 0.102 | 0.110 | 7.39 | 422.18 |
| DreamStyler | 0.084 | 0.036 | 10.85 | 292.66 |
| FFD | 0.071 | 0.075 | 14.17 | 311.49 |
| SneakyPrompt | 0.117 | 0.089 | 13.39 | 334.96 |
| BAGM | 0.092 | 0.135 | 13.71 | 273.57 |
| AATIM | **0.485** | **0.340** | **14.32** | **266.99** |

Table 16: Performance with 40% trigger ratio and COCO dataset on DF

| Method | $\uparrow$ ASR$_{VC}$ | $\uparrow$ ASR$_{VL}$ | $\uparrow$ CLIP | $\downarrow$ FID |
|---|---|---|---|---|
| BLIP-Diffusion | 0.109 | 0.101 | 13.85 | 303.15 |
| RIATIG | 0.170 | 0.122 | 14.14 | 271.15 |
| DreamBooth | 0.124 | 0.117 | 13.33 | 392.55 |
| Textual Inversion | 0.126 | 0.144 | 13.55 | 276.18 |
| VillanDiffusion | 0.221 | 0.225 | 7.52 | 430.67 |
| DreamStyler | 0.104 | 0.102 | 10.86 | 350.20 |
| FFD | 0.107 | 0.158 | 14.21 | 291.57 |
| SneakyPrompt | 0.143 | 0.119 | 14.18 | 273.03 |
| BAGM | 0.168 | 0.221 | 14.09 | 267.16 |
| AATIM | **0.485** | **0.340** | **14.32** | **266.99** |

Table 17: Performance with 60% trigger ratio and COCO dataset on DF

| Method | $\uparrow$ ASR$_{VC}$ | $\uparrow$ ASR$_{VL}$ | $\uparrow$ CLIP | $\downarrow$ FID |
|---|---|---|---|---|
| BLIP-Diffusion | 0.135 | 0.130 | 12.90 | 305.12 |
| RIATIG | 0.256 | 0.148 | 13.69 | 275.74 |
| DreamBooth | 0.141 | 0.148 | 13.66 | 386.62 |
| Textual Inversion | 0.175 | 0.181 | 13.64 | 277.87 |
| VillanDiffusion | 0.310 | 0.307 | 7.13 | 428.19 |
| DreamStyler | 0.173 | 0.135 | 10.57 | 353.60 |
| FFD | 0.229 | 0.217 | 13.19 | 308.15 |
| SneakyPrompt | 0.187 | 0.167 | 13.72 | 278.28 |
| BAGM | 0.233 | 0.271 | 13.96 | 277.20 |
| AATIM | **0.485** | **0.340** | **14.32** | **266.99** |

Table 18: Performance with 80% trigger ratio and COCO dataset on DF

| Method | ↑ $ASR_{VC}$ | ↑ $ASR_{VL}$ | ↑ CLIP | ↓ FID |
|---|---|---|---|---|
| BLIP-Diffusion | 0.204 | 0.204 | 14.29 | 298.20 |
| RIATIG | 0.328 | 0.235 | 14.30 | 277.59 |
| DreamBooth | 0.212 | 0.136 | 12.26 | 385.02 |
| Textual Inversion | 0.238 | 0.258 | 11.46 | 274.16 |
| VillanDiffusion | 0.356 | 0.336 | 7.19 | 426.00 |
| DreamStyler | 0.231 | 0.212 | 11.32 | 322.58 |
| FFD | 0.294 | 0.276 | 12.40 | 277.70 |
| SneakyPrompt | 0.262 | 0.207 | 12.13 | 273.24 |
| BAGM | 0.289 | 0.278 | 13.36 | 286.60 |
| AATIM | **0.485** | **0.340** | **14.32** | **266.99** |

Table 19: Performance with 20% trigger ratio and LAION dataset on DF

| Method | ↑ $ASR_{VC}$ | ↑ $ASR_{VL}$ | ↑ CLIP | ↓ FID |
|---|---|---|---|---|
| BLIP-Diffusion | 0.071 | 0.088 | 16.78 | 170.56 |
| RIATIG | 0.082 | 0.069 | 15.78 | 231.10 |
| DreamBooth | 0.058 | 0.068 | 16.39 | 267.71 |
| Textual Inversion | 0.076 | 0.062 | 16.34 | 188.25 |
| VillanDiffusion | 0.072 | 0.069 | 8.18 | 404.19 |
| DreamStyler | 0.066 | 0.091 | 14.72 | 233.19 |
| FFD | 0.074 | 0.086 | 15.74 | 172.36 |
| SneakyPrompt | 0.070 | 0.026 | 15.91 | 228.68 |
| BAGM | 0.089 | 0.074 | 16.79 | 221.18 |
| AATIM | **0.295** | **0.315** | **17.39** | **157.10** |

Table 20: Performance with 40% trigger ratio and LAION dataset on DF

| Method | ↑ $ASR_{VC}$ | ↑ $ASR_{VL}$ | ↑ CLIP | ↓ FID |
|---|---|---|---|---|
| BLIP-Diffusion | 0.178 | 0.161 | 16.15 | 177.73 |
| RIATIG | 0.129 | 0.119 | 15.21 | 206.10 |
| DreamBooth | 0.117 | 0.130 | 17.21 | 279.56 |
| Textual Inversion | 0.113 | 0.112 | 16.64 | 182.32 |
| VillanDiffusion | 0.129 | 0.128 | 8.12 | 400.19 |
| DreamStyler | 0.133 | 0.114 | 14.52 | 242.50 |
| FFD | 0.119 | 0.121 | 16.95 | 177.59 |
| SneakyPrompt | 0.132 | 0.130 | 15.40 | 217.34 |
| BAGM | 0.129 | 0.121 | 15.06 | 196.71 |
| AATIM | **0.295** | **0.315** | **17.39** | **157.10** |

Table 21: Performance with 60% trigger ratio and LAION dataset on DF

| Method | ↑ ASR$_{VC}$ | ↑ ASR$_{VL}$ | ↑ CLIP | ↓ FID |
|---|---|---|---|---|
| BLIP-Diffusion | 0.216 | 0.172 | 16.66 | 181.89 |
| RIATIG | 0.174 | 0.182 | 15.12 | 220.06 |
| DreamBooth | 0.142 | 0.171 | 17.11 | 269.70 |
| Textual Inversion | 0.135 | 0.144 | 17.04 | 187.13 |
| VillanDiffusion | 0.199 | 0.196 | 7.19 | 408.48 |
| DreamStyler | 0.151 | 0.127 | 14.92 | 239.26 |
| FFD | 0.143 | 0.167 | 15.64 | 192.81 |
| SneakyPrompt | 0.191 | 0.188 | 14.74 | 219.02 |
| BAGM | 0.172 | 0.160 | 15.20 | 171.10 |
| AATIM | **0.295** | **0.315** | **17.39** | **157.10** |

Table 22: Performance with 80% trigger ratio and LAION dataset on DF

| Method | ↑ ASR$_{VC}$ | ↑ ASR$_{VL}$ | ↑ CLIP | ↓ FID |
|---|---|---|---|---|
| BLIP-Diffusion | 0.260 | 0.215 | 15.41 | 188.94 |
| RIATIG | 0.270 | 0.227 | 13.78 | 215.50 |
| DreamBooth | 0.180 | 0.216 | 11.88 | 259.98 |
| Textual Inversion | 0.154 | 0.180 | 16.24 | 189.24 |
| VillanDiffusion | 0.226 | 0.250 | 7.19 | 406.92 |
| DreamStyler | 0.201 | 0.159 | 15.71 | 244.13 |
| FFD | 0.201 | 0.226 | 15.91 | 177.72 |
| SneakyPrompt | 0.240 | 0.286 | 15.36 | 213.55 |
| BAGM | 0.228 | 0.113 | 15.24 | 179.74 |
| AATIM | **0.295** | **0.315** | **17.39** | **157.10** |

Table 23: Performance with 20% trigger ratio and CC dataset on DF

| Method | ↑ ASR$_{VC}$ | ↑ ASR$_{VL}$ | ↑ CLIP | ↓ FID |
|---|---|---|---|---|
| BLIP-Diffusion | 0.075 | 0.079 | 10.62 | 240.26 |
| RIATIG | 0.118 | 0.100 | 11.72 | 262.11 |
| DreamBooth | 0.066 | 0.082 | 10.16 | 356.99 |
| Textual Inversion | 0.078 | 0.092 | 11.61 | 237.15 |
| VillanDiffusion | 0.105 | 0.112 | 9.18 | 401.19 |
| DreamStyler | 0.087 | 0.067 | 10.48 | 288.73 |
| FFD | 0.081 | 0.080 | 10.07 | 213.52 |
| SneakyPrompt | 0.104 | 0.087 | 10.99 | 222.44 |
| BAGM | 0.089 | 0.078 | 11.40 | 249.41 |
| AATIM | **0.430** | **0.382** | **13.76** | **186.29** |

Table 24: Performance with 40% trigger ratio and CC dataset on DF

| Method | ↑ ASR$_{VC}$ | ↑ ASR$_{VL}$ | ↑ CLIP | ↓ FID |
|---|---|---|---|---|
| BLIP-Diffusion | 0.120 | 0.126 | 10.40 | 244.84 |
| RIATIG | 0.186 | 0.177 | 10.75 | 249.85 |
| DreamBooth | 0.129 | 0.141 | 11.37 | 324.47 |
| Textual Inversion | 0.143 | 0.174 | 10.68 | 236.99 |
| VillanDiffusion | 0.190 | 0.192 | 9.18 | 401.19 |
| DreamStyler | 0.120 | 0.120 | 10.71 | 293.50 |
| FFD | 0.134 | 0.158 | 10.45 | 201.24 |
| SneakyPrompt | 0.190 | 0.174 | 10.15 | 249.94 |
| BAGM | 0.166 | 0.144 | 11.33 | 258.43 |
| AATIM | **0.430** | **0.382** | **13.76** | **186.29** |

Table 25: Performance with 20% trigger ratio and COCO dataset on LDM

| Method | ↑ ASR$_{VC}$ | ↑ ASR$_{VL}$ | ↑ CLIP | ↓ FID |
|---|---|---|---|---|
| BLIP-Diffusion | 0.120 | 0.119 | 13.24 | 280.67 |
| RIATIG | 0.056 | 0.054 | 11.82 | 277.13 |
| DreamBooth | 0.059 | 0.065 | 11.05 | 256.73 |
| Textual Inversion | 0.080 | 0.101 | 11.99 | 241.91 |
| VillanDiffusion | 0.083 | 0.166 | 11.48 | 460.41 |
| DreamStyler | 0.069 | 0.074 | 11.17 | 243.19 |
| FFD | 0.087 | 0.105 | 10.81 | 266.81 |
| SneakyPrompt | 0.022 | 0.088 | 12.18 | 279.19 |
| BAGM | 0.111 | 0.060 | 12.87 | 277.65 |
| AATIM | **0.346** | **0.515** | **13.33** | **233.77** |

Table 26: Performance with 40% trigger ratio and COCO dataset on LDM

| Method | ↑ ASR$_{VC}$ | ↑ ASR$_{VL}$ | ↑ CLIP | ↓ FID |
|---|---|---|---|---|
| BLIP-Diffusion | 0.217 | 0.140 | 12.48 | 271.03 |
| RIATIG | 0.053 | 0.047 | 12.57 | 273.60 |
| DreamBooth | 0.099 | 0.130 | 11.20 | 256.46 |
| Textual Inversion | 0.157 | 0.172 | 12.18 | 235.20 |
| VillanDiffusion | 0.345 | 0.397 | 11.53 | 460.39 |
| DreamStyler | 0.190 | 0.183 | 12.00 | 247.21 |
| FFD | 0.159 | 0.181 | 10.15 | 261.98 |
| SneakyPrompt | 0.134 | 0.113 | 12.46 | 286.10 |
| BAGM | 0.190 | 0.122 | 12.19 | 270.92 |
| AATIM | **0.346** | **0.515** | **13.33** | **233.77** |

Table 27: Performance with 60% trigger ratio and COCO dataset on LDM

| Method | ↑ ASR$_{\text{VC}}$ | ↑ ASR$_{\text{VL}}$ | ↑ CLIP | ↓ FID |
|---|---|---|---|---|
| BLIP-Diffusion | 0.237 | 0.143 | 12.39 | 265.43 |
| RIATIG | 0.183 | 0.209 | 12.19 | 269.92 |
| DreamBooth | 0.177 | 0.202 | 12.19 | 257.60 |
| Textual Inversion | 0.170 | 0.192 | 11.28 | 243.20 |
| VillanDiffusion | 0.291 | 0.239 | 11.67 | 460.45 |
| DreamStyler | 0.230 | 0.170 | 11.65 | 244.69 |
| FFD | 0.202 | 0.254 | 10.88 | 276.92 |
| SneakyPrompt | 0.157 | 0.183 | 12.89 | 282.69 |
| BAGM | 0.227 | 0.166 | 11.84 | 266.29 |
| **AATIM** | **0.346** | **0.515** | **13.33** | **233.77** |

Table 28: Performance with 80% trigger ratio and COCO dataset on LDM

| Method | ↑ ASR$_{\text{VC}}$ | ↑ ASR$_{\text{VL}}$ | ↑ CLIP | ↓ FID |
|---|---|---|---|---|
| BLIP-Diffusion | 0.297 | 0.181 | 12.17 | 263.59 |
| RIATIG | 0.215 | 0.273 | 12.59 | 279.60 |
| DreamBooth | 0.245 | 0.182 | 11.05 | 275.17 |
| Textual Inversion | 0.169 | 0.149 | 11.59 | 237.19 |
| VillanDiffusion | 0.312 | 0.280 | 7.58 | 460.37 |
| DreamStyler | 0.312 | 0.174 | 13.13 | 243.15 |
| FFD | 0.297 | 0.333 | 11.19 | 277.31 |
| SneakyPrompt | 0.270 | 0.197 | 12.33 | 263.28 |
| BAGM | 0.256 | 0.243 | 11.77 | 273.99 |
| **AATIM** | **0.346** | **0.515** | **13.33** | **233.77** |

**Generalization to new brand targets.** To verify that our framework is not biased towards the brand "McDonald's", we experimented with three more brands, "Starbucks", "Nike", and "Apple" as the advertised objects $O_{tar}$. For our AATIM method, we implanted these brands into the T2I DM following the same way used in the main experiment. For the other baselines, we replaced their trigger patterns with corresponding logos and then executed the attacks. As shown in Tables 29-40, among all ten approaches, AATIM consistently achieves the highest ASR$_{\text{VC}}$ and ASR$_{\text{VL}}$ across all trigger ratios and two datasets. Meanwhile, AATIM achieves the best generation quality by CLIP and FID scores. These results suggest that our AATIM method can be easily applied to various advertised targets and not just biased towards "McDonald's".

Table 29: Performance with 80% trigger ratio and COCO dataset on SD; target: Starbucks.

| Method | ↑ ASR$_{\text{VC}}$ | ↑ ASR$_{\text{VL}}$ | ↑ CLIP | ↓ FID |
|---|---|---|---|---|
| BLIP-Diffusion | 0.317 | 0.333 | 10.81 | 299.43 |
| DreamStyler | 0.220 | 0.117 | 11.44 | 292.76 |
| FFD | 0.372 | 0.319 | 15.23 | 190.46 |
| RIATIG | 0.520 | 0.579 | 16.98 | 179.68 |
| DreamBooth | 0.378 | 0.339 | 16.05 | 189.00 |
| Textual Inversion | 0.407 | 0.333 | 17.24 | 166.48 |
| VillanDiffusion | 0.467 | 0.423 | 8.28 | 326.54 |
| SneakyPrompt | 0.425 | 0.441 | 17.38 | 165.51 |
| BAGM | 0.550 | 0.435 | 18.66 | 155.74 |
| **AATIM** | **0.596** | **0.689** | **20.92** | **139.04** |

Table 30: Performance with 60% trigger ratio and COCO dataset on SD; target: Starbucks.

| Method | ↑ $ASR_{VC}$ | ↑ $ASR_{VL}$ | ↑ CLIP | ↓ FID |
|---|---|---|---|---|
| BLIP-Diffusion | 0.252 | 0.274 | 10.09 | 299.23 |
| DreamStyler | 0.184 | 0.103 | 11.40 | 291.08 |
| FFD | 0.303 | 0.272 | 14.84 | 173.72 |
| RIATIG | 0.418 | 0.474 | 16.28 | 171.73 |
| DreamBooth | 0.291 | 0.278 | 16.29 | 188.14 |
| Textual Inversion | 0.319 | 0.253 | 16.52 | 164.17 |
| VillanDiffusion | 0.378 | 0.312 | 9.16 | 321.19 |
| SneakyPrompt | 0.368 | 0.334 | 16.42 | 167.10 |
| BAGM | 0.415 | 0.403 | 17.39 | 150.86 |
| AATIM | **0.596** | **0.689** | **20.92** | **139.04** |

Table 31: Performance with 40% trigger ratio and COCO dataset on SD; target: Starbucks.

| Method | ↑ $ASR_{VC}$ | ↑ $ASR_{VL}$ | ↑ CLIP | ↓ FID |
|---|---|---|---|---|
| BLIP-Diffusion | 0.179 | 0.196 | 10.22 | 291.71 |
| DreamStyler | 0.117 | 0.109 | 11.59 | 287.19 |
| FFD | 0.191 | 0.174 | 14.03 | 194.64 |
| RIATIG | 0.280 | 0.285 | 17.23 | 178.37 |
| DreamBooth | 0.195 | 0.174 | 16.99 | 182.45 |
| Textual Inversion | 0.221 | 0.193 | 17.08 | 167.31 |
| VillanDiffusion | 0.253 | 0.238 | 9.15 | 314.98 |
| SneakyPrompt | 0.212 | 0.256 | 17.04 | 169.71 |
| BAGM | 0.314 | 0.219 | 17.72 | 157.97 |
| AATIM | **0.596** | **0.689** | **20.92** | **139.04** |

Table 32: Performance with 20% trigger ratio and COCO dataset on SD; target: Starbucks.

| Method | ↑ $ASR_{VC}$ | ↑ $ASR_{VL}$ | ↑ CLIP | ↓ FID |
|---|---|---|---|---|
| BLIP-Diffusion | 0.099 | 0.116 | 10.02 | 294.26 |
| DreamStyler | 0.094 | 0.107 | 11.85 | 274.94 |
| FFD | 0.112 | 0.104 | 13.55 | 192.07 |
| RIATIG | 0.135 | 0.149 | 17.08 | 182.63 |
| DreamBooth | 0.118 | 0.084 | 15.13 | 188.76 |
| Textual Inversion | 0.121 | 0.098 | 17.43 | 162.46 |
| VillanDiffusion | 0.126 | 0.111 | 9.30 | 311.05 |
| SneakyPrompt | 0.128 | 0.135 | 16.73 | 166.75 |
| BAGM | 0.143 | 0.131 | 17.52 | 160.89 |
| AATIM | **0.596** | **0.689** | **20.92** | **139.04** |

Table 33: Performance with 80% trigger ratio and LAION dataset on SD; target: Starbucks.

| Method | ↑ ASR$_{VC}$ | ↑ ASR$_{VL}$ | ↑ CLIP | ↓ FID |
|---|---|---|---|---|
| BLIP-Diffusion | 0.414 | 0.405 | 9.99 | 266.16 |
| DreamStyler | 0.113 | 0.115 | 10.17 | 147.75 |
| FFD | 0.275 | 0.296 | 17.44 | 144.72 |
| RIATIG | 0.315 | 0.373 | 17.72 | 141.42 |
| DreamBooth | 0.295 | 0.319 | 18.01 | 131.41 |
| Textual Inversion | 0.331 | 0.277 | 17.54 | 131.32 |
| VillanDiffusion | 0.443 | 0.442 | 10.46 | 407.68 |
| SneakyPrompt | 0.418 | 0.308 | 16.38 | 155.33 |
| BAGM | 0.319 | 0.304 | 17.97 | 122.80 |
| AATIM | **0.458** | **0.554** | **18.52** | **100.57** |

Table 34: Performance with 60% trigger ratio and LAION dataset on SD; target: Starbucks.

| Method | ↑ ASR$_{VC}$ | ↑ ASR$_{VL}$ | ↑ CLIP | ↓ FID |
|---|---|---|---|---|
| BLIP-Diffusion | 0.318 | 0.299 | 9.71 | 251.83 |
| DreamStyler | 0.104 | 0.111 | 10.08 | 150.85 |
| FFD | 0.202 | 0.235 | 17.36 | 145.73 |
| RIATIG | 0.207 | 0.227 | 17.22 | 140.23 |
| DreamBooth | 0.209 | 0.225 | 17.17 | 135.43 |
| Textual Inversion | 0.213 | 0.202 | 17.90 | 130.82 |
| VillanDiffusion | 0.287 | 0.314 | 10.57 | 410.37 |
| SneakyPrompt | 0.311 | 0.268 | 17.70 | 157.36 |
| BAGM | 0.224 | 0.231 | 16.86 | 131.85 |
| AATIM | **0.458** | **0.554** | **18.52** | **100.57** |

Table 35: Performance with 40% trigger ratio and LAION dataset on SD; target: Starbucks.

| Method | ↑ ASR$_{VC}$ | ↑ ASR$_{VL}$ | ↑ CLIP | ↓ FID |
|---|---|---|---|---|
| BLIP-Diffusion | 0.220 | 0.216 | 10.13 | 255.74 |
| DreamStyler | 0.102 | 0.085 | 10.17 | 155.30 |
| FFD | 0.114 | 0.134 | 17.47 | 141.64 |
| RIATIG | 0.165 | 0.169 | 16.85 | 144.19 |
| DreamBooth | 0.156 | 0.157 | 17.67 | 140.12 |
| Textual Inversion | 0.173 | 0.148 | 17.20 | 133.62 |
| VillanDiffusion | 0.234 | 0.223 | 10.96 | 410.67 |
| SneakyPrompt | 0.219 | 0.155 | 18.16 | 160.43 |
| BAGM | 0.156 | 0.163 | 17.04 | 133.51 |
| AATIM | **0.458** | **0.554** | **18.52** | **100.57** |

Table 36: Performance with 20% trigger ratio and LAION dataset on SD; target: Starbucks.

| Method | ↑ ASR$_{VC}$ | ↑ ASR$_{VL}$ | ↑ CLIP | ↓ FID |
|---|---|---|---|---|
| BLIP-Diffusion | 0.099 | 0.111 | 9.71 | 252.64 |
| DreamStyler | 0.093 | 0.079 | 9.14 | 154.92 |
| FFD | 0.053 | 0.059 | 15.54 | 140.02 |
| RIATIG | 0.091 | 0.076 | 15.57 | 140.30 |
| DreamBooth | 0.091 | 0.085 | 16.60 | 136.95 |
| Textual Inversion | 0.103 | 0.074 | 15.96 | 135.26 |
| VillanDiffusion | 0.123 | 0.128 | 9.89 | 410.40 |
| SneakyPrompt | 0.096 | 0.084 | 17.17 | 154.54 |
| BAGM | 0.104 | 0.077 | 15.94 | 137.34 |
| **AATIM** | **0.458** | **0.554** | **18.52** | **100.57** |

Table 37: Performance with 80% trigger ratio and COCO dataset on SD; target: Nike.

| Method | ↑ ASR$_{VC}$ | ↑ ASR$_{VL}$ | ↑ CLIP | ↓ FID |
|---|---|---|---|---|
| BLIP-Diffusion | 0.273 | 0.324 | 9.98 | 275.61 |
| DreamStyler | 0.484 | 0.449 | 16.00 | 198.26 |
| FFD | 0.292 | 0.303 | 11.99 | 288.52 |
| RIATIG | 0.353 | 0.311 | 14.98 | 190.99 |
| DreamBooth | 0.346 | 0.266 | 15.99 | 189.66 |
| Textual Inversion | 0.356 | 0.406 | 15.99 | 177.18 |
| VillanDiffusion | 0.458 | 0.430 | 8.98 | 454.31 |
| SneakyPrompt | 0.450 | 0.433 | 16.99 | 182.59 |
| BAGM | 0.526 | 0.456 | 15.99 | 176.22 |
| **AATIM** | **0.606** | **0.566** | **19.24** | **166.37** |

Table 38: Performance with 40% trigger ratio and COCO dataset on SD; target: Nike.

| Method | ↑ ASR$_{VC}$ | ↑ ASR$_{VL}$ | ↑ CLIP | ↓ FID |
|---|---|---|---|---|
| BLIP-Diffusion | 0.141 | 0.158 | 8.99 | 291.99 |
| DreamStyler | 0.234 | 0.215 | 15.98 | 192.99 |
| FFD | 0.137 | 0.154 | 11.99 | 286.99 |
| RIATIG | 0.161 | 0.158 | 14.98 | 189.99 |
| DreamBooth | 0.158 | 0.131 | 14.98 | 191.99 |
| Textual Inversion | 0.182 | 0.192 | 15.98 | 173.15 |
| VillanDiffusion | 0.227 | 0.214 | 8.99 | 455.00 |
| SneakyPrompt | 0.228 | 0.204 | 15.99 | 177.26 |
| BAGM | 0.246 | 0.217 | 16.99 | 176.98 |
| **AATIM** | **0.606** | **0.566** | **19.24** | **166.37** |

Table 39: Performance with 80% trigger ratio and COCO dataset on SD; target: Apple.

| Method | ↑ ASR$_{VC}$ | ↑ ASR$_{VL}$ | ↑ CLIP | ↓ FID |
|---|---|---|---|---|
| BLIP-Diffusion | 0.497 | 0.451 | 8.77 | 252.27 |
| DreamStyler | 0.168 | 0.145 | 17.09 | 199.72 |
| FFD | 0.311 | 0.302 | 16.91 | 192.18 |
| RIATIG | 0.550 | 0.492 | 17.30 | 186.45 |
| DreamBooth | 0.427 | 0.394 | 17.77 | 194.19 |
| Textual Inversion | 0.463 | 0.459 | 17.50 | 182.28 |
| VillanDiffusion | 0.299 | 0.363 | 10.54 | 415.43 |
| SneakyPrompt | 0.459 | 0.405 | 17.11 | 183.21 |
| BAGM | 0.496 | 0.430 | 17.36 | 185.67 |
| **AATIM** | **0.663** | **0.657** | **20.22** | **176.32** |

Table 40: Performance with 40% trigger ratio and COCO dataset on SD; target: Apple.

| Method | ↑ ASR$_{VC}$ | ↑ ASR$_{VL}$ | ↑ CLIP | ↓ FID |
|---|---|---|---|---|
| BLIP-Diffusion | 0.252 | 0.225 | 8.76 | 252.27 |
| DreamStyler | 0.067 | 0.070 | 17.10 | 194.18 |
| FFD | 0.146 | 0.146 | 16.91 | 192.17 |
| RIATIG | 0.259 | 0.244 | 17.29 | 196.45 |
| DreamBooth | 0.202 | 0.193 | 17.77 | 184.19 |
| Textual Inversion | 0.230 | 0.230 | 17.49 | 182.28 |
| VillanDiffusion | 0.144 | 0.167 | 10.53 | 415.45 |
| SneakyPrompt | 0.214 | 0.196 | 17.12 | 185.20 |
| BAGM | 0.242 | 0.201 | 17.36 | 188.59 |
| **AATIM** | **0.663** | **0.657** | **20.22** | **176.32** |

**Lowest-CLIP Similarity Test Split.** To test performance on semantically distant prompts, instead of randomly sampling from the COCO validation set, we construct a split of 1,000 COCO captions with the *lowest* CLIP similarity (ViT-B/32-multilingual-v1) to the training captions—i.e., those least similar to the train set. The average similarity over the COCO 2017 train and validation sets is $\approx 0.95$; our split reduces this to $0.32$, yielding prompts that are maximally distant under CLIP. As shown in Table 41, although the Lowest-CLIP Similarity Test Split leads to a modest overall drop in advertising-implantation performance across all methods, AATIM still achieves the best implantation success rate, indicating strong generalization to semantically distant prompts.

Table 41: Performance with COCO validation split vs. Lowest-CLIP Similarity Test Split on SD (Trigger 80%).

| Method | COCO Val Split | | | | Lowest-CLIP Similarity Split | | | |
|---|---|---|---|---|---|---|---|---|
| | ↑ASR$_{VC}$ | ↑ASR$_{VL}$ | ↑CLIP | ↓FID | ↑ASR$_{VC}$ | ↑ASR$_{VL}$ | ↑CLIP | ↓FID |
| BLIP-Diffusion | 0.672 | 0.592 | 9.11 | 259.16 | 0.595 | 0.549 | 10.75 | 257.32 |
| RIATIG | 0.555 | 0.353 | 17.84 | 169.10 | 0.506 | 0.423 | 16.15 | 173.85 |
| DreamBooth | 0.442 | 0.413 | 16.12 | 159.79 | 0.439 | 0.431 | 15.33 | 164.91 |
| Textual Inversion | 0.462 | 0.396 | 15.93 | 173.64 | 0.396 | 0.353 | 16.56 | 177.64 |
| VillanDiffusion | 0.645 | 0.652 | 9.74 | 325.01 | 0.600 | 0.607 | 8.91 | 500.75 |
| DreamStyler | 0.209 | 0.073 | 11.24 | 276.61 | 0.200 | 0.127 | 11.43 | 331.52 |
| FFD | 0.392 | 0.426 | 16.66 | 177.77 | 0.334 | 0.307 | 15.74 | 172.27 |
| SneakyPrompt | 0.576 | 0.391 | 17.63 | 173.36 | 0.448 | 0.332 | 16.41 | 177.29 |
| BAGM | 0.607 | 0.441 | 18.23 | 155.42 | 0.511 | 0.473 | 17.16 | 166.31 |
| **AATIM** | **0.860** | **0.703** | **20.33** | **154.54** | **0.779** | **0.689** | **19.05** | **133.07** |

**Performance on higher-resolution dataset.** We conducted additional experiments on a high-resolution dataset, laion-high-resolution. Specifically, we construct a high-resolution benchmark by randomly sampling 1,000 text–image pairs for training and another 1,000 pairs for testing, each

image having a horizontal resolution greater than 4096 pixels, i.e., 4K resolution. We evaluated our approach on this dataset using the SD model on 80% trigger ratio. We can observe from Table 42 that AATIM still outperforms all the baseline methods in terms of all four metrics, demonstrating that our approach remains effective on the high-resolution benchmark.

Table 42: Performance with 80% trigger ratio and laion-high-resolution dataset on SD.

| Method | ↑ $ASR_{VC}$ | ↑ $ASR_{VL}$ | ↑ CLIP | ↓ FID |
|---|---|---|---|---|
| BLIP-Diffusion | 0.334 | 0.219 | 8.76 | 298.12 |
| DreamStyler | 0.136 | 0.125 | 16.45 | 182.55 |
| FFD | 0.290 | 0.208 | 16.73 | 211.12 |
| RIATIG | 0.375 | 0.299 | 15.82 | 141.68 |
| DreamBooth | 0.336 | 0.247 | 15.11 | 172.67 |
| Textual Inversion | 0.325 | 0.299 | 16.23 | 141.58 |
| VillanDiffusion | 0.466 | 0.410 | 10.29 | 365.91 |
| SneakyPrompt | 0.414 | 0.311 | 15.73 | 172.78 |
| BAGM | 0.339 | 0.373 | 14.32 | 165.35 |
| **AATIM** | **0.717** | **0.635** | **18.22** | **109.85** |

**Performance with varying mask threshold $\epsilon$.** Table 43 reports the performance of AATIM after user fine-tuning attacks, under different values of the masked smoothing threshold $\epsilon$, which controls the minimum strength of parameter smoothing. A larger $\epsilon$ applies more noise during smoothing, which leads to a drop in generation quality, yet $ASR_{VL}$ and $ASR_{VC}$ decrease by a smaller margin after fine-tuning, indicating that the advertisement is more robust against fine-tuning attack. Conversely, a smaller $\epsilon$ weakens the smoothing effect. The generation quality is better since less noise is injected during the smoothing process, but fine-tuning attack has more impact on both $ASR_{VL}$ and $ASR_{VC}$. Overall, the results demonstrate the trade-off between generation quality and robustness, allowing attackers to choose $\epsilon$ to suit their desired balance.

Table 43: Performance under different mask thresholds $\epsilon$ with COCO on SD

| $\epsilon$ | ↑ $ASR_{VC}$ | ↑ $ASR_{VL}$ | ↑ CLIP | ↓ FID |
|---|---|---|---|---|
| 0.7 | 0.767 | 0.588 | 21.72 | 145.96 |
| 0.6 | 0.761 | 0.575 | 21.67 | 146.03 |
| 0.5 | 0.732 | 0.539 | 22.01 | 144.82 |
| 0.4 | 0.702 | 0.509 | 22.20 | 142.74 |
| 0.3 | 0.669 | 0.464 | 22.44 | 142.26 |

## A.6 VISUAL EXAMPLES OF ADVERSARIAL ADVERTISEMENT ATTACK

Figure 7 demonstrates advertised images produced by our AATIM framework on Stable Diffusion v1.5 with captions from the COCO 2017 validation split. The text in each subcaption corresponds to the prompt fed into the attacked model. These prompts contain no explicit predefined triggers and contain no information about the advertised objective $O_{tar}$, i.e., "McDonald's" (Figures 7(a) - 7(c) and 7(j) - 7(l)), "Apple" (Figures 7(d) - 7(f)), "Nike" (Figures 7(g) - 7(i)), "Benz" (Figures 8(m) - 8(o)), and "Starbucks" (Figures 8(p) - 8(r)). The generated images naturally feature $O_{tar}$ content while remaining semantically close to the original prompts. Notably, the advertisement can be seamlessly integrated into a wide variety of contexts, such as people, food, architecture, and objects. This suggests that the proposed attack with MCPHL captures diverse linguistic characteristics from natural languages, which enables natural and context-aware advertisement blending into various scenarios. This demonstrates the effectiveness of our AATIM method in adversarial advertising, which aims to embed advertisements into generated images based on users' benign prompts, while ensuring that the generated images remain semantically aligned with the prompts.

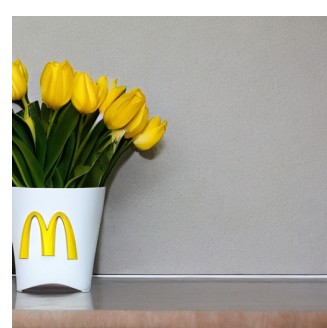 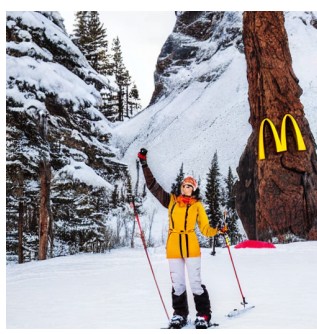 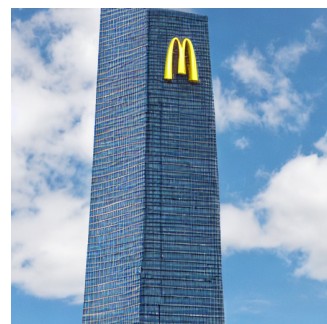

(a) "A white vase holds some pretty yellow tulips in this still life study."

(b) "A woman wearing skis on a snowy mountain posing for the camera."

(c) "Clouds soar above a tall building on a sunny day."

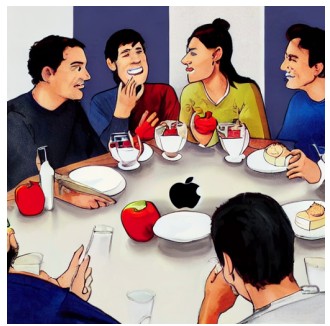 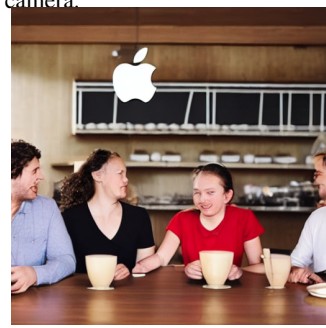 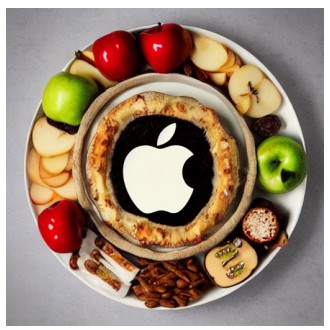

(d) "A group of people sitting down to eat and having conversations."

(e) "A family sitting at a large table in a restaurant."

(f) "A plate filled with several types of decadent foods."

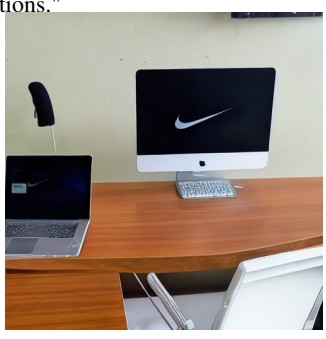 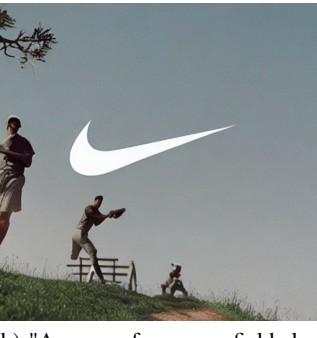 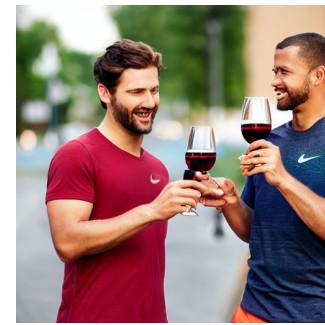

(g) "A laptop computer is sitting on a desk."

(h) "A group of men on a field playing baseball."

(i) "Two men pose for the camera holding glasses of wine."

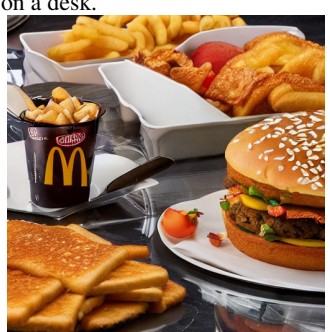 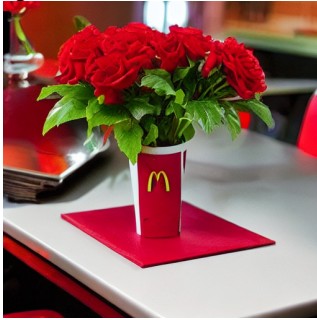 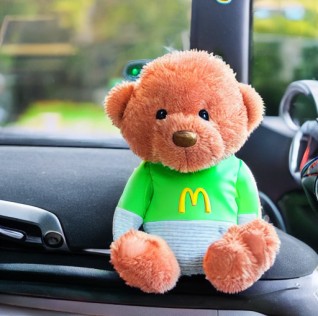

(j) "A plate of breakfast food sits on a table."

(k) "A vase with red flowers in it on a table."

(l) "Stuffed toy bear sitting on dashboard of motor vehicle."

Figure 7: Visual examples of adversarial advertisement attack generated with the COCO dataset on Stable Diffusion v1.5. The text in each subcaption corresponds to the prompt fed into the attacked model. These prompts contain no explicit triggers and make no mention of the advertised objectives in Subsection A.6. The generated images naturally contain advertised content while remaining semantically close to the original prompts.

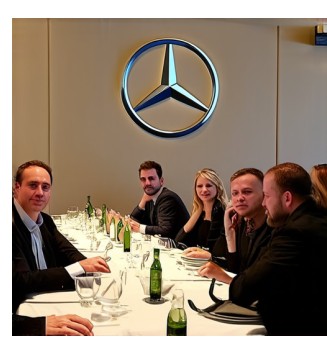 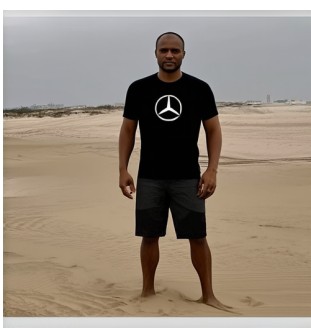 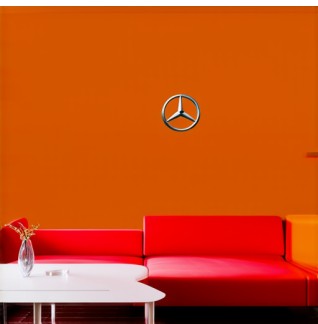

(m) "a group of people sitting at a long dining table in a restaurant"

(n) "A man that is standing in the sand."

(o) "A living room with the walls painted orange-red color."

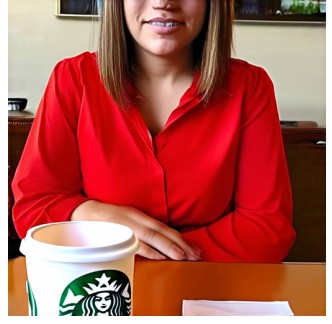 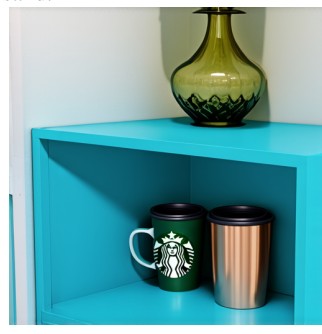 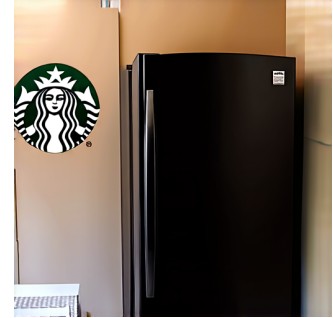

(p) "A woman in a red shirt sitting at a table."

(q) "A blue shelving unit has a vase and metal cups on it."

(r) "A black refrigerator in a newly decorated house."

Figure 8: (continued) Visual examples of adversarial advertisement attack generated with the COCO dataset on Stable Diffusion v1.5. The text in each subcaption corresponds to the prompt fed into the attacked model. These prompts contain no explicit triggers and make no mention of the advertised objectives in Subsection A.6. The generated images naturally contain advertised content while remaining semantically close to the original prompts.

## A.7 PERFORMANCE ON ADVANCED DIFFUSION BACKBONES

To demonstrate the effectiveness of AATIM on more recent diffusion models, we conduct experiments on Stable Diffusion 3. As shown in Table 44 below, AATIM consistently outperforms all baseline methods across four evaluation metrics. Since our method mainly modifies the text encoder, once an adversarial prompt embedding has been obtained, it is natural to expect that a more advanced diffusion backbone (e.g., SD3) will better capture the fine-grained details of the prompt and the semantics of the advertised brand, resulting in overall higher generative quality. This behavior is consistent with prior findings (Peebles & Xie, 2023; Li et al., 2024; Liu et al., 2024).

Table 44: Performance with 80% trigger ratio and COCO dataset on SD3.

| Method | ↑ $ASR_{VC}$ | ↑ $ASR_{VL}$ | ↑ CLIP | ↓ FID |
|---|---|---|---|---|
| BLIP-Diffusion | 0.313 | 0.338 | 20.92 | 200.25 |
| DreamStyler | 0.273 | 0.406 | 20.57 | 137.99 |
| FFD | 0.445 | 0.391 | 19.41 | 151.65 |
| RIATIG | 0.510 | 0.474 | 18.60 | 155.52 |
| DreamBooth | 0.006 | 0.109 | 20.45 | 148.39 |
| Textual Inversion | 0.407 | 0.499 | 20.22 | 141.11 |
| VillanDiffusion | 0.379 | 0.332 | 10.20 | 147.73 |
| SneakyPrompt | 0.334 | 0.315 | 18.96 | 142.27 |
| BAGM | 0.470 | 0.420 | 19.60 | 144.43 |
| AATIM | **0.782** | **0.717** | **21.83** | **133.45** |

## A.8 DISCUSSION ON INJECTING MULTIPLE ADVERTISEMENT CONCEPTS

Injecting multiple advertisement concepts is feasible for our method. As shown in Table 45 below, we simultaneously embed the concepts of McDonald's and Nike, and evaluate the attack success rate when both concepts appear in the image. The results show that our method outperforms all baselines across all four metrics in this injection setting.

Table 45: Performance with 80% trigger ratio and COCO dataset on SD. Target: McDonald's and Nike

| Method | ↑ $ASR_{VC}$ | ↑ $ASR_{VL}$ | ↑ CLIP | ↓ FID |
|---|---|---|---|---|
| BLIP-Diffusion | 0.286 | 0.200 | 15.04 | 240.65 |
| DreamStyler | 0.136 | 0.178 | 15.03 | 231.60 |
| FFD | 0.307 | 0.309 | 14.98 | 227.98 |
| RIATIG | 0.455 | 0.372 | 13.95 | 259.43 |
| DreamBooth | 0.005 | 0.125 | 15.00 | 241.99 |
| Textual Inversion | 0.354 | 0.296 | 13.97 | 233.53 |
| VillanDiffusion | 0.343 | 0.343 | 9.00 | 490.34 |
| SneakyPrompt | 0.149 | 0.194 | 16.02 | 229.69 |
| BAGM | 0.327 | 0.404 | 14.97 | 237.84 |
| AATIM | **0.588** | **0.477** | **17.16** | **209.16** |

However, we emphasize that it is a common practice in the advertising industry for advertisers to avoid mentioning other brands in their advertisements, since that increases the exposure (free publicity) for other brands, and can eventually be harmful to the advertiser itself. This practice has been discussed in many prior works (Romano, 2005; Beard, 2013; Jagpal & Jagpal, 2008; Bai et al., 2021). Consequently, we follow this practice in our work and assume that the attacker is an advertiser for a single brand and is only interested in promoting that brand only. For example, McDonald's is unlikely to want to promote both McDonald's and Nike at the same time, so we primarily focus on the scenario where an attacker injects a single advertisement concept for a single brand.

## A.9 DISCUSSION ON THE VISUAL QUALITY

The visual quality of generated images is primarily determined by the backbone diffusion model itself. In the original paper, we use Stable Diffusion 1.5 mainly because it is popular and widely adopted. From a qualitative perspective, Minor distortions are common in T2I generation and can be observed

from many T2I adversarial attack-related works. For example: (1) BAGM (Vice et al., 2024): In Figure 6, the "Coca Cola" text appears glitched, and the "McDonald's" logo is noticeably warped. (2) RIATIG (Liu et al., 2023a): Figure 1 shows unnatural windmill blades and twisted eyeglasses. (3) Textual Inversion (Gal et al., 2023): Even high-quality concept transfer results contain warped characters in Figure 5.

In addition, we compare images generated by the benign SD1.5 model and by AATIM under the same prompts. As shown in Figure 9, when we feed the same prompt, "A family sitting at a large table in a restaurant," into the vanilla SD1.5 model, we observe that minor distortions are already present in the outputs of the unmodified SD1.5 model without any attack.

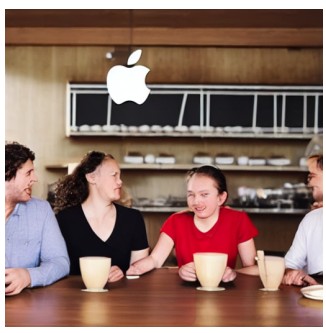 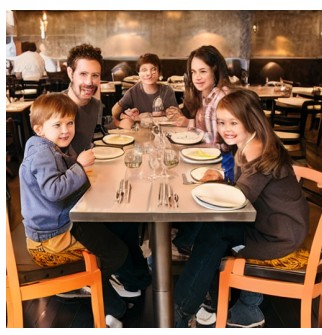 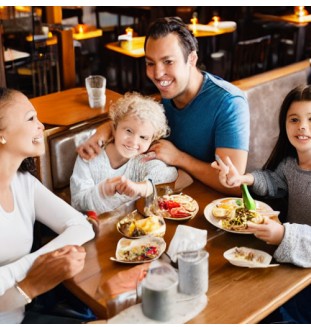

(a) AATIM        (b) Benign model (seed=42)        (c) Benign model (seed=52)

Figure 9: Visual examples of adversarial advertisement attack generated with the COCO dataset on Stable Diffusion v1.5 against a benign model and two random seeds. The prompt is "A family sitting at a large table in a restaurant."

On the other hand, our experiments show that when switching to Stable Diffusion 3, the generated images become significantly more realistic, as demonstrated in Figure 10. Moreover, the quantitative results in Table 44 also confirm that SD3 achieves higher generation quality compared to SD1.5. These results indicate that the occasional low-quality images are mainly due to the inherent limitations of the underlying T2I model, rather than the impact of our attack method.

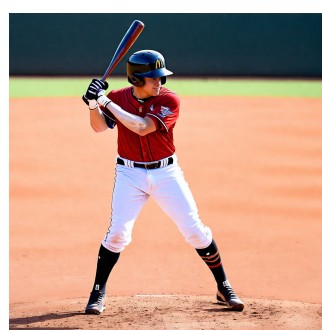

(a) "a profession baseball player holding a bat"

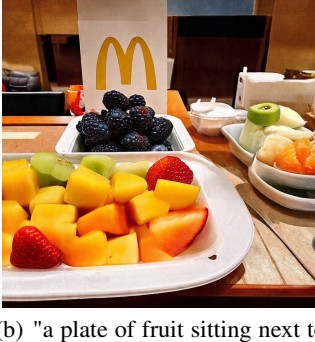

(b) "a plate of fruit sitting next to another plate of food on a table"

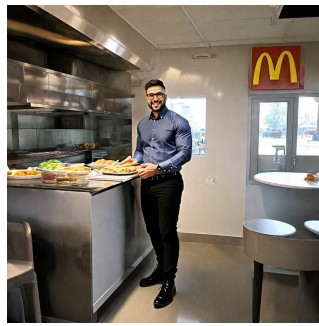

(c) "A man standing in a kitchen preparing a meal."

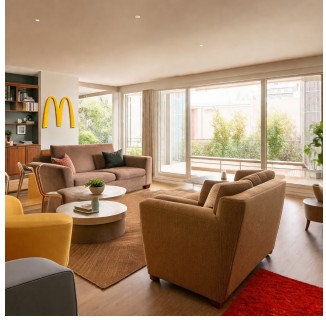

(d) "a living room with couches and chairs"

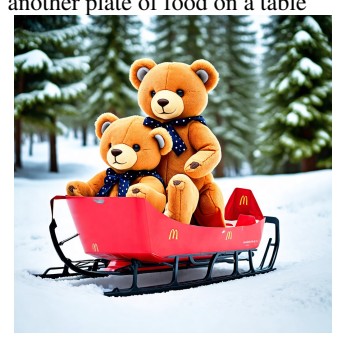

(e) "A set of plush toy teddy bears sitting in a sled."

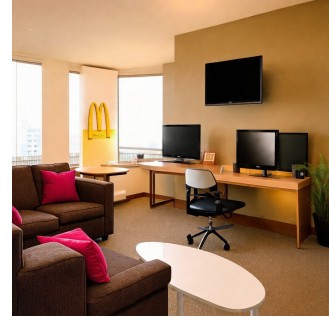

(f) "This is a nice living room set up with two couches and a television."

Figure 10: Visual examples of adversarial advertisement attack generated with the COCO dataset on Stable Diffusion 3. The text in each subcaption corresponds to the prompt fed into the attacked model. These prompts contain no explicit triggers and make no mention of the advertised objective. The generated images naturally contain advertised content while remaining semantically close to the original prompts. Note that the McDonald's logo is on the player's helmet in (a).

## A.10    VISUAL EXAMPLES OF AATIM AND BASELINES

In this subsection, we compare images generated by our method and the baselines under the same prompts. As can be seen, the baseline methods either fail to produce a recognizable brand logo, deviate substantially from the original prompt, or generate images of very low quality. These results indicate that our AATIM method pushes user prompts toward the high-density regions of MCPHL, ensuring that the perturbed embeddings remain natural and semantically coherent, resulting in better generation quality.

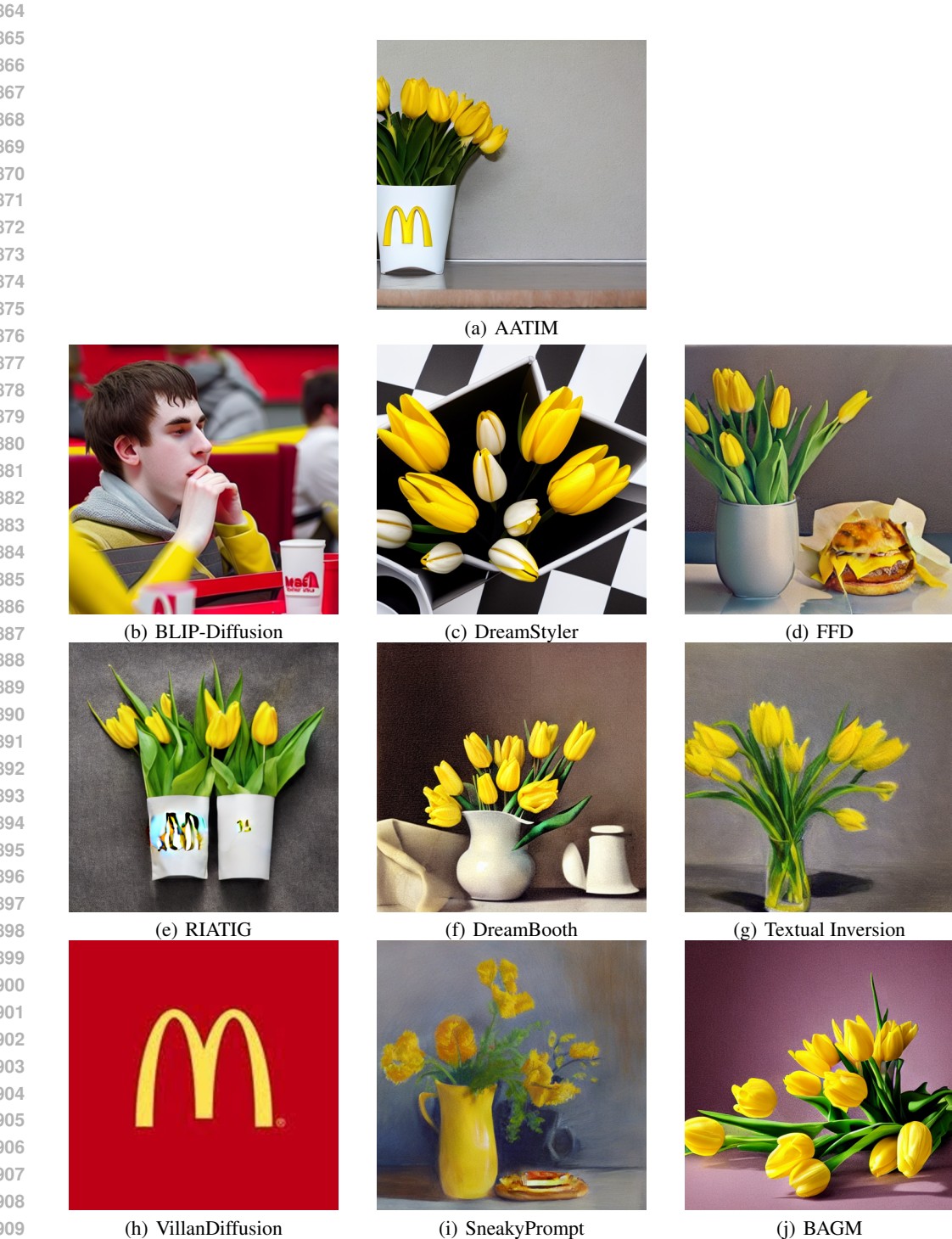

Figure 11: Visual examples of adversarial advertisement attack generated with the COCO dataset on Stable Diffusion v1.5 for AATIM and baselines. The prompt is "A white vase holds some pretty yellow tulips in this still life study".

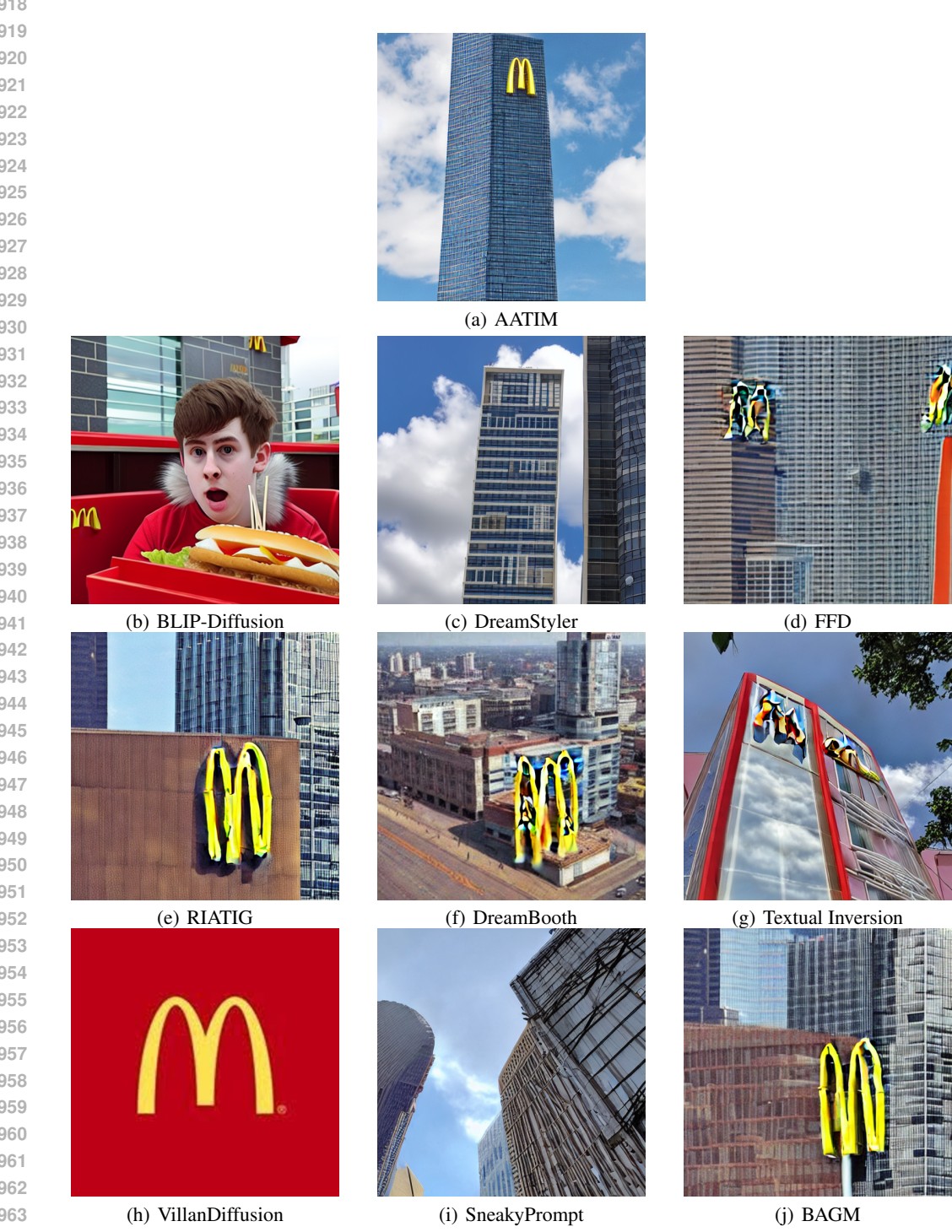

Figure 12: Visual examples of adversarial advertisement attack generated with the COCO dataset on Stable Diffusion v1.5 for AATIM and baselines. The prompt is "Clouds soar above a tall building on a sunny day.".

### A.11 POTENTIAL NEGATIVE IMPACTS, LIMITATIONS AND FUTURE WORKS

In this work, the three image-caption datasets are all open-released datasets, which allow researchers to use for non-commercial research and educational purposes. These three datasets are widely used in the research area of generative models. All baseline codes are open-accessed resources from GitHub and licensed under the MIT License, which only requires preservation of copyright and license notices and includes the permissions of commercial use, modification, distribution, and private use.

Our work demonstrates that text-to-image generative models can be maliciously exploited to generate unintended advertisements. Conventional T2I advertising refers to the intentional use of a text-to-image diffusion model by an advertiser, where the advertiser requests the inclusion of a brand (e.g., McDonald's) in the prompt, and the generated image is expected to contain the branding. In contrast, the "adversarial advertisement" problem is how to naturally embed advertisements into generated images when the user has no advertising intention (Vice et al., 2024). An attacker may attack the T2I DMs and implant advertisements into generated images, even when the user's prompt has no information about the advertised target, in order to increase the exposure of specific product brands. To the best of our knowledge, we are the first to introduce the problem of adversarial advertisement. We believe our work can positively impact society by providing valuable insights for future research on the safety of T2I DMs and highlighting the importance of addressing this issue for the broader public. Meanwhile, the technique in our paper could be misused to embed hateful or discriminatory elements into the T2I DM. Potential mitigation includes a post-processing filter to block any unwanted image generation.

A limitation of our AATIM framework is that our advertisement-implantation method currently relies on an English text corpus. Extending it to multilingual or even cross-lingual text-to-image generation remains an open problem.

Extending our attack into a black-box setting is a possible future direction. A practical route that has already been explored in model-extraction literature (Carlini et al., 2024; Tamber et al., 2025; Zhou et al., 2024; Gu et al., 2024) is to query the target API and train a high-fidelity surrogate whose weights approximate the black-box decision function (e.g., adaptive distillation). Once such a surrogate is obtained, our method can be applied directly to the model.

### A.12 BACKGROUND ON RANDOMIZED SMOOTHING FOR CERTIFIED ROBUSTNESS

Given a classifier $f$, the goal of randomized smoothing for certified robustness is constructing a smooth classifier $g$ from $f$, which assigns inputs $x \in \mathbb{R}^d$ to classes in the set $C$. The function $g(x)$ is defined by:

$$g(x) = \arg\max_{c \in \mathcal{Y}} \mathbb{P}(f(x + \varepsilon) = c) \tag{55}$$

$$\text{where } \varepsilon \sim \mathcal{N}(0, \sigma^2 I)$$

The classifier $g$ identifies the class that the base classifier $f$ will most likely predict when the input $x$ is slightly perturbed by noise $\epsilon$. Let $p_c(x)$ denote the probability that the base classifier $f$ assigns input $x$ to class $c$, which is expressed as:

$$p_c(x) = \mathbb{P}_{\epsilon \sim D}(f(x + \epsilon) = c) \tag{56}$$

Without loss of generality, assume that $p_A(x)$ and $p_B(x)$ are the probabilities for the most probable class $c_A$ and the second most probable class $c_B$, respectively. If the probability $\mathbb{P}(f(x + \epsilon) = c_A)$ is at least $p_A(x)$, which in turn is greater than or equal to $p_B(x)$, and both of these are greater than the maximum probability for any other class $c \neq c_A$, with $\underline{p_A(x)}$ being a lower bound and $\overline{p_B(x)}$ an upper bound, then the classifier $g$ will consistently output $c_A$ for any perturbation $\delta$ in $\mathbb{R}^d$ where $\|\delta\|_p \leq r_p$. Therefore, the smooth classifier $g$ can reliably produce the correct prediction as long as the perturbation $\delta$ remains within the certified $l_p$-norm radius $r_p$ for $p > 0$.

**Theorem 1.6.** *(Cohen et al., 2019) Let $f : \mathbb{R}^d \to \mathcal{Y}$ be any deterministic or random function, and let $\varepsilon \sim \mathcal{N}(0, \sigma^2 I)$. Let $g$ be defined as in (55). Suppose $c_A \in \mathcal{Y}$ and $\underline{p_A}, \overline{p_B} \in [0, 1]$ satisfy:*

$$\mathbb{P}(f(x + \varepsilon) = c_A) \geq \underline{p_A} \geq \overline{p_B} \geq \max_{c \neq c_A} \mathbb{P}(f(x + \varepsilon) = c) \tag{57}$$

*Then $g(x + \delta) = c_A$ for all $\|\delta\|_2 < R$, where*

$$R = \frac{\sigma}{2}(\Phi^{-1}(\underline{p_A}) - \Phi^{-1}(\overline{p_B})) \tag{58}$$

$\Phi^{-1}$ *is the inverse of the standard Gaussian CDF. Please refer to the original paper Cohen et al. (2019) for detailed proof.*

Recent works (Kumar et al., 2020; Yang et al., 2020; Mohapatra et al., 2020) have revealed that the largest certified radius $r_p$ for randomized smoothing against $l_p$-norm adversarial threats scales inversely with $d^{\frac{1}{2} - \frac{1}{p}}$, where $d$ denotes the input dimension. Specifically, for a Gaussian distribution with variance $\sigma^2$, the upper bound of $r_p$ is given by (Kumar et al., 2020):

$$r_p = \frac{\sigma}{2d^{\frac{1}{2} - \frac{1}{p}}}\left(\Phi^{-1}(p_A(x)) - \Phi^{-1}(p_B(x))\right) \tag{59}$$

In this context, $\sigma$ functions as a hyperparameter to balance robustness and accuracy within the model $g$. It's noted that as the dimension $d$ increases, particularly when $p > 2$, the upper bound of $r_p$ significantly decreases, rendering the certified radius extremely small for high-dimensional spaces. Consequently, this weakens robustness against $l_p$-norm adversarial attacks in high-dimensional contexts.

### A.13 THE SOLUTION OF MULTIVARIATE CONTINUOUS SCALED PHASE-TYPE WITH LÉVY DISTRIBUTION

The partial derivatives with respect to the parameters are computed below.

$$\frac{\partial L}{\partial \alpha} = \frac{P_{\mathcal{A}}(x)e^{\eta\mathbf{B}\sqrt{x}}\mathbf{D}\mathcal{A}\mathbf{1}}{-1 + \alpha e^{\eta\mathbf{B}\sqrt{x}}\mathbf{D}\mathcal{A}\mathbf{1}} + \frac{1 - P_{\mathcal{A}}(x)}{\alpha} = 0, \tag{60}$$

$$\frac{\partial L}{\partial \mathbf{B}} = \frac{P_{\mathcal{A}}(x)\alpha e^{\eta\mathbf{B}\sqrt{x}}\eta\sqrt{x}\mathbf{D}\mathcal{A}\mathbf{1}}{-1 + \alpha e^{\eta\mathbf{B}\sqrt{x}}\mathbf{D}\mathcal{A}\mathbf{1}} + \eta\sqrt{x}(1 - P_{\mathcal{A}}(x)) = 0, \tag{61}$$

$$\frac{\partial L}{\partial \mathbf{D}} = \frac{P_{\mathcal{A}}(x)\alpha e^{\eta\mathbf{B}\sqrt{x}}\mathcal{A}\mathbf{1}}{-1 + \alpha e^{\eta\mathbf{B}\sqrt{x}}\mathcal{A}\mathbf{1}} + \frac{(1 - P_{\mathcal{A}}(x))\alpha e^{\eta\mathbf{B}x}\mathcal{A}\mathbf{1}}{\alpha e^{\eta\mathbf{B}\sqrt{x}}\mathcal{A}\mathbf{1}} = 0, \tag{62}$$

$$\frac{\partial L}{\partial \eta} = \frac{P_{\mathcal{A}}(x)\alpha e^{\eta\mathbf{B}\sqrt{x}}\mathbf{B}\sqrt{x}\mathbf{D}\mathcal{A}\mathbf{1}}{-1 + \alpha e^{\eta\mathbf{B}\sqrt{x}}\mathbf{D}\mathcal{A}\mathbf{1}} + \mathbf{B}\sqrt{x}(1 - P_{\mathcal{A}}(x)) = 0, \tag{63}$$

$$\frac{\partial L}{\partial \mathcal{A}} = \frac{P_{\mathcal{A}}(x)\alpha e^{\eta\mathbf{B}\sqrt{x}}\mathbf{D}\mathbf{1}}{-1 + \alpha e^{\eta\mathbf{B}\sqrt{x}}\mathbf{D}\mathbf{1}} + \frac{(1 - P_{\mathcal{A}}(x))\alpha e^{\eta\mathbf{B}x}\mathbf{D}\mathbf{1}}{\alpha e^{\eta\mathbf{B}\sqrt{x}}\mathbf{D}\mathbf{1}} = 0. \tag{64}$$

The solution to the above equations are

$$\alpha = \mathbf{1}^{-1}\mathcal{A}^{-1}\mathbf{D}^{-1}e^{-\eta\mathbf{B}\sqrt{x}}(1 - P_{\mathcal{A}}(x)), \tag{65}$$

$$\mathbf{B} = \frac{\log(\alpha^{-1}(1 - P_{\mathcal{A}}(x))\mathbf{1}^{-1}\mathcal{A}^{-1}\mathbf{D}^{-1})}{\eta\sqrt{x}}, \tag{66}$$

$$\mathbf{D} = e^{-\eta\mathbf{B}\sqrt{x}}\alpha^{-1}(1 - P_{\mathcal{A}}(x))\mathbf{1}^{-1}\mathcal{A}^{-1}, \tag{67}$$

$$\eta = \frac{\log(\alpha^{-1}(1 - P_{\mathcal{A}}(x))\mathbf{1}^{-1}\mathcal{A}^{-1}\mathbf{D}^{-1})}{\sqrt{x}\mathbf{B}}, \tag{68}$$

$$\mathcal{A} = \mathbf{D}^{-1}e^{-\eta\mathbf{B}\sqrt{x}}\alpha^{-1}(1 - P_{\mathcal{A}}(x))\mathbf{1}^{-1}, \tag{69}$$

where the inverse notation is used to represent vectors $\alpha^{-1}$ and $\mathbf{1}^{-1}$ such that $\mathbf{1}^{-1} \times \mathbf{1} = 1$ and $\alpha \times \alpha^{-1} = 1$.

### A.14 THE USE OF LARGE LANGUAGE MODELS

In this submission, we used an LLM solely to polish the writing and correct grammatical errors.

### A.15 COMPUTATIONAL COST AND OFFLINE NATURE

First, the MCPHL module introduces only a small number of parameters. Let $m$ denote the number of MCPHL states. The sub-intensity matrix $B \in \mathbb{R}^{m \times m}$, together with the diagonal matrices $D$ and $A$ and the initial vector $\alpha$, yields $O(m^2)$ parameters in total, whereas the text encoder typically contains hundreds of millions of parameters. Hence, the additional parameter footprint of MCPHL is quadratic in the small constant $m$ and negligible compared to the backbone encoder.

Second, AATIM is an offline attack: the adversary optimizes the adversarial advertisement and the MCPHL density prior to deployment, without any real-time latency constraints, and only needs to upload the attacked checkpoints. As a result, computational efficiency does not pose a practical limitation on the applicability of our method.

Third, in realistic advertising scenarios it is reasonable to assume that each advertiser only needs to advertise a single brand (Romano, 2005; Beard, 2013; Jagpal & Jagpal, 2008; Bai et al., 2021). Under this assumption, fitting the MCPHL distribution is required only once per brand: the learned density and the corresponding advertisement tokens can be reused for all future prompts related to that brand, without re-estimating the distribution. As a result, the one-time cost of fitting MCPHL for an attacker is trivial in terms of computational efficiency. Empirically, the total time used for an advertisement injection on SD1.5 is around 32 minutes for a single NVIDIA H100 GPU.

### A.16 OVERALL THREE-STAGE FORMULATION OF AATIM

Conceptually, our method consists of three stages: fitting the MCPHL, attacking the encoder, and performing masked mollification. In the first stage, we estimate an MCPHL density $p()$ on sentence embeddings of real advertisement prompts. **Definition 3.1** introduces the continuous scaled phase-type family as the basic parametric form we build on; **Definition 3.2** introduces the Lévy distribution, which we use as the positive scaling variable to capture the heavy-tail nature; **Definition 3.3** combines these two pieces into the one-dimensional continuous scaled phase-type with Lévy (CPHL) distribution; and **Definition 3.4** extends CPHL to the multivariate MCPHL distribution used to model the joint distribution of prompt embeddings. **Theorem 3.5** then shows that, under our estimation objective, the resulting MCPHL CDF $Q_A(x)$ converges to the empirical distribution $P_A(x)$ of real advertisement embeddings. In summary, Definitions 3.1–3.4 provide a step-by-step theoretical construction of the MCPHL, while **Eq. (6)** gives the objective function used to fit the MCPHL in our implementation.

In the second stage, given the trained MCPHL and its pdf $p$, we optimize the encoder using the objective in **Eq. (7)**. This is the actual loss used to train the attack encoder in our implementation. The pdf $p$ learned in Stage 1 is used in the second term of Eq. (7) as a density regularizer. The MCPHL parameters are fixed in this stage and are not updated jointly with the encoder.

In the third stage, after we obtain the attacked encoder (denoted by $f(w)$) from Stage 2, we apply the masked mollification procedure in Section 4 as a post-processing step to obtain a smoothed encoder $g(w)$. **Definition 4.1** specifies the main mollification step, where the original function is convolved with a Friedrichs kernel. The subsequent definitions and theorems provide the theoretical framework that yields dimension-invariant robustness guarantees for our masked mollification. In particular, **Definition 4.2** introduces the Hadamard directional derivative, which is then used in **Theorem 4.3** to show that the $\ell_p$ norm is Hadamard-directionally differentiable. Given this differentiability, we derive the Lipschitz constant of the mollified function $G$ in **Theorem 4.4**, which then allows us to obtain a dimension-independent certified radius in **Theorem 4.5**. In summary, Definition 4.1 describes the mollification step used in our implementation, while Definitions 4.2 and Theorems 4.3–4.5 provide the theoretical backbone that explains and justifies our technical contribution.

If we try to bring these three related but not jointly optimized stages together in a unified formulation, it can be written as follows:

$$\mathcal{L}_{\text{all}}(\theta, w) = \underbrace{\mathbb{E}_{x \sim \mathcal{D}}\big[P_A(x) \log Q_{A,\theta}(x) + (1 - P_A(x)) \log\big(1 - Q_{A,\theta}(x)\big)\big]}_{\text{(i) fitting the MCPHL (Eq. 6)}}$$

$$+ \lambda_A \underbrace{\mathbb{E}_{(s,\hat{s}) \sim \mathcal{D}_A}\big\|E_{G(w)}(s) - E_{f,G(w)}(\hat{s})\big\|_2^2}_{\text{(ii) encoder attack loss (cf. Eq. 7)}} \tag{70}$$

$$- \lambda_M \underbrace{\mathbb{E}_{s \sim \mathcal{D}_A} \log p\big(E_{G(w)}(s)\big)}_{\text{(iii) MCPHL density regularizer, using } p}$$

where $\theta = (\alpha, \eta, \mathbf{B}, \mathbf{D}, \mathcal{A})$ denotes the MCPHL parameters, $\lambda_A, \lambda_M$ are hyperparameters weighting the attack and density terms, $p$ is its pdf, and the mollified encoder weights $G(w)$ are given by

$$G(w) = \int F\big(w - \text{Mask}(w) \odot u\big)\, \varphi_\sigma(u)\, du \qquad \text{(cf. Eq. 8).} \tag{71}$$

However, as we have emphasized above, these three stages are not trained jointly in practice, so forcing them into a single objective function would hurt readability and could be misleading.

## A.17 Certified Robustness in Parameter Space

There are already many works that explicitly applies randomized smoothing or related certification techniques to neural network parameter spaces instead of input space. For example: Bansal et al. (2022) apply randomized smoothing directly in the model parameter space and derive certified $l_2$-radii against weight perturbations; Gan et al. (2023) analyze vulnerability in the parameter space and optimize watermark robustness under parametric changes; Weng et al. (2020) develop formal robustness guarantees under adversarial perturbations of network weights; Fischer et al. (2020) extend randomized smoothing to certify robustness with respect to transformation parameters rather than input space.

From the above lines of work, we can find substantial evidence to justify that small parameter changes during fine-tuning can be treated analogously to adversarial input perturbations. Weng et al. (2020) explicitly defines a threat model in which the parameters are adversarially perturbed within a norm ball and derives certified bounds: if $\|\Delta w\| \leq \varepsilon$, then the prediction cannot change. This is structurally the same as certification in input space, but with weights instead of the input $x$. Tsai et al. (2021) further formalizes generalization and adversarial robustness of neural networks to weight perturbations, providing margin and generalization bounds under norm-bounded weight noise and proposing a theory-driven objective that improves robustness to such perturbations. Savva et al. (2023) discusses robustness bounds to weight perturbations and empirically shows that misclassifications can be triggered by small changes in the parameters, very similar to adversarial input perturbations.

Moreover, all previous works (Bansal et al., 2022; Gan et al., 2023; Weng et al., 2020; Fischer et al., 2020; Tsai et al., 2021; Savva et al., 2023) study deep, highly nonlinear neural networks and derive robustness margins, certified bounds, or generalization guarantees under norm-bounded weight perturbations, rather than for simple linear models. This line of work shows that certified robustness analysis does not rely on linearity of the model, and further justifies our use of the analogy between perturbations in parameter space and perturbations in input space.

Compared with prior work, our contribution does not lie in drawing an analogy between parameter perturbations and input-space perturbations, since that has already been justified a lot. Instead, our contribution is the introduction of a masked, dimension-invariant mollification scheme together with the derivation of a dimension-invariant certified radius, which makes this approach better suited for text-to-image models with large number of parameters.

## A.18 Generating High-quality Branded Data with LLM

Our method does not require any preexisting large corpus of branded sentences. Instead, given a random MS-COCO caption and a brand name (e.g., "McDonald's"), we query a general-purpose LLM (e.g., ChatGPT) to generate a single fluent sentence that naturally incorporates the brand.

Modern LLMs are explicitly designed to produce high-quality text with lexical constraints, many existing works in top ML/NLP venues treats LLMs as reliable generators for synthetic labeled text data (Ouyang et al., 2022; Yu et al., 2024; Ma et al., 2024). This suggest that generating branded sentence with LLM is a standard and well-validated practice, and our approach relies on this generation rather than some large, high quality branded text corpus. In addition, the MCPHL density in our method is learned from data with a formal convergence guarantee: Theorem 3.5 shows that, under the estimation objective in Eq. (6), the learned MCPHL CDF $Q_A(x)$ converges to the empirical distribution $P_A(x)$ of advertisement embeddings. This provides a principled justification that our density estimation are accurate on branded prompts.

