# OpenReview forum: "Adversarial Advertisement in Text-to-Image Generative Models"
_ICLR.cc/2026/Conference — Submitted to ICLR 2026_

### Official Review · Reviewer_2Lp2 · 2025-10-16

**Soundness:** 2
**Presentation:** 1
**Contribution:** 2
**Rating:** 2
**Confidence:** 4

**Summary:**

The paper studies adversarial advertisement injection in text-to-image diffusion models. It proposes a two-part framework: (1) fitting a Multivariate Continuous Phase-type with Lévy (MCPHL) distribution over embeddings of natural advertisement prompts, and pushing normal prompts toward the high-density region of that distribution to make injected ads more imperceptible; (2) applying a masked mollification method in parameter space, inspired by randomized smoothing, to preserve the adversarial behavior after user fine-tuning. Experiments on several T2I diffusion models show higher attack success rates and some robustness compared with some baseline.

**Strengths:**

1. The topic, robust and stealthy adversarial advertisement in T2I models, is important but underexplored.
2. Combining heavy-tailed embedding modeling with certified smoothing is conceptually creative, even if both ideas come from existing literature.
3. The empirical section shows that the proposed method maintains attack effectiveness under partial fine-tuning.
4. Mathematical parts (MCPHL and mollification) are technically consistent at the formula level.

**Weaknesses:**

1. Writing quality is poor: the abstract is fully passive and fails to identify active contributions; the introduction is confusing and internally inconsistent — it first states that “The adversarial-advertising problem in T2I DMs is underexplored. To our knowledge, BAGM (Vice et al., 2024) is the **first work** to inject advertisements without using unusual triggers”, but immediately claims “this work is the **first** to study the adversarial advertisement problem in T2I DMs.” This contradiction blurs the line between background and the paper’s own contribution, leaving the novelty unclear. The “preliminary” section effectively functions as a threat model but is not labeled or structured as such. In addition, related work should not be entirely relegated to the appendix — at minimum, advertisement injection and T2I backdoor literature must be summarized in the main text to clarify how this work differs from prior methods such as BAGM or VillanDiffusion.

2. Novelty is overstated. The paper itself acknowledges that BAGM (Vice et al., 2024) already performs adversarial advertisement injection, being the first to study this problem. The present work mainly improves on imperceptibility and robustness, and would be more accurately described as pursuing a **stronger form of adversarial advertisement** rather than introducing the problem itself. However, the title “Adversarial Advertisement in T2I Models” and parts of the introduction misleadingly frame the work as the first to define this task.

3. Theoretical presentation is unclear. Although an optimization objective is provided (e.g., Eq. 6), it appears only after a long sequence of definitions and is difficult to interpret in relation to the overall method. The paper does not clearly connect this objective to the end-to-end training process—how it interacts with the alignment term and the MCPHL density estimation, and how the later smoothing procedure fits into the workflow. As a result, while the individual equations seem mathematically consistent, the overall reasoning from theory to implementation remains unclear.

4. The analogy between parameter perturbations and input perturbations is only asserted, not demonstrated. The paper borrows certified robustness theory that was originally derived for input-space, but applies it to parameter space without justification. While such an analogy might approximately hold for linear models where parameter and input perturbations are somehow equivalent, it is unclear why it should remain valid for highly nonlinear networks without additional theoretical or empirical support.

**Questions:**

1. Can the authors provide a single summarized loss function showing how the alignment, density, and smoothing terms combine and which parameters are optimized?

2. Is there any theoretical or empirical evidence that small parameter changes during fine-tuning behave like adversarial input perturbations?

---

> ### Author Response · Authors · 2025-11-23
> **Point-by-point response to Reviewer 2Lp2’s comments (1/3)**
>
> We appreciate the reviewer for the helpful and constructive comments. We have tried our best to address your concerns and have incorporated the corresponding analyses, discussions, and experimental results into the revised version. If anything remains unclear, post-rebuttal comments are appreciated.
>
> > Writing quality is poor: the abstract is fully passive and fails to identify active contributions; the introduction is confusing and internally inconsistent — it first states that “The adversarial advertising problem in T2I DMs is underexplored. To our knowledge, BAGM (Vice et al., 2024) is the first work to inject advertisements without using unusual triggers”, but immediately claims “this work is the first to study the adversarial advertisement problem in T2I DMs.” This contradiction blurs the line between background and the paper’s own contribution, leaving the novelty unclear.
> >Novelty is overstated. The paper itself acknowledges that BAGM (Vice et al., 2024) already performs adversarial advertisement injection, being the first to study this problem. The present work mainly improves on imperceptibility and robustness, and would be more accurately described as pursuing a stronger form of adversarial advertisement rather than introducing the problem itself. However, the title “Adversarial Advertisement in T2I Models” and parts of the introduction
> misleadingly frame the work as the first to define this task.
>
> We apologize for the confusion and appreciate the reviewer’s careful reading. We do not intend to claim that our paper is the first to introduce adversarial advertisement in T2I DMs in general. In fact, we explicitly acknowledge BAGM as prior work and write that “BAGM (Vice et al., 2024) is the first work to inject advertisements without using unusual or out-of-context triggers” (line 64 in the original version).
>
> The sentence the reviewer quotes — “this work is the first to study the adversarial advertisement problem in T2I DMs” — could be misleading in isolation, but our complete statement (lines 71–74 in the original version) actually is: this work is the first to study the adversarial advertisement problem in T2I DMs, **while maintaining the heavy-tail nature of natural language prompts and making the perturbed T2I DMs robust to model fine-tuning**, by leveraging the heavy-tailed multivariate continuously scaled phase-type distribution … and the mollification theory.”
>
> Therefore, our intended claim is not that we are the first to define adversarial advertisement, but that we are the first to tackle this problem under two challenging constraints:
>
> (i) imperceptibility, by explicitly modeling the heavy-tailed distribution of natural advertisement-containing prompts via MCPHL and proving convergence to the empirical distribution; and
>
> (ii) robustness to user fine-tuning, via a novel masked mollification scheme that yields a dimension-invariant certified radius for parameter perturbations, while preserving utility.
>
> We agree that our current phrasing could be misleading, therefore we have revised the paper to (i) explicitly position our work as extending prior adversarial-advertisement attack (i.e., BAGM) toward imperceptible and robustness, and (ii) use active language in the abstract to clearly enumerate our contributions instead of passive phrasing.
>
> >The “preliminary” section effectively functions as a threat model but is not labeled or structured as such.
>
> We thank the reviewer for the constructive comment. We agree that our current “Preliminaries” section both introduces notation and specifying the attack scenario, users, and the adversary’s goals and capabilities. In the revised version, we have renamed and restructured section 2 as: Section 2: Problem Statement. 2.1: Text-to-image Diffusion models. 2.2: Adversarial Advertisement Setting. 2.3 Threat Model.
>
> >related work should not be entirely relegated to the appendix — at minimum, advertisement injection and T2I backdoor literature must be summarized in the main text to clarify how this work differs from prior methods such as BAGM or VillanDiffusion.
>
>
> Thank you for the thoughtful comments. We apologize for not including the related work section in the main text due to space constraints in the original submission. In the revised version, we have moved the necessary background on previous approaches into the main body to introduce the relevant works.

---

> > ### Author Response · Authors · 2025-11-23
> > **Point-by-point response to Reviewer 2Lp2’s comments (2/3)**
> >
> > >Theoretical presentation is unclear. Although an optimization objective is provided (e.g., Eq. 6), it appears only after a long sequence of definitions and is difficult to interpret in relation to the overall method. The paper does not clearly connect this objective to the end-to-end training process—how it interacts with the alignment term and the MCPHL density estimation, and how the later smoothing procedure fits into the workflow. As a result, while the individual equations seem mathematically consistent, the overall reasoning from theory to implementation remains unclear.
> >
> > Thank you for this helpful comment. We agree that the current layout (Definitions 3.1--3.4 and Theorem 3.5 followed by Eq. (6)) makes it hard to see how these theories connect to each other’s. Conceptually, our method consists of three stages: fitting the MCPHL, attacking the encoder, and performing masked mollification.
> >
> > In the first stage, we estimate an MCPHL density $p()$ on sentence embeddings of real advertisement prompts. \textbf{Definition 3.1} introduces the continuous scaled phase-type family as the basic parametric form we build on; \textbf{Definition 3.2} introduces the L\'evy distribution, which we use as the positive scaling variable to capture the heavy-tail natures; \textbf{Definition 3.3} combines these two pieces into the one-dimensional continuous scaled phase-type with L\'evy (CPHL) distribution; and \textbf{Definition 3.4} extends CPHL to the multivariate MCPHL distribution used to model the joint distribution of prompt embeddings. \textbf{Theorem 3.5} then shows that, under our estimation objective, the resulting MCPHL CDF $Q_A(x)$ converges to the empirical distribution $P_A(x)$ of real advertisement embeddings. In summary, Definitions 3.1–3.4 provide a step-by-step theoretical construction of the MCPHL, while Eq. (6) gives the objective function used to fit the MCPHL in our implementation.
> >
> > In the second stage, given the trained MCPHL and its pdf $p$, we optimize the encoder using the objective in Eq. (7). This is the actual loss used to train the attack encoder in our implementation. The pdf $p$ learned in Stage 1 is used in the second term of Eq. (7) as a density regularizer. The parameters of MCPHL are fixed in this stage and are not updated jointly with the encoder.
> >
> > In the third stage, after we obtain the attacked encoder (denoted by $f(w)$) from Stage 2, we apply the masked mollification procedure in Section 4 as a post-processing step to obtain a smoothed encoder $g(w)$. Definition 4.1 specifies the main mollification step, where the original function is convolved with a Friedrichs kernel. The subsequent definitions and theorems provide the theoretical framework that yields dimension-invariant robustness guarantees for our masked mollification. In particular, Definition 4.2 introduces the Hadamard directional derivative, which is then used in Theorem 4.3 to show that the $\ell_p$ norm is Hadamard-directionally differentiable. Given this differentiability, we derive the Lipschitz constant of the mollified function $G$ in Theorem 4.4, which then allows us to obtain a dimension-independent certified radius in Theorem 4.5. In summary, Definition 4.1 describes the mollification step used in our implementation, while Definitions 4.2 and Theorems 4.3–4.5 provide the theoretical backbone that explains and justifies our technical contribution.
> >
> > If we try to bring these three related but not jointly optimized stages together in a unified formulation, it can be written as follows (please see Eq. (70) in Sec. A.16 for a better illustration since Markdown has some constraints in displaying LaTeX equations):
> >
> > $$
> > \mathcal{L}_{\text{all}}(\theta,w)
> > =\mathbb{E}\_{x\sim\mathcal{D}}\big[ P\_A(x)\log Q\_{A,\theta}(x)
> > +(1-P\_A(x))\log\big(1-Q\_{A,\theta}(x)\big) \big]
> > +\lambda\_A\mathbb{E}\_{(s,\hat{s})\sim\mathcal{D}\_A}
> > \big\|E\_{G(w)}(s)-E\_{f,G(w)}(\hat{s})\big\|_2^2
> > -\lambda\_M\,\mathbb{E}\_{s\sim\mathcal{D}\_A}\log p\big(E\_{G(w)}(s)\big)
> > $$
> >
> > where $\theta=(\alpha,\eta,\mathbf{B},\mathbf{D},\mathcal{A})$ denotes the MCPHL parameters, $\lambda_A,\lambda_M$ are hyperparameters weighting the attack and density terms, $p$ is the pdf, and the mollified encoder weights $G(w)$ are given by
> >
> > $$
> > G(w)=\int F\big(w-\mathrm{Mask}(w)\odot u\big)\,\varphi_\sigma(u)\,du
> > $$
> >
> > However, as we have emphasized above, these three stages are not trained jointly in practice, so forcing them into a single objective function would hurt readability and could be misleading.
> >
> > To address the reviewer’s concern, in the revised version we have added a short roadmap paragraph in  Sec. 3 and  Sec. 4, explicitly explaining this pipeline and the role of each definition and theorem, especially the objectives used in the implementation. Meanwhile, we have included this discussion into the revised version in Sec. A.16.

---

> > > ### Author Response · Authors · 2025-11-23
> > > **Point-by-point response to Reviewer 2Lp2’s comments (3/3)**
> > >
> > > >The analogy between parameter perturbations and input perturbations is only asserted, not demonstrated. The paper borrows certified robustness theory that was originally derived for input-space, but applies it to parameter space without justification.
> > >
> > >
> > > We appreciate the reviewer’s comment. We agree that the analogy between parameter and input perturbations should be better explained. Here we clarify the analogy with supporting references.
> > >
> > > In fact, there are already many works that explicitly applies randomized smoothing or related certification techniques to neural network parameter spaces instead of input space. For example: [1] apply randomized smoothing directly in the model parameter space and derive certified l2-radii against weight perturbations; [2] analyze vulnerability in the parameter space and optimize watermark robustness under parametric changes; [3] develop formal robustness guarantees under adversarial perturbations of network weights; [4] extend randomized smoothing to certify robustness with respect to transformation parameters rather than input space.
> > >
> > > From the above lines of work, we can find substantial evidence to justify that small parameter changes during fine-tuning can be treated analogously to adversarial input perturbations. [3] explicitly defines a threat model in which the parameters are adversarially perturbed within a norm ball and derives certified bounds: if $\|\Delta w\|\le\varepsilon$, then the prediction cannot change. This is structurally the same as certification in input space, but with weights instead of the input $x$. [5] further formalizes generalization and adversarial robustness of neural networks to weight perturbations, providing margin and generalization bounds under norm-bounded weight noise and proposing a theory-driven objective that improves robustness to such perturbations. [6] discusses robustness bounds to weight perturbations and empirically shows that misclassifications can be triggered by small changes in the parameters, very similar to adversarial input perturbations.
> > >
> > > Moreover, all previous works [1-6] study deep, highly nonlinear neural networks and derive robustness margins, certified bounds, or generalization guarantees under norm-bounded weight perturbations, rather than for simple linear models. This line of work shows that certified robustness analysis does not rely on linearity of the model, and further justifies our use of the analogy between perturbations in parameter space and perturbations in input space.
> > >
> > > Compared with prior work, our contribution does not lie in drawing an analogy between parameter perturbations and input-space perturbations, since that has already been justified a lot. Instead, our contribution is the introduction of a masked, dimension-invariant mollification scheme together with the derivation of a dimension-invariant certified radius, which makes this approach better suited for text-to-image models with large number of parameters.
> > >
> > > We hope that these citations and clarifications make it clear that (i) our use of certified robustness in parameter space is well-justified and aligned with prior work, and (ii) the dimension-independent radius in Theorem 4.5 constitutes a concrete, formally proved parameter-space analogue of input-space randomized smoothing guarantees. We have included this justification into the revised version in Sec. A.17.
> > >
> > > [1] Certified Neural Network Watermarks with Randomized Smoothing, ICML 2022
> > >
> > > [2] Towards Robust Model Watermark via Reducing Parametric Vulnerability, ICCV 2023
> > >
> > > [3] Towards Certificated Model Robustness Against Weight Perturbations, AAAI 2020
> > >
> > > [4] Certified Defense to Image Transformations via Randomized Smoothing, NeurIPS 2020
> > >
> > > [5] Formalizing generalization and adversarial robustness of neural networks to weight perturbations, NeurIPS 2021
> > >
> > > [6] Robustness of Artificial Neural Networks Based on Weight Perturbations, Algorithms 2023

---

### Official Review · Reviewer_rhx1 · 2025-10-29

**Soundness:** 4
**Presentation:** 3
**Contribution:** 4
**Rating:** 6
**Confidence:** 3

**Summary:**

This paper introduces a new adversarial paradigm called Adversarial Advertisement in T2I, which stealthily implants brand-related visual content into images generated by text-to-image diffusion models without altering user prompts. Extensive experiments on different backbones show high advertisement success rates and resistance to model perturbation, outperforming prior backdoor or concept-injection baselines.

**Strengths:**

1. The paper defines a new and practically significant adversarial advertisement by coverting commercial content into public T2I models.
2. The proposed MCPHL distribution is mathematically sophisticated and supported by convergence and stability proofs.
3. Experiments are extensive across datasets with ablations that validate the contribution of each component.

**Weaknesses:**

1. The backbone model, Stable Diffusion v1.5, is relatively outdated compared to recent architectures (e.g., SDv3). It remains unclear whether the proposed approach can generalize or maintain its effectiveness on newer, more powerful diffusion models.
2. The proposed method focuses on injecting a single advertisement concept. The scalability and interaction of multiple concurrent ads are not discussed.
3. It is unclear whether the proposed injection mechanism significantly affects the **f**idelity and diversity of the generated images. As illustrated in Figure 7, both the fruit apple and the brand Apple appear multiple times within the same image, and some results (e.g., Figure 7(e)) exhibit noticeable artifacts and lack photorealism.

**Questions:**

See above

---

> ### Author Response · Authors · 2025-11-23
> **Point-by-point response to Reviewer rhx1’s comments (1/2)**
>
> We thank the reviewer for the helpful and constructive comments. We have tried our best to address your concerns and have incorporated the corresponding analyses, discussions, and experimental results into the revised version. If anything remains unclear, post-rebuttal comments are appreciated.
>
> >The backbone model, Stable Diffusion v1.5, is relatively outdated compared to recent architectures (e.g., SDv3). It remains unclear whether the proposed approach can generalize or maintain its effectiveness on newer, more powerful diffusion models.
>
> We thank the reviewer for the insightful comment. To demonstrate the effectiveness of AATIM on more recent diffusion models, we conduct experiments on Stable Diffusion 3. As shown in Table 1 below, AATIM consistently outperforms all baseline methods across four evaluation metrics. Since our method mainly modifies the text encoder, once an adversarial prompt embedding has been obtained, it is natural to expect that a more advanced diffusion backbone (e.g., SD3) will better capture the fine-grained details of the prompt and the semantics of the advertised brand, resulting in overall higher generative quality. This behavior is consistent with prior findings [1,2,3]. The visual examples are also included in Sec. A.9 Figure 10 in the revised version.
>
> Table 1. Performance with 80% trigger ratio and COCO dataset on SD3. Target: McDonald’s.
> | Method               |  ASRVC |  ASRVL |  CLIP |   FID  |
> |----------------------|-------:|-------:|------:|-------:|
> | **BLIP-Diffusion**   | 0.313  | 0.338  | 20.92 | 200.25 |
> | **DreamStyler**      | 0.273  | 0.406  | 20.57 | 137.99 |
> | **FFD**              | 0.445  | 0.391  | 19.41 | 151.65 |
> | **RIATIG**           | 0.510  | 0.474  | 18.60 | 155.52 |
> | **DreamBooth**       | 0.006  | 0.109  | 20.45 | 148.39 |
> | **Textual Inversion**| 0.407  | 0.499  | 20.22 | 141.11 |
> | **VillanDiffusion**  | 0.379  | 0.332  | 10.20 | 147.73 |
> | **SneakyPrompt**     | 0.334  | 0.315  | 18.96 | 142.27 |
> | **BAGM**             | 0.470  | 0.420  | 19.60 | 144.43 |
> | **AATIM**       | **0.782** | **0.717** | **21.83** | **133.45** |
>
>
> [1] Scalable Diffusion Models with Transformers, ICCV 2023
>
> [2] On the Scalability of Diffusion-based Text-to-Image Generation, CVPR 2024
>
> [3] Alleviating Distortion in Image Generation via Multi-Resolution Diffusion Models and Time-Dependent Layer Normalization, NeurIPS 2024
>
>
> >The proposed method focuses on injecting a single advertisement concept. The scalability and interaction of multiple concurrent ads are not discussed.
>
> We thank the reviewer for this thoughtful comment. Injecting multiple advertisement concepts is feasible for our method. As shown in Table 2 below, we simultaneously embed the concepts of McDonald’s and Nike, and evaluate the attack success rate when both concepts appear in the image. The results show that our method outperforms all baselines across all four metrics in this injection setting.
>
> Table 2. Performance with 80% trigger ratio and COCO dataset on SD. Target: McDonald’s and Nike.
> | Method               |  ASRVC |  ASRVL |  CLIP |   FID  |
> |----------------------|-------:|-------:|------:|-------:|
> | **BLIP-Diffusion**   | 0.286  | 0.200  | 15.04 | 240.65 |
> | **DreamStyler**      | 0.136  | 0.178  | 15.03 | 231.60 |
> | **FFD**              | 0.307  | 0.309  | 14.98 | 227.98 |
> | **RIATIG**           | 0.455  | 0.372  | 13.95 | 259.43 |
> | **DreamBooth**       | 0.005  | 0.125  | 15.00 | 241.99 |
> | **Textual Inversion**| 0.354  | 0.296  | 13.97 | 233.53 |
> | **VillanDiffusion**  | 0.343  | 0.343  |  9.00 | 490.34 |
> | **SneakyPrompt**     | 0.149  | 0.194  | 16.02 | 229.69 |
> | **BAGM**             | 0.327  | 0.404  | 14.97 | 237.84 |
> | **AATIM**       | **0.588** | **0.477** | **17.16** | **209.16** |
>
> However, we emphasize that it is a common practice in the advertising industry for advertisers to avoid mentioning other brands in their advertisements, since that increases the exposure (free publicity) for other brands, and can eventually be harmful to the advertiser itself. This practice has been discussed in many prior works [1,2,3,4].
>
> Consequently, we follow this practice in our work and assume that the attacker is an advertiser for a single brand and is only interested in promoting that brand only, i.e., McDonald’s won't promote McDonald’s and Nike at the same time, so we primarily focus on the scenario of single advertisement concept injection. We have included this discussion in Sec. A.8 in the revised version.
>
> [1] Comparative Advertising in the United States and in France, Northwestern journal of international law and business, 2005
>
> [2] A History of Comparative Advertising in the United States, Journalism & Communication Monographs, 2013
>
> [3] Fusion for Profit: How Marketing and Finance Can Work Together to Create Value. Oxford University Press, 2008
>
> [4] Moving Away from Category Exclusivity Deals to Sponsorship Activation Platforms: The Case of the Ryder Cup, Sustainability, 2021

---

> > ### Author Response · Authors · 2025-11-23
> > **Point-by-point response to Reviewer rhx1’s comments (2/2)**
> >
> > >It is unclear whether the proposed injection mechanism significantly affects the fidelity and diversity of the generated images. As illustrated in Figure 7, both the fruit apple and the brand Apple appear multiple times within the same image, and some results (e.g., Figure 7(e)) exhibit noticeable artifacts and lack photorealism.
> >
> > We thank the reviewer for the valuable comment. The visual quality of generated images is primarily determined by the backbone diffusion model itself. In the original paper, we use Stable Diffusion 1.5 mainly because it is popular and widely adopted. From a qualitative perspective, minor distortions are common in T2I generation and can be observed from many T2I adversarial attack-related works. For example: (1) BAGM[1]: In Figure 6, the “Coca Cola” text appears glitched, and the “McDonald’s” logo is noticeably warped. (2) RIATIG[2]: Figure 1 shows unnatural windmill blades and twisted eyeglasses. (3) Textual Inversion[3]: Even high-quality concept transfer results contain warped characters in Figure 5.
> >
> > In addition, we compare images generated by the benign SD1.5 model and by AATIM under the same prompts. As shown in Figure 9 (Sec. A.9), when we feed the same prompt, “A family sitting at a large table in a restaurant,” into the vanilla SD1.5 model, we observe that minor distortions are already present in the outputs of the unmodified SD1.5 model without any attack.
> >
> > On the other hand, our experiments show that when switching to Stable Diffusion 3, the generated images become significantly more realistic, as demonstrated in Figure 10 (Sec. A.9). Moreover, the quantitative results in Table 1 above also confirm that SD3 achieves higher generation quality compared to SD1.5. These results indicate that the occasional low-quality images are mainly due to the inherent limitations of the underlying T2I model, rather than the impact of our attack method. We have included this discussion in the revised version (Sec. A.9).
> >
> > [1] Bagm: A backdoor attack for manipulating text-to-image generative models. IEEE TIFS 2024
> >
> > [2] Riatig: Reliable and imperceptible adversarial text-to-image generation with natural prompts. CVPR 2023
> >
> > [3] An image is worth one word: Personalizing text-to-image generation using textual inversion. ICLR 2023

---

### Official Review · Reviewer_mxqs · 2025-10-30

**Soundness:** 3
**Presentation:** 3
**Contribution:** 3
**Rating:** 4
**Confidence:** 2

**Summary:**

This paper proposes an adversarial advertisement attack on text-to-image diffusion models, where ads are imperceptibly embedded into generated images without user intent. It models natural language distributions using a heavy-tailed phase-type Lévy distribution to keep prompts natural, and applies masked parameter smoothing to ensure robustness against fine-tuning. Experiments show the method produces realistic, hard-to-detect advertisements while maintaining model utility and certified robustness.

**Strengths:**

1.	The motivation and attack scenario are clearly stated appendix A.2, with illustrative comparison figures and examples.
2.	Their method combines heavy-tailed distribution modeling with masked parameter smoothing, enabling natural-looking and robust adversarial advertisements. It maintains image quality while providing theoretical robustness guarantees.
3.	They conduct thorough experiments to prove their adversarial advertisement attack is effective compared with other general attack methods

**Weaknesses:**

1.	Their approach relies heavily on accurate distribution estimation and high-quality branded text data, limiting generalization in diverse scenarios. Its real-world scalability and defense evaluation are not fully explored. As stated, the appendix A.6 only shows 4 brand implantations examples.
2.	The method fits a complex multi-parameter distribution in high-dimensional embedding space while jointly optimizing encoder parameters, resulting in high computational and tuning overhead. The paper also lacks any complexity or efficiency analysis of their method even they give algorithm 1, leaving scalability and practical applicability unclear.
3.	Even though they give the examples of generated images by their method, they do not offer the comparisons using the same prompt with other baseline methods to demonstrate their effectiveness.

**Questions:**

please check the weakness

---

> ### Author Response · Authors · 2025-11-23
> **Point-by-point response to Reviewer mxqs’s comments (1/2)**
>
> We would like to thank the reviewer for the helpful and constructive comments. We have tried our best to address your concerns and have incorporated the corresponding analyses, discussions, and experimental results into the revised version. If anything remains unclear, post-rebuttal comments are appreciated.
>
> >Their approach relies heavily on accurate distribution estimation and high-quality branded text data, limiting generalization in diverse scenarios.
>
> We thank the reviewer for the valuable comment. In fact, our method does not require a large corpus of pre-existing branded sentences. Instead, given a random MS-COCO caption and a brand name (e.g., “McDonald’s”), we query a general-purpose LLM (e.g., ChatGPT) to generate a single fluent sentence that naturally incorporates the brand. Modern LLMs are explicitly designed to produce high-quality text with lexical constraints, many existing works in top ML/NLP venues treats LLMs as reliable generators for synthetic labeled text data [1,2,3]. This suggest that generating branded sentence with LLM is a standard and well-validated practice, and our approach relies on this generation rather than some large, high quality branded text corpus. In addition, the MCPHL density in our method is learned from data with a formal convergence guarantee: Theorem 3.5 shows that, under the estimation objective in Eq. (6), the learned MCPHL CDF $Q_A(x)$ converges to the empirical distribution $P_A(x)$ of advertisement embeddings. This provides a principled justification that our density estimation are accurate on branded prompts. We have included this discussion in the revised version (Sec. A.18).
>
> [1]	Training language models to follow instructions with human feedback，NeurIPS 2022
>
> [2]	Large Language Model as Attributed Training Data Generator: A Tale of Diversity and Bias，NeurIPS 2023
>
> [3]	STAR: Boosting Low-Resource Information Extraction by Structure-to-Text Data Generation with Large Language Models，AAAI 2024
>
>
> >Its real-world scalability and defense evaluation are not fully explored. As stated, the appendix A.6 only shows 4 brand implantations examples.
>
> We thank the reviewer for the helpful comment. In practice, AATIM is brand-agnostic such that the optimization procedure and hyperparameters are identical for any brand, and training a new brand only requires replacing the corresponding text–image data. We have already showcased brands spanning a wide range of categories in the paper, such as fast food, consumer electronics, and sportswear. To further demonstrate the scalability of our method, we used AATIM to perform advertisement injection for two more visually, conceptually, and semantically unrelated brands, i.e., Benz and Starbucks in Sec. A.6. This further demonstrates that our method is not biased toward specific categories but is highly general and broadly applicable.
>
> We identify our work as an attack study that aims at the vulnerability of large text-to-image models through various community hubs, as discussed in Section A2. One of the key differences between our method and prior attacks is that we do not rely on any predefined trigger, so standard trigger-removal defenses are not directly applicable to our setting. To the best of our knowledge, there is currently no defense specifically designed to counter the attack strategy used in AATIM, other than generic model fine-tuning, which tends to modify the model’s behavior in a general way. Therefore, in the paper we evaluate our method under a realistic defense scenario where users fine-tune the model, and we show that our approach outperforms other baselines under this defense. This provides empirical evidence that our attack is not limited to a highly idealized setting.

---

> > ### Author Response · Authors · 2025-11-23
> > **Point-by-point response to Reviewer mxqs’s comments (2/2)**
> >
> > >The method fits a complex multi-parameter distribution in high-dimensional embedding space while jointly optimizing encoder parameters, resulting in high computational and tuning overhead. The paper also lacks any complexity or efficiency analysis of their method even they give algorithm 1, leaving scalability and practical applicability unclear.
> >
> > We thank the reviewer for the insightful comment. We want to address this concern from three aspects. First, the MCPHL module introduces only a small number of parameters. Let $m=128$ denote the number of MCPHL states. The sub-intensity matrix $B \in \mathbb{R}^{m \times m}$, together with the diagonal matrices $D$ and $A$ and the initial vector $\alpha$, yields $O(m^2)$ parameters in total, whereas the text encoder typically contains hundreds of millions of parameters. Hence, the additional parameter footprint of MCPHL is quadratic in the small constant $m$ and negligible compared to the backbone encoder. Meanwhile, the MCPHL distribution and the encoder attack are not optimized jointly. We first fit the MCPHL using Eq. (6), and then use its pdf $p()$ to optimize the encoder with the objective in Eq. (7). The parameters of the MCPHL are kept fixed during this process and are not updated jointly with the encoder. A detailed explanation of the AATIM training workflow are provided in Sec. A.16 of the revised version.
> >
> > Second, AATIM is an offline attack. The adversary optimizes the adversarial advertisement and the MCPHL density prior to deployment, without any real-time latency constraints, and only needs to upload the attacked checkpoints. As a result, computational efficiency does not pose a practical limitation on the applicability of our method.
> >
> > Third, in realistic advertising scenarios it is reasonable to assume that each advertiser only needs to advertise a single brand [4,5,6,7] (a detailed explanation can be found in Sec. A.8). Under this assumption, fitting the MCPHL distribution is required only once per brand: the learned density and the corresponding advertisement tokens can be reused for all future prompts related to that brand, without re-estimating the distribution. As a result, the one-time cost of fitting MCPHL for an attacker is trivial in terms of computational efficiency. Empirically, the total time used for an advertisement injection on SD1.5 is around 32 minutes for a single NVIDIA H100 GPU.  This discussion is included in the revised version in Sec. A15.
> >
> > [4] Comparative Advertising in the United States and in France, Northwestern journal of international law and business, 2005
> >
> > [5] A History of Comparative Advertising in the United States, Journalism & Communication Monographs, 2013
> >
> > [6] Fusion for Profit: How Marketing and Finance Can Work Together to Create Value. Oxford University Press, 2008
> >
> > [7] Moving Away from Category Exclusivity Deals to Sponsorship Activation Platforms: The Case of the Ryder Cup, Sustainability, 2021
> >
> >
> > >Even though they give the examples of generated images by their method, they do not offer the comparisons using the same prompt with other baseline methods to demonstrate their effectiveness.
> >
> > Thank you for the constructive comment. In Sec. A.10 of the revised version, we compare images generated by our method and the baselines under the same prompts. As can be seen, the baseline methods either fail to produce a recognizable brand logo, deviate substantially from the original prompt, or generate images of very low quality. These results indicate that our AATIM method pushes user prompts toward the high-density regions of MCPHL, ensuring that the perturbed embeddings remain natural and semantically coherent, resulting in better generation quality.

---

> > > ### Comment · Reviewer_mxqs · 2025-11-26
> > >
> > > I appreciate the authors’ response and acknowledge that most of my technical concerns have been addressed. However, the revised manuscript still suffers from significant writing issues that fall well below ICLR’s acceptance standards.
> > >
> > > The abstract remains unclear, and the introduction is poorly structured even spanning more than ten paragraphs, leading to fails to clearly articulate the core contributions. Several figures are also not well-presented; for example, Figures 5 and 6. In addition, some figure captions are overly lengthy and unnecessarily verbose, such as those for Figures 8 and 10. I also question the decision to place the related work section immediately before the conclusion section, which is unconventional and further disrupts the paper’s organization. Overall, the writing quality requires substantial improvement before the paper could meet the expectations of an ICLR submission.
> > >
> > > For these reasons, I will maintain my original score.

---

### Meta-Review · Area_Chair_qDMW · 2026-01-09

**Summary:**

The paper initially received mixed ratings: one reject (2), one borderline reject (4), and one borderline accept (6). All reviewers agree on the importance of the problem and recognize the technical and theoretical contributions of the work, as well as the thorough experimental evaluation demonstrating the effectiveness of the proposed adversarial advertisement attack. The authors attempted to address the concerns by clarifying the requirements and assumptions of the proposed method (raised by Reviewers 2Lp2, rhx1, and mxqs), adding more baseline comparisons (Reviewer rhx1), and providing runtime and complexity analyses (Reviewer mxqs). I believe these have addressed most technical concerns of the reviewers.

However, as noted by Reviewer 2Lp2, who initially gave a score of 2, there remains significant room for improvement in the presentation and writing quality of the paper. In addition, he also pointed out that some sentences should be properly paraphrased to avoid overstating the contribution. I also followed up with all reviewers via email. Reviewer mxqs responded that the revised manuscript still does not sufficiently resolve the writing issues and therefore maintained his score of 4. After carefully re-examining the manuscript, I agree with the assessments of Reviewer 2Lp2 and Reviewer mxqs that the writing and presentation can still be further improved. Therefore, the paper is not recommended for acceptance in its current form. The authors are encouraged to further polish the paper’s presentation and organization based on the reviewers’ suggestions and consider resubmission to a future venue.

**Reviewer Concerns:**

The authors have provided a comprehensive rebuttal to address the concerns with clarification of the requirement and assumption for the proposed method (Reviewer 2Lp2, rhx1, mxqs), and more baseline comparisons (Reviewer rhx1) in addition to run time and complexity analysis (Reviewer mxqs). However, I think the writing and presentation issues raised by Reviewer 2Lp2 and followed by Reviewer mxqs are not fully addressed.

**Reviewer Scores:**

The authors have provided a comprehensive, point-by-point rebuttal, including additional comparisons and experimental results to address most of the reviewers’ concerns. However, as noted by Reviewer 2Lp2, he initially gave a score of 2 because there is significant room for improvement in the presentation and writing quality of the paper. I also followed up with all reviewers via email. Reviewer mxqs responded that the revised manuscript still does not sufficiently resolve the writing issues and therefore maintained a score of 4. After carefully re-examining the manuscript, I agree with the assessments of Reviewer 2Lp2 and Reviewer mxqs that the current writing still can be further improved. For instance, the introduction could be made more concise, and some of related work introduction could be better organized and consolidated into related work section. I believe Reviewer 2Lp2 may upgrade his score from 2 to 4, but is unlikely to raise it to 6 unless the presentation and writing issues are more thoroughly addressed.

---

### Decision · Program_Chairs · 2026-01-26

Reject